# Causal-Adapter: Taming Text-to-Image Diffusion for Faithful Counterfactual Generation

**Lei Tong** [* 1] **Zhihua Liu** [* 2 3] **Chaochao Lu** [4] **Dino Oglic** [1] **Tom Diethe** [1] **Philip Teare** [1] **Sotirios A. Tsaftaris** [2 3] **Chen Jin** [1]

## Abstract

We present Causal-Adapter, a modular framework that adapts frozen text-to-image diffusion for counterfactual generation. Our method enables causal interventions on target attributes, consistently propagating their effects to causal dependents without altering the core identity of the image. In contrast to prior approaches that rely on prompt engineering without explicit causal mechanism, Causal-Adapter leverages structural causal modeling augmented with two attribute regularization strategies: prompt-aligned injection, which aligns causal attributes with textual embeddings for precise semantic control, and a conditioned token contrastive loss to disentangle attribute factors and reduce spurious correlations. Causal-Adapter achieves state-of-the-art performance on both synthetic and real-world datasets, with up to 91% MAE reduction on Pendulum for accurate attribute control and 87% FID reduction on ADNI for high-fidelity MRI generation. These results show that our approach enables robust, generalizable counterfactual editing with faithful attribute modification and strong identity preservation. Code, training data, and models are available on the project page.

## 1. Introduction

Answering counterfactual questions (*e.g.* inferring what an event would have happened under an alternative action) requires understanding the cause–effect relationships among variables and performing hypothetical reasoning (Pearl, 2009; Schölkopf et al., 2021; Weinberg et al., 2024). Clas-

[1]Centre for AI, DS&AI, Astrazeneca, UK [2]Causality in Healthcare AI Hub (CHAI), UK [3]Institute for Imaging, Data and Communications (IDCOM), School of Engineering, University of Edinburgh, Edinburgh, UK [4]Shanghai Artificial Intelligence Laboratory. Correspondence to: Chen Jin <chen.jin@astrazeneca.com>.

*Proceedings of the 43rd International Conference on Machine Learning*, Seoul, South Korea. PMLR 306, 2026. Copyright 2026 by the author(s).

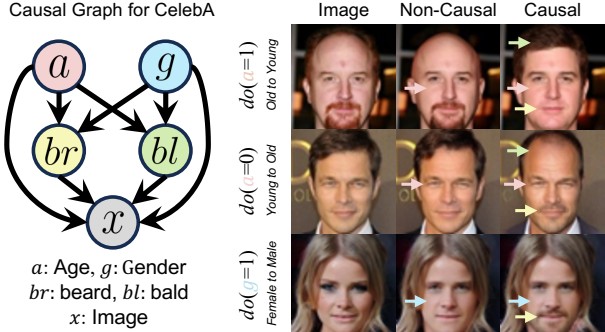

**Causal Graph for CelebA**

$a$: Age, $g$: Gender
$br$: beard, $bl$: bald
$x$: Image

Image | Non-Causal | Causal

$do(a=1)$ *Old to Young*
$do(=0)$ *Young to Old*
$do(g=1)$ *Female to Male*

*Figure 1.* Non-causal editing modifies only the target attribute (*e.g.* age, gender); causal editing propagates changes to related attributes (*e.g.* beard, baldness) enforced by the causal graph.

sical generative models typically tackle counterfactual-style tasks such as image editing or style transfer in a non-causal perspective. **Taking a pre-defined causal graph and semantic attributes as standard inputs**, subsequent work augments such models with explicit structural causal models (SCMs) to implement *abduction–action–prediction* (Pearl, 2013; Pawlowski et al., 2020; De Sousa Ribeiro et al., 2023; Komanduri et al., 2024c; Rasal et al., 2025). This design drives advances in counterfactual image generation (Figure 1) by enforcing edits consistent with a causal graph which enables critical domain-specific counterfactual generation applications, such as simulating medical images with fine-grained anatomical changes associated with aging or disease progression to improve clinical interpretation (Starck et al., 2025). Faithful counterfactual generation remains challenging, as real-world attributes are often causally entangled (*e.g.*, only males can grow beards). This entanglement complicates disentangling factors and generalizing edits, making it nontrivial to ensure that interventions yield the intended visual effect while keeping non-intervened attributes invariant and preserving identity-specific details (Komanduri et al., 2024a).

**Counterfactual Image Generation.** Early approaches modeled counterfactual image generation using normalizing flows (Papamakarios et al., 2021; Winkler et al., 2020), variational autoencoders (VAEs) (Kingma & Welling, 2013; Yang et al., 2021), hierarchical VAEs (Vahdat & Kautz, 2020;

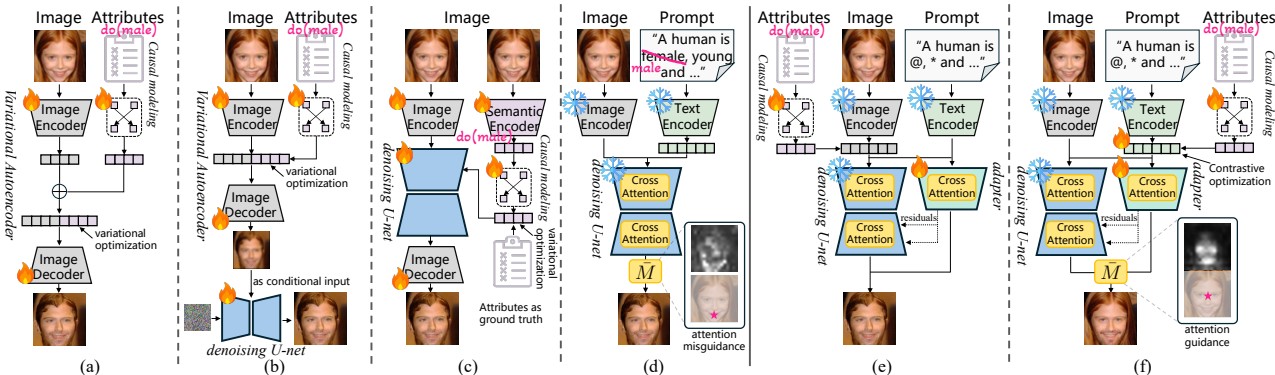

*Figure 2.* A sketch comparison of counterfactual image generation methods based on: (a) **VAE or GAN**, which fail to achieve high-fidelity results. (b) **Diffusion SCM** and (c) **Diffusion autoencoder**, which are sensitive to spurious correlations. (d) **T2I based editing**, which requires heavy prompt engineering. (e) **Vanilla Causal-Adapter**, which injects causal attributes into image-embedding. (f) **Causal-Adapter with attribute regularization**, which injects causal attributes into learnable textual embeddings with contrastive optimization. Detailed discussion is presented in Appendix A.

De Sousa Ribeiro et al., 2023), and generative adversarial networks (GANs) (Goodfellow et al., 2014; Kocaoglu et al., 2018), and encouraged attribute disentanglement through variational objectives (Higgins et al., 2017). However, variational optimization inevitably introduces uncertainty into the latent space, which can lead to posterior collapse of meaningful factors, creating a trade-off between image fidelity and controllable attribute manipulation (Figure 2a). Recent works integrate diffusion models with the SCM, capitalizing on its high perceptual quality to explore counterfactual identifiability (Sanchez & Tsaftaris, 2022; Rasal et al., 2025; Komanduri et al., 2024c; Pan & Bareinboim, 2024; Xia et al., 2025). Despite domain-specific tuning, previous approaches perform disentanglement only in auxiliary encoders, which has limited effect on diffusion latents and leads to incomplete disentanglement (Figure 2b and c). This makes models prone to spurious correlations, where target factor interventions often cause unintended changes in non-intervened attributes.

**Text-to-Image based Editing.** An alternative perspective is to view counterfactual image generation as a text-to-image (T2I) based editing, which typically relies on an inversion process (Song et al., 2021). The aim is to generate a target edited image from projected latent states with condition manipulation (Hertz et al., 2023; Ho & Salimans, 2021). To reduce reconstruction error and preserve essential contents, several methods optimize either the unconditional text embedding (Mokady et al., 2023; Xu et al., 2024; Miyake et al., 2025; Ju et al., 2024; Dong et al., 2023), or learn conditional concept embeddings for better attribute disentanglement (Gal et al., 2023; Vinker et al., 2023; Jin et al., 2024). However, generic T2I based editing remains insufficient for counterfactual generation. Existing methods heavily rely on carefully engineered inversion prompts to obtain reliable editing guidance. These approaches lack an explicit, learn-

able SCM over semantic attributes, making them difficult to guarantee both causal faithfulness and identity preserved counterfactual image generation (Figure 2d). A broader discussion of related works can be found in Appendix A.

Herein, we propose **Causal-Adapter**, an adaptive and modular framework that tames text-to-image diffusion model, such as Stable-Diffusion (SD) (Rombach et al., 2022), for counterfactual generation. Unlike prior diffusion-based methods that require considerable re-training or fine-tuning (Sanchez & Tsaftaris, 2022; Komanduri et al., 2024c; Pan & Bareinboim, 2024; Rasal et al., 2025), our method simply injects causal semantic attributes into a frozen backbone via a pluggable adapter (Figure 2e). Inspired by advances in controllable diffusion (Zhang et al., 2023; Zhao et al., 2023; Mou et al., 2024; Li et al., 2025), we investigate integrating semantic attributes with image embeddings as additional conditions. We find that naive fusion fails to achieve sufficient disentanglement or semantic alignment with spatial features in the diffusion latents. To address this, we introduce two regularization strategies that align causal semantic attributes with textual embeddings, improving disentanglement and causal faithfulness (Figure 2f). Our main contributions are summarized as follows:

• Our motivational studies revealed that relying solely on a frozen text-to-image diffusion model with prompt tuning is insufficient for counterfactual image generation, as it fails to jointly represent causal semantic attributes and image embeddings, leading to imprecise reasoning and edits. This underscores the need to inject causal semantics into diffusion models.

• We propose Causal-Adapter, a framework that employs an adapter encoder to learn causal interactions between semantic variables. These interactions are injected into a frozen diffusion and jointly optimized, introducing a

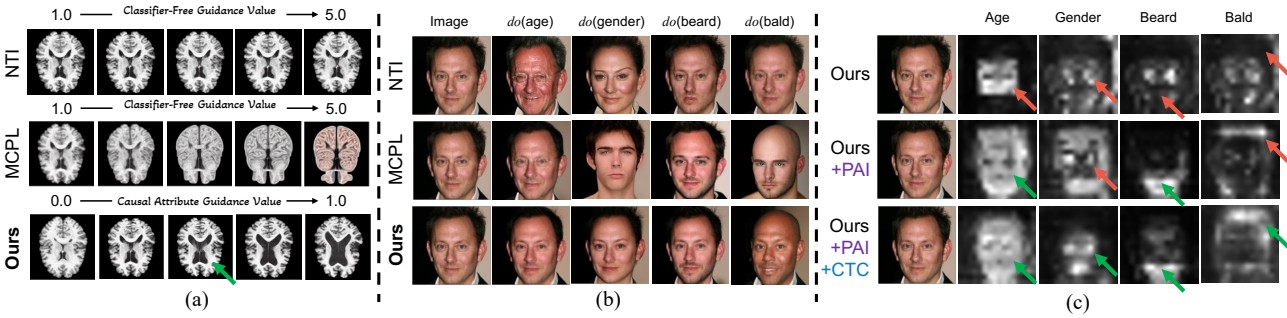

*Figure 3.* **Motivational study and preliminary counterfactual generation results between T2I methods and Causal-Adapter.** (a) Fine-grained anatomical counterfactual editing of brain ventricular volume using inversion-based editing (NTI (Mokady et al., 2023)), multi-concept prompt-learning editing (MCPL (Jin et al., 2024)), and our approach. (b) Comparison of counterfactual editing results on human faces. (c) Cross-attention maps: base vs. regularized Causal-Adapter. Details are presented in Appendix B.

dynamic prior for faithful counterfactual generation.

- To enhance causal disentanglement between semantic variables, we introduce two regularization strategies: Prompt-Aligned Injection (PAI) and Conditioned Token Contrastive Loss (CTC). These strategies separate token embeddings across conditions, enhancing causal representation learning and reducing spurious correlations for more precise counterfactual reasoning.

- We validate Causal-Adapter through extensive experiments on both synthetic and real-world datasets, including human face editing and medical image generation. Our method consistently achieves state-of-the-art performance across key metrics: counterfactual effectiveness (up to 50% MAE reduction on ADNI), realism (81% FID reduction on CelebA), composition (86% LPIPS reduction on CelebA), and minimality (4% CLD reduction on ADNI).

## 2. Methodology

### 2.1. Preliminaries

**Structural Causal Model (SCM).** SCM provides a formal framework for modeling causal relationships between variables (Pearl, 2010). An SCM is defined as a triplet $\mathcal{S} = \langle Y, F, U \rangle$, where $Y = \{y_i\}_{i=1}^{K}$ denotes a set of $K$ endogenous (observed) variables, $U = \{u_i\}_{i=1}^{K}$ is a set of exogenous (latent) variables, and $F = \{f_i\}_{i=1}^{K}$ is a set of deterministic functions defining $y_i = f_i(\text{Pa}_i, u_i)$, with $\text{Pa}_i \subseteq Y \setminus \{y_i\}$ denoting the parent variables of $y_i$. The structural assignments of the SCM induce a directed acyclic graph (DAG) $\mathcal{G}$, where each node represents a variable $y_i$, and each edge $\text{Pa}_i \rightarrow y_i$ represents a direct causal dependency between variables $\text{Pa}_i$ and $y_i$. A SCM is called Markovian if the exogenous variables are mutually independent $p(U) = \prod_{i=1}^{K} p(u_i)$. A Markovian causal model induces the unique joint observational distribution that satisfies the causal Markov condition

$$p_{\mathcal{S}}(Y) = \prod_{i=1}^{K} p(y_i \mid \text{Pa}_i).$$

**Counterfactual Reasoning.** Counterfactual reasoning aims to answer queries "Given observation $Y$, what would $y_i$ have been if its parents $\text{Pa}_i$ had been different?" This is formalized by the *abduction–action–prediction* procedure (Pearl, 2013; Pawlowski et al., 2020; Shen et al., 2022): 1. *Abduction*: inferring posterior over exogenous variables consistent with observation, $p_{\mathcal{S}}(U \mid Y)$. 2. *Action*: perform an intervention $do(y_i = \tilde{y}_i)$, which replaces the structural function $f_i$ with a constant assignment. This yields a modified SCM $\tilde{\mathcal{S}} = \langle \tilde{Y}, \tilde{F}, U \rangle$, where $\tilde{Y}$ and $\tilde{F}$ denote the modified endogenous variables and structural mechanisms respectively. 3. *Prediction*: compute the counterfactual outcome by evaluating $p_{\tilde{\mathcal{S}}}(\tilde{Y})$ under the modified structural mechanism and the inferred exogenous noise.

### 2.2. Motivational Study

To examine the generalization limits of existing text-to-image (T2I) based models for counterfactual image generation, we conducted a motivational study on two real-world datasets: CelebA for human faces and ADNI for brain MRI. Preliminary results are shown in Figure 3, including counterfactual image generation based on different conditional signals in brain volume MRI, as well as interventions on different target attributes in the CelebA dataset. Our study reveals two systematic limitations of directly applying T2I based models to counterfactual generation:

- **Text-only prompting is inadequate for counterfactual generation.** Counterfactual generation should yield consistent visual changes when a single attribute is intervened. However, current T2I based models ignore continuous attributes, making fine-grained edits infeasible, and hence particularly concerning for safety-critical domains such as medical imaging (Figure 3a).

- **Existing T2I based counterfactual generation suffer**

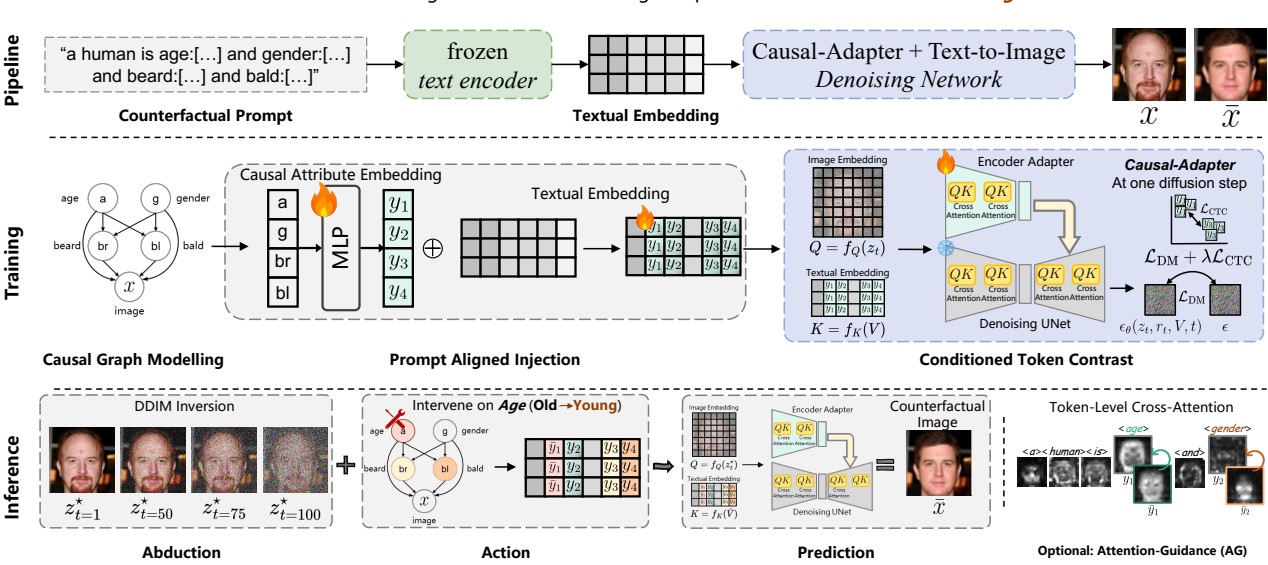

*Figure 4.* Method overview. A counterfactual prompt and input image $x$ are fed into a pretrained text-to-image diffusion model with a learnable Causal-Adapter $\ddot{\epsilon}_\psi$. Causal mechanisms, modeled over a known causal graph and attributes $y_i$, are injected into token embeddings via *Prompt-Aligned Injection (PAI)* to align semantic and spatial features. The adapter $\ddot{\epsilon}_\psi$ operates alongside the frozen diffusion U-Net $\epsilon_\theta$, optimized with MSE $\mathcal{L}_{DM}$ and a *Conditioned Token Contrastive (CTC)* loss $\mathcal{L}_{CTC}$ to enforce disentanglement. At inference, interventions on $y_i$ update token embeddings, and the counterfactual $\bar{x}$ is generated using the abducted exogenous noise $z_t^\star$. Optionally, Attention Guidance (AG) updates the cross-attention map of intervened tokens (*e.g.* age, beard, bald) to achieve localized editing and preserving non-intervened attributes identity (*e.g.* human, gender).

**from attribute entanglement.** Current methods often confuse (entangle) unrelated attributes or require instance-specific fine-tuning, underscoring the need for explicit causal modeling and controllable semantic representations for faithful counterfactual generation (Figure 3b).

### 2.3. Overview of Causal-Adapter

Recent advances on controllable diffusion methods (Zhang et al., 2023; Mou et al., 2024; Li et al., 2025) show that a frozen T2I diffusion backbone can be steered by auxiliary control signals (*e.g.*, segmentation masks or human poses) supplied through a trainable side module. We adopt the same high-level recipe by treating causal semantic attributes as the auxiliary control signals. The causal mechanism between attributes is then learned and injected explicitly into the diffusion backbone via a compact modular encoder that we term as *Causal-Adapter* (Figure 4).

**Causal Mechanism Modeling.** Let $x$ denote an image and $Y = \{y_i\}_{i=1}^K$ denote a vector of semantic variables, where each $y_i$ represents a scalar value corresponding to a high-level semantic attribute. We assume a known causal graph $\mathcal{G}$ encodes the causal relationships among the variables in $Y$. Let $A \in \{0,1\}^{K \times K}$ be the binary adjacency matrix of $\mathcal{G}$, where the $i$-th row $A_i \in \{0,1\}^K$ indicates the parent variables $\mathrm{Pa}_i$ of the $i$-th attribute $y_i$, *i.e.*, $A_{ij} = 1$ if and only if $y_j \in \mathrm{Pa}_i$. We model each causal mechanism $f_i$

using a nonlinear additive noise model such that:

$$\bar{y}_i := f_i(\mathrm{Pa}_i, u_i) = f_i(A_i \odot Y; \omega_i) + u_i, \quad u_i \sim \mathcal{N}(0, \sigma_i^2), \tag{1}$$

where $\omega_i$ are the learnable parameters and $\odot$ denotes element-wise multiplication. $\bar{y}_i$ is the model's prediction of $y_i$ under the change of parent variables $\mathrm{Pa}_i$. For root nodes ($A_i = 0$), we simply set $\bar{y}_i = y_i$ as no causal parents exist. The row vector $A_i$ acts as a binary mask on $Y$, passing only the true parents of $y_i$ to $f_i$. To estimate the parameters of mechanisms $F = \{f_i\}_{i=1}^K$ and the noise variances $\{\sigma_i\}_{i=1}^K$, we minimize the negative log-likelihood of the observations:

$$\begin{aligned} \mathcal{L}_{\mathrm{NLL}} &= -\sum_{i=1}^K \log p\big(y_i \mid f_i(\mathrm{Pa}_i, u_i)\big) \\ &= \frac{1}{2} \sum_{i=1}^K \frac{\|y_i - \bar{y}_i\|_2^2}{\sigma_i^2} + \log(2\pi\sigma_i^2). \end{aligned} \tag{2}$$

Unlike prior methods that infer endogenous and exogenous from latent image features (Yang et al., 2021; Komanduri et al., 2024b), we operate directly in the observed semantic attribute space. This reduces reliance on the quality of learned visual representations and enables exact, interpretable *do*-interventions on attributes. The mechanism modeling can be transferred across domains by simply replacing the attribute set and causal graph $\mathcal{G}$. Moreover, it

is modular that any differentiable alternatives (*e.g.*, Deep-SCM (Pawlowski et al., 2020)) can be seamlessly integrated.

**Integrating Causal Conditions into a Frozen T2I Backbone.** Prior attempts have treated causal attributes as the conditioning signal to the diffusion backbone, which is computationally expensive as it requires retraining the entire model when switching across datasets (Sanchez & Tsaftaris, 2022; Komanduri et al., 2024c; Rasal et al., 2025). To overcome this limitation, we introduce a branch module that delivers causal guidance without modifying the backbone parameters to preserve the flexibility and scalability of a pretrained T2I model. Given the image latent representation $z_t = \mathcal{E}(x, t)$ at diffusion step $t$ and the text embeddings $V$, we instantiate a half-scale replica $\ddot{\epsilon}_\psi$ of the denoiser $\epsilon_\theta$. This replica specializes $\ddot{\epsilon}_\psi$ in injecting control signals and is parameterized by $\psi$. The causal attributes $Y$ are added to $z_t$ and processed by $\ddot{\epsilon}_\psi$:

$$r_t := \ddot{\epsilon}_\psi\big(z_t \oplus Y, t, V\big), \qquad (3)$$

the residuals $r_t$ are injected into the mid- and up-sampling blocks of the frozen denoiser $\epsilon_\theta$. During training, we keep all backbone parameters fixed and optimize only $\psi$ with a T2I noise prediction objective.

**Counterfactual Image Generation.** We implement counterfactual generation under the abduction–action–prediction procedure via DDIM inversion and sampling (Song et al., 2021). Let $H_\theta(z_t, V, t)$ denote the DDIM inversion operator and let $H_\theta^{-1}$ be its generative inverse. The procedure is as follows: (1) *Abduction:* given an observed image $x$, we inject the condition residuals $r_t$ and run $H_\theta(z_{t=0}, r_t, V, t)$ to recover the corresponding exogenous noise $z_T^\star$. (2) *Action:* we intervene $y_i$ by setting $y_i \leftarrow \bar{y}_i$ via $do(y_i = \bar{y}_i)$, propagate the attribute change through Eqn. 1 and Eqn. 3 to obtain the updated $\bar{Y} = \{\bar{y}_i\}_{k=1}^K$ and its residuals $\bar{r}_t$. (3) *Prediction:* the counterfactual outcome is generated with $H_\theta^{-1}(z_{t=T}^\star, \bar{r}_t, V, t)$. Optionally, classifier-free guidance (CFG) (Ho & Salimans, 2021) can be applied to amplify the counterfactual signal with guidance weight $\alpha$:

$$\tilde{\epsilon}_\theta(z_t, \bar{r}_t, t, V, \varnothing) = \alpha\,\epsilon_\theta(z_t, \bar{r}_t, t, V) + (1-\alpha)\,\epsilon_\theta(z_t, r_t^\varnothing, t, \varnothing),$$
$$(4)$$

where $\varnothing = c_\phi(\text{" "})$ is the null-text embedding, and $r_t^\varnothing := \ddot{\epsilon}_\psi(z_t \oplus \bar{Y}, t, \varnothing)$ are the residuals computed under the null-text condition.

## 2.4. Regularizing the Causal-Adapter

A key challenge in counterfactual generation is ensuring that interventions modify only the intended attribute while leaving others unaffected. Prior work highlights that achieving such disentanglement in the latent space is crucial for faithful counterfactuals (Yang et al., 2021; Shen et al., 2022;

Komanduri et al., 2024b; De Sousa Ribeiro et al., 2023; Komanduri et al., 2024c; Rasal et al., 2025). To assess the degree of disentanglement in a multi-modal setting, we use cross-attention maps to reveal attribute separation in latent space. Figure 3c visualizes averaged cross-attention maps across total denoising time steps among all Causal-Adapter variants. Plain Causal-Adapter, which directly injects causal attributes into the image embedding, fails to align attribute semantics with spatial features and disrupts attribute independence, resulting in entangled, off-target edits. To tackle this issue, we introduce two regularization terms that reinforce the conditional causal attribute disentanglement.

**Prompt Aligned Injection.** Causal conditions $Y$ are numeric attributes that lack direct alignment with pixel-level representations but are semantically closer to text embeddings. Inspired by prior finding (Yang et al., 2024) that disentangled tokens in the prompt can guide cross-attention to align semantics with spatial structure in diffusion latents, we inject $Y$ through the prompt channel. This allows the cross-attention module to propagate attribute semantics into spatial image features during generation. We introduce a prompt-aligned injection (PAI) mechanism that maps each causal attribute to a learnable token embedding. Let $C = [\,c_1, \ldots, c_K\,]^\top \in \mathbb{R}^{K \times d}$ denote the embeddings of $K$ placeholder tokens with dimension $d$. Each attribute $y_i$ is mapped into the text space via a linear projector $g_i : \mathbb{R} \to \mathbb{R}^d$. We form the attribute-injected token embeddings:

$$v_i(y_i) = c_i + g_i(y_i), \qquad i = 1, \ldots, K,$$
$$V(Y) = [\,v_1(y_1), \ldots, v_K(y_K)\,]^\top \in \mathbb{R}^{K \times d}.$$
$$(5)$$

The set $V(Y)$ serve as the conditioning input to both the frozen denoiser and the adapter via cross-attention. During training, we jointly optimize the adapter $\psi$, the placeholders token embeddings $C$, and the projectors $G = \{g_i\}_{i=1}^K$ using the standard noise prediction loss $\mathcal{L}_{\text{DM}}$:

$$\min_{\psi,\,G,\,C}\ \mathbb{E}_{z,r,Y,\epsilon,t}\left[\left\|\epsilon - \widehat{\epsilon}_{\theta,\psi}(z_t, r_t, V(\bar{Y}), t)\right\|_2^2\right], \quad (6)$$

where $\widehat{\epsilon}_{\theta,\psi}$ denotes the U-Net prediction modulated by the adapter with PAI. At inference, the learned placeholders $C$ and projectors $G$ are reused to construct $V(\bar{Y})$ for counterfactual query. Notably, PAI only updates token-level embeddings without fine-tuning the CLIP tokenizer or pretrained text encoder, ensuring compatibility between counterfactual queries and existing T2I backbones.

**Conditioned Token Contrast (CTC).** Motivated by prior efforts that contrastive objectives improve concept separation and reduce cross-factor leakage (Jin et al., 2024; Liu et al., 2025), we introduce a token-conditioned contrastive loss that enforces each placeholder token to capture only a single causal factor. Given a batch with $B$ samples and $K$

attributes (tokens), PAI produces a batch of textual embeddings $\{v_b^k\}_{b=1,k=1}^{B,K}$. For a specific anchor $(b,k)$, positive pairs are same attribute token across samples $\{v_{b'}^k \mid b' \neq b\}$ and negative pairs are different attribute tokens across samples $\{v_b^{k'} \mid k' \neq k\}$. By enforcing inter-token invariance (positive pairs) and intra-token separation (negative pairs), CTC reduces attribute entanglement and suppresses spurious correlations. We implement the objective using InfoNCE (Oord et al., 2018; Chen et al., 2020):

$$\mathcal{L}_{\text{CTC}} = \frac{1}{BK} \sum_{k=1}^{K} \sum_{b=1}^{B} \left[ -\frac{\text{sim}\left(v_b^k, v_{b'}^k\right)}{\tau} + \log\left( \sum_{\substack{k'=1 \\ k' \neq k}}^{K} \sum_{\substack{b'=1 \\ b' \neq b}}^{B} \exp\left(\frac{\text{sim}\left(v_b^k, v_{b'}^{k'}\right)}{\tau}\right) \right) \right] \tag{7}$$

where $\text{sim}$ denotes the cosine similarity and $\tau$ is the temperature. Our final training objective becomes:

$$\mathcal{L} = \mathcal{L}_{\text{DM}} + \lambda\,\mathcal{L}_{\text{CTC}}, \tag{8}$$

$\lambda$ denotes a scaling coefficient. The proposed regularizers improve semantic–spatial alignment in the latent space (Figure 3c) and reduce spurious correlations (Figure 9c). The learned attention maps can be leveraged for localized editing through attention-guided manipulation. Following Ju et al. (2024), interventions are applied only to cross-attention weights corresponding to targeted tokens, while attention for non-intervened tokens is preserved. This ensures that edits remain spatially localized (e.g., gender) without altering unrelated attributes (e.g., hairstyle), improving identity preservation without introducing extra training objectives. Full algorithm is presented in Appendix C.3.

# 3. Experiments

We conduct experiments on four counterfactual image generation datasets across different domains: the synthetic **Pendulum** dataset (Yang et al., 2021), the human-face dataset **CelebA** (Liu et al., 2015) and its high-resolution restoration **CelebA-HQ** (Karras et al., 2017), and the medical imaging dataset **Alzheimer's Disease Neuroimaging Initiative (ADNI)** (Petersen et al., 2010). We follow the benchmarking experimental settings (Melistas et al., 2024; Komanduri et al., 2024c; Rasal et al., 2025) for fair comparison. During evaluation, we follow the official setups for each of the datasets and report: (1) Effectiveness: whether the intervention succeeds, measured by pretrained classifiers using F1-score or MAE depending on the attribute type. (2) Composition: reconstruction quality under null interventions, measured by MAE and LPIPS distance. (3) Realism: visual quality of counterfactual images, evaluated with Fréchet Inception Distance (FID). (4) Minimality: non-intervened attributes remain minimally affected, assessed with the Counterfactual Latent Divergence (CLD) metric. Implementation details are presented in Appendix C.

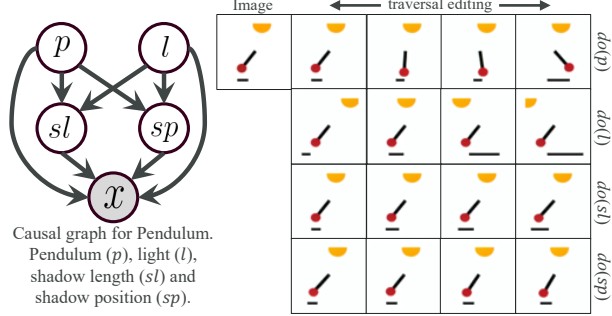

*Figure 5.* Qualitative results on the Pendulum dataset demonstrating counterfactual traversals. When intervening on the pendulum angle or light position, our approach induces causally consistent changes in the descendant shadow attributes (length and position), in accordance with the underlying causal laws.

## 3.1. Results

**Synthetic Imaging Counterfactuals.** We first evaluate our approach on the Pendulum dataset, which consists of four continuous causal variables (pendulum angle, light position, shadow length, and shadow position). We compare against CausalVAE (Yang et al., 2021), as well as diffusion-based methods DisDiffAE (Preechakul et al., 2022) and CausalDiffAE (Komanduri et al., 2024c), using the causal graph shown in Figure 5. As reported in Table 1, our method (row 4) achieves state-of-the-art intervention performance across most attributes. In particular, under $do(l)$ (light intervention), we obtain up to a 91% reduction in MAE for light prediction (from 0.434 to 0.041), indicating highly accurate control over light movement. Moreover, when intervening on light, our approach correctly preserves the pendulum angle while inducing causal changes in the descendant shadow attributes (shadow length and position), consistent with the real-world physical law. We further conduct an investigation into the role of causal mechanisms. Without explicit causal mechanisms (row 5), the model exhibits larger intervention errors, underscoring the necessity of updating semantic variables with causal dependencies during generation. In contrast, injecting synthetic ground-truth labels (row 6) yields performance close to ours (row 4), indicating that our causal mechanism provides a principled approximation to ground-truth causal reasoning, generating faithful counterfactuals over all attributes as shown in Figure 5.

**Human Face Counterfactuals.** Following the benchmarking of Melistas et al. (2024), we evaluate Causal-Adapter on CelebA test set for human face counterfactual generation across four categorical attributes (age, gender, beard, bald) with the causal graph shown in Figure 1. We also incorporate attention guidance to perform localized editing and assess the utility of our learned attention maps. Table 2 reports intervention effectiveness, composition, real-

*Table 1.* Intervention effectiveness on Pendulum test set. We report MAE from pretrained regressors under four interventions. w/ CM: with causal mechanisms; w/o CM: without causal mechanisms; w/ GT: ground-truth labels injected. "∼" denotes descendant attributes remain unaffected. The table follows the causal graph: e.g., under the "Pendulum ($p$) MAE", $do(p)$ or $do(l)$ report the pendulum MAE after intervening on $p$ or $l$.

| Method | Pendulum ($p$) MAE ↓ | | | | Light ($l$) MAE ↓ | | | | Shadow Length ($sl$) MAE ↓ | | | | Shadow Position ($sp$) MAE ↓ | | | |
|---|---|---|---|---|---|---|---|---|---|---|---|---|---|---|---|---|
| | $do(p)$ | $do(l)$ | $do(sl)$ | $do(sp)$ | $do(p)$ | $do(l)$ | $do(sl)$ | $do(sp)$ | $do(p)$ | $do(l)$ | $do(sl)$ | $do(sp)$ | $do(p)$ | $do(l)$ | $do(sl)$ | $do(sp)$ |
| 1. CausalVAE | 24.86 | 23.03 | 20.47 | 11.58 | 34.20 | 26.01 | 35.49 | 47.06 | 1.946 | 1.430 | 2.020 | 1.720 | 52.52 | 72.50 | 57.03 | 32.78 |
| 2. DisDiffAE | 0.668 | 0.648 | 0.647 | 0.647 | 0.656 | 0.654 | 0.630 | 0.651 | 0.550 | 0.527 | 0.560 | 0.516 | 0.474 | 0.475 | 0.479 | 0.534 |
| 3. CausalDiffAE | 0.297 | 0.132 | **0.031** | **0.034** | 0.045 | 0.434 | **0.035** | **0.064** | 0.136 | 0.322 | 0.492 | **0.082** | 0.146 | 0.303 | 0.064 | 0.471 |
| 4. Ours$_{w/ CM}$ | **0.014** | **0.035** | 0.043 | 0.259 | **0.045** | **0.041** | 0.058 | 0.120 | **0.028** | **0.051** | 0.489 | 0.110 | **0.030** | **0.033** | 0.041 | **0.336** |
| 5. Ours$_{w/o CM}$ | 0.159 | 0.183 | ∼ | ∼ | 0.060 | 0.173 | ∼ | ∼ | 0.143 | 0.235 | ∼ | ∼ | 0.086 | 0.155 | ∼ | ∼ |
| 6. Ours$_{w/ GT}$ | 0.013 | 0.033 | ∼ | ∼ | 0.043 | 0.039 | ∼ | ∼ | 0.025 | 0.036 | ∼ | ∼ | 0.028 | 0.035 | ∼ | ∼ |

*Table 2.* CelebA results. We report target attribute effectiveness and include composition, realism, and minimality metrics.

| Method | (a) F1↑ | (g) F1↑ | (br) F1↑ | (bl) F1↑ | LPIPS↓ | FID↓ | CLD↓ |
|---|---|---|---|---|---|---|---|
| | $do(a)$ | $do(g)$ | $do(br)$ | $do(bl)$ | (Comp.) | (Real.) | (Min.) |
| 1. VAE | 35.0 | 90.9 | 29.6 | 41.2 | 0.282 | 59.397 | **0.299** |
| 2. HVAE | **65.4** | 94.9 | 44.1 | **61.1** | 0.122 | 35.712 | 0.305 |
| 3. GAN | 41.3 | 98.2 | 23.3 | 49.2 | 0.276 | 27.861 | 0.304 |
| 4. Ours | 58.5 | **99.9** | **52.1** | 58.8 | **0.017** | 8.152 | 0.310 |
| *with attention guidance for localized editing* | | | | | | | |
| 5. Ours (AG) | 57.1 | 99.7 | 48.2 | 51.4 | – | **5.213** | 0.301 |

*Table 3.* Intervention target effectiveness on ADNI.

| Method | (b) MAE↓ | (v) MAE↓ | (s) F1↑ | LPIPS↓ | FID↓ | CLD↓ |
|---|---|---|---|---|---|---|
| | $do(b)$ | $do(v)$ | $do(s)$ | (Comp.) | (Real.) | (Min.) |
| 1. VAE | 0.17 | 0.20 | 0.46 | 0.306 | 278.245 | 0.352 |
| 2. HVAE | 0.09 | 0.04 | 0.41 | 0.101 | 74.696 | 0.347 |
| 3. GAN | 0.17 | 0.22 | 0.05 | 0.268 | 113.749 | 0.353 |
| 4. Ours | **0.09** | **0.03** | **0.48** | **0.035** | 9.130 | 0.346 |
| *with attention guidance for localized editing* | | | | | | |
| 5. Ours (AG) | 0.10 | 0.04 | 0.46 | – | **9.066** | **0.332** |

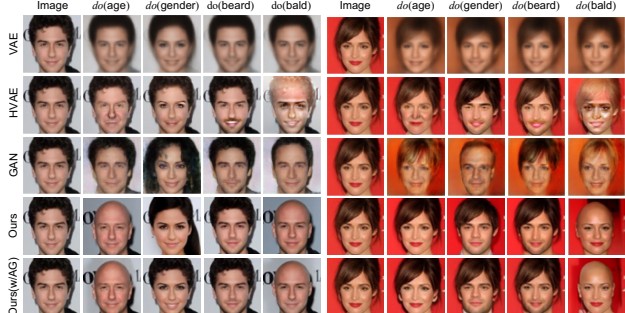

*Figure 6.* Qualitative comparison on the CelebA dataset. The results demonstrate that our method preserves core identity features while effectively enforcing causal interventions.

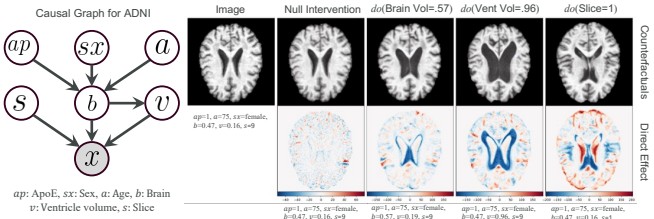

*Figure 7.* ADNI brain MRI counterfactual results from Causal-Adapter. Direct causal effects are shown (red: increase; blue: decrease). The results show sharp, localized interventional changes consistent with the causal graph (left), while preserving the observation's identity.

ism, and minimality. Our method achieves best performance across most interventions, including up to an $86\%$ reduction in LPIPS (composition, from 0.122 to 0.017) and an $79\%$ reduction in FID (realism, from 27.861 to 8.152). HVAE achieves competitive effectiveness via post-training classifier optimization but produces classifier-biased artifacts and reduced fidelity (Figure 6). Further localized editing with attention guidance balances intervention effectiveness and identity preservation, achieving the best FID (from 8.152 to 5.213). Qualitative results confirm that the learned attention maps enable precise, localized detailed editing (e.g., modifying gender without altering hairstyle), enable Causal-Adapter to preserve core identity while enforcing causal interventions. Full results are in Appendix E.2.

**Brain Imaging Counterfactuals.** We further evaluate our method on ADNI dataset, which includes six attributes in both categorical variables (ApoE, Sex, Slice) and continuous variables (Age, Brain Volume, Ventricular Volume). Following Melistas et al. (2024), we intervene on three generative conditioning attributes (Brain Volume, Ventricular Volume, Slice) and report the results in Table 3. Our approach achieves best performance in intervention effectiveness (up to $50\%$ MAE reduction in fine-grained edits) and minimality, while also delivering strong realism ($87\%$ FID reduction) even without attention guidance. Qualitative results (Figure 7) further show that our model produces sharp and localized interventional changes consistent with the causal graph. In particular, fine-grained edits to Ventricular Volume visibly enlarge the ventricle region while preserving subject identity. Full results are in Appendix E.3.

*Table 4.* CelebA-HQ results. Intervention effectiveness, reversibility, and identity preservation for eyeglasses and smiling.

| Method | (g) F1↑ $do(g)$ | (g) $L_1$↓ (Rev.) | (g) LPIPS↓ (IDP) | (s) F1↑ $do(s)$ | (s) $L_1$↓ (Rev.) | (s) LPIPS↓ (IDP) |
|---|---|---|---|---|---|---|
| 1. VCI | 3.39 | – | – | 33.81 | – | – |
| 2. HVAE | 65.31 | – | – | 75.33 | – | – |
| 3. DiffCounter | 96.86 | 0.185 | 0.096 | **94.93** | 0.183 | 0.066 |
| 4. Ours | **99.26** | **0.049** | 0.084 | 94.15 | **0.028** | **0.028** |
| 5. Ours (DiT) | 97.39 | 0.086 | **0.060** | 94.71 | 0.089 | 0.035 |

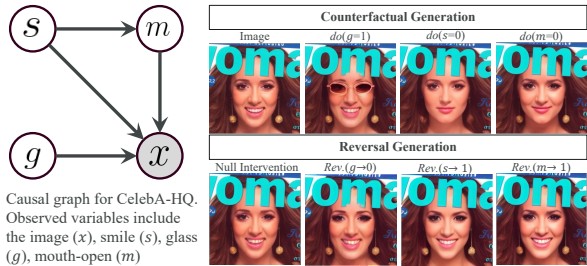

*Figure 8.* CelebA-HQ counterfactuals, together with reversible cases that map back to the original observations, illustrating identity preservation and reversibility.

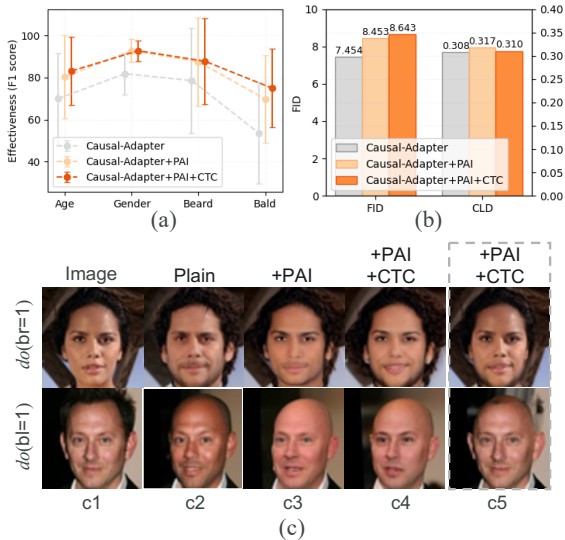

*Figure 9.* Ablation study on CelebA validation set. (a) Average intervention effectiveness. (b) Realism and minimality. (c) Qualitative examples, with dotted boxes indicating results of localized editing. Both quantitative and qualitative results demonstrate the necessity of the proposed PAI and CTC.

**High-Resolution Face Counterfactuals.** We further compare Causal-Adapter with recent SOTA methods, VCI (Wu et al., 2025), HVAE (De Sousa Ribeiro et al., 2023), and DiffCounter (Rasal et al., 2025) on high-resolution human face dataset CelebA-HQ. We follow the same settings in DiffCounter for fair comparison, focusing on three categorical variables (eyeglass, smiling, mouth-open). Quantitative results are reported in Table 4, with qualitative examples shown in Figure 8. Our method (row 4) achieves intervention effectiveness comparable to DiffCounter, while substantially improving both reversibility and identity preservation. We further evaluate generalization by instantiating Causal-Adapter with a diffusion transformer (DiT) backbone (Esser et al., 2024). As shown in Table 4 (row 5) and Appendix E.4, Causal-Adapter transfers effectively across distinct T2I backbones and produces causally faithful, high-resolution counterfactuals.

**Ablation Study.** We conduct an ablation study on the CelebA validation set to evaluate the contribution of each regularizer. As shown in Figures 9(a) and (b), the plain adapter achieves the lowest intervention effectiveness. Adding the PAI module yields a consistent average gain of +11.6% F1 across all attributes with slight increases in FID and CLD, indicating that aligning causal semantics with spatial features in the diffusion latents enables more effective edits. Incorporating CTC further improves intervention effectiveness by enforcing token embedding disentanglement, while also reducing CLD (from 0.317 to 0.310) and keeping FID stable (from 8.453 to 8.643), resulting more faithful counterfactual generation. Figure 9c presents quali-

tative ablation results. Spurious correlations arise in plain adapter, *e.g.*, beard interventions in females induce male facial features (c2). PAI alleviates attribute entanglement (c3), while the complete regularization with CTC produces bearded female, demonstrating mitigation of such correlations (c4). Under bald interventions, the baselines alter skin color or age (c2–c3), whereas the full regularization edits baldness with only minor facial changes (c4). Finally, attention guidance enhances identity preservation with localizing edits (c5). Extended results are presented in Appendix D.

**Beyond Pre-defined Graphs.** Existing counterfactual generation methods often assume a pre-defined causal graph (Pearl, 2009; Pawlowski et al., 2020; Yang et al., 2021), which can be restrictive when the graph is misspecified, partially known, or unavailable. To relax this assumption, we make the adjacency matrix $A$ learnable and regularize it with an acyclicity constraint following differentiable DAG methods such as NOTEARS (Zheng et al., 2018) and DAGMA (Bello et al., 2022). This extension requires no architectural changes and only adds a structural loss term.

As shown in Figure 10(a), this extension enables Causal-Adapter to perform causal structure learning and achieve superior performance compared with SOTA differentiable causal discovery methods, including SDCD (Nazaret et al., 2024). Moreover, our method recovers more true causal edges than competing baselines. The learning dynamics of $A$ during training are visualized in Figure 10(b), showing stable convergence toward the ground-truth graph and demon-

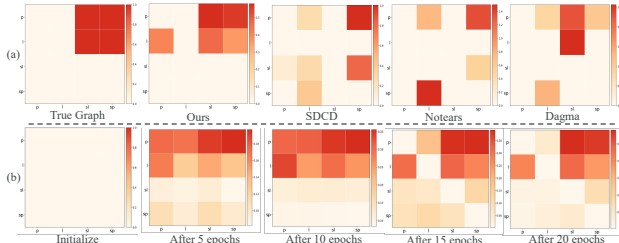

*Figure 10.* Causal discovery performance on the Pendulum dataset. (a) Comparison of Causal-Adapter with SDCD, NOTEARS, and DAGMA. (b) Learning trajectory of the adjacency matrix $A$.

*Table 5.* Ablation study on different causal graphs for SCM modeling. w/o SCM denotes the setting without using a causal graph, while w/ SCM1 and w/ SCM2 correspond to the causal graphs shown in Figure 11.

| Method | $(p)$ **MAE** $\downarrow$ | | $(l)$ **MAE** $\downarrow$ | | $(sl)$ **MAE** $\downarrow$ | | $(sp)$ **MAE** $\downarrow$ | |
|---|---|---|---|---|---|---|---|---|
| | $do(p)$ | $do(l)$ | $do(p)$ | $do(l)$ | $do(p)$ | $do(l)$ | $do(p)$ | $do(l)$ |
| 1. w/o SCM | 0.159 | 0.183 | 0.060 | 0.173 | 0.143 | 0.235 | 0.086 | 0.155 |
| 2. w/ SCM1 | **0.011** | 0.040 | 0.048 | 0.062 | 0.039 | 0.104 | 0.031 | 0.033 |
| 3. w/ SCM2 | 0.014 | **0.035** | **0.045** | **0.041** | **0.028** | **0.051** | **0.030** | **0.033** |

strating the interpretable behavior of our model. Overall, these results suggest that Causal-Adapter has the potential to unify causal discovery and counterfactual generation within a single, simple, and efficient framework. Extended results are presented in Appendix G.

**Ablation on Causal Graphs.** To validate the robustness of Causal-Adapter to causal graph specification, we conduct an ablation study under different graph settings. As shown in Table 5 and the results in Figure 11, we compare three settings: (1) without SCM, (2) with an incorrect causal graph, denoted as SCM1, and (3) with the correct causal graph, denoted as SCM2. Without SCM, interventions can still affect shadow-related variables because pendulum, light, and shadow attributes are strongly correlated in the training data. However, these changes are primarily correlation-driven rather than graph-faithful: intervening on one factor may spuriously modify another non-intervened factor, such as changing the light while also moving the pendulum. With SCM1, the incorrectly specified causal edge leads to an inconsistent propagation pattern and results in a different shadow evolution behavior. In contrast, with the correct graph SCM2, the intervention effects propagate more consistently with the underlying physical mechanisms.

Importantly, our design separates SCM modeling from conditional disentanglement. This allows the framework to adapt to different causal graphs at inference time by replacing the SCM, without retraining the diffusion backbone. These results show that, even without a causal graph, the model can still function as a disentangled editing method; however, with the correct causal graph, the generated coun-

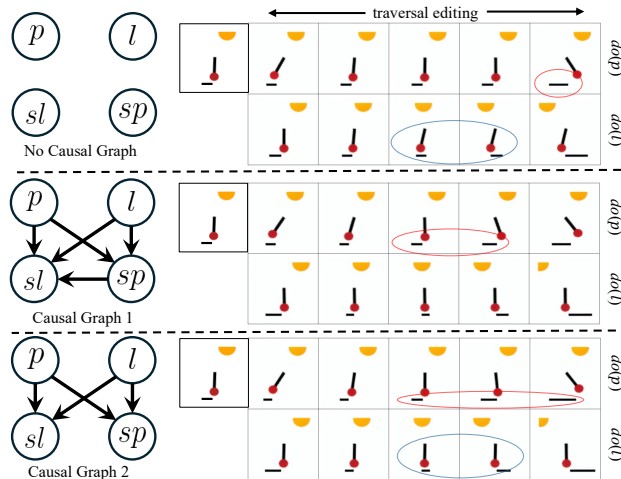

*Figure 11.* Pendulum counterfactuals under three graph settings: no causal graph, an incorrect causal graph (SCM1), and the correct causal graph (SCM2).

terfactuals better match the underlying data-generating structure and achieve stronger quantitative performance. Extended results are presented in Appendix G.

**Generalization to Broader Settings.** To further assess the practical effectiveness of Causal-Adapter, we compare it with recent commercial foundation editing models, including Nano Banana 2, Qwen-Image-Edit, and Flux.1-Kontext-Dev, on our three benchmark datasets and a less tightly aligned full-body human image dataset. Results in Appendix E.6 and Appendix E.7 support its practical applicability across broader image domains.

## 4. Conclusion

We introduced Causal-Adapter, a modular framework to tame Text-to-Image diffusion models for counterfactual generation. Our motivational study revealed that current Text-to-Image diffusion based editing approaches lack an explicit structural causal model for attribute control, making it difficult to generate faithful counterfactual images. In contrast, Causal-Adapter is a simple yet effective framework that leverages a frozen diffusion backbone and injects causal semantic attributes through a pluggable adapter network to explicitly learn causal semantics. We further proposed prompt aligned injection and conditioned token contrastive optimization, which align attribute semantics with spatial features and promote disentanglement in the latent space, reducing spurious correlations while preserving identity for generations. Causal-adapter achieves superior counterfactual generation performance on multiple datasets. Extensive evaluation across diverse settings further confirm that Causal-Adapter provides a robust, scalable, and practical alternative for enabling causal editing in modern T2I systems.

## Impact Statement

This paper presents work whose goal is to advance the field of Machine Learning. There are many potential societal consequences of our work, none which we feel must be specifically highlighted here.

## Acknowledgment

We thank the area chair and reviewers for their constructive feedbacks. Z.Liu and S.Tsaftaris were supported by the UKRI AI programme and the Engineering and Physical Sciences Research Council, for CHAI - Causality in Healthcare AI Hub [grant number EP/Y028856/1].

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

# Appendix Table of Contents

# A. Extended Related Works [Back to Contents]

**Counterfactual Image Generation** aims to synthesize images that reflect the visual effect of a hypothetical intervention while preserving non-intervened attributes according to an underlying causal graph and maintaining instance-specific identity details (Komanduri et al., 2024a; Melistas et al., 2024). Existing approaches often augment generative models with explicit structural causal models (SCMs) and follow Pearl's *abduction–action–prediction* paradigm (Pearl, 2009; 2013; Pawlowski et al., 2020; Shen et al., 2022; Sanchez & Tsaftaris, 2022; De Sousa Ribeiro et al., 2023; Wu et al., 2025; Spyrou et al., 2025). Early work predominantly relied on **VAEs or GANs** (Kingma & Welling, 2013; Goodfellow et al., 2014; Kocaoglu et al., 2018; Pawlowski et al., 2020), as their noise-injection and variational sampling mechanisms offered a natural way to represent exogenous uncertainty, while their objectives encouraged disentanglement of causal factors, inspired by $\beta$-VAE (Higgins et al., 2017). De Sousa Ribeiro et al. (2023) extends this line by incorporating hierarchical VAEs (Vahdat & Kautz, 2020) to estimate direct, indirect, and total causal effects, whereas Wu et al. (2025) combined variational Bayesian inference with adversarial training to improve abduction and preserve identity. Other works (Yang et al., 2021; Shen et al., 2022; Komanduri et al., 2024b; Li et al., 2024) integrates SCM priors directly into the latent space of VAEs, enabling implicit causal reasoning on image encodings. However, variational optimization inevitably introduces uncertainty into the learned representations, which can lead to posterior collapse or the loss of semantically meaningful factors. This results in an inherent trade-off between high-fidelity image synthesis and flexible attribute control (Higgins et al., 2017; Alemi et al., 2018). Sanchez & Tsaftaris (2022) introduces **DiffSCM**, the first framework to integrate diffusion with SCMs for counterfactual generation. DiffSCM employed DDIM inversion (Song et al., 2021) for abduction and conditioned the generative process on causal labels, but it was limited to small parent sets and simple causal structures. Pan & Bareinboim (2024) extends DiffSCM by combining a VAE for preliminary counterfactual generation with diffusion-based refinement, shifting the conditioning from attribute labels to pre-generated images to produce refined counterfactuals. Chao et al. (2024) models structural equations directly using diffusion processes, enabling counterfactual sampling on a predefined causal graph. While DCM also answers interventional queries, it trains a dedicated diffusion model per causal node. Recent works (Komanduri et al., 2024c; Rasal et al., 2025; Xia et al., 2025) build on **Diffusion Autoencoder** framework (Preechakul et al., 2022), equipping diffusion models with variational encoders to inject semantic attributes into diffusion latents. However, these methods require heavy post-training or fine-tuning and remain prone to spurious correlations. A key limitation is that disentanglement is typically enforced through auxiliary encoders with limited influence on the intrinsic latent space of diffusion models, failing to align semantic attributes with disentangled spatial representations of images and leaving causal disentanglement incomplete (Wu et al., 2025; Yang et al., 2024).

To address these limitations, we propose **Causal-Adapter**, an adaptive and modular framework that employs an adapter encoder to explicitly learn causal interactions between semantic attributes. We further introduce two regularization strategies: Prompt Aligned Injection (PAI) and Conditioned Token Contrastive Loss (CTC). These strategies align semantic attributes with spatial features in the diffusion latents and separate token embeddings across conditions, thereby enhancing causal representation learning and reducing spurious correlations, all while preserving the pre-trained diffusion weights.

**Text-to-Image based Editing** aims to manipulate existing images according to user-provided natural language instructions (Brooks et al., 2023). Most approaches rely on an inversion process (Song et al., 2021), where the image is projected into a latent state and then synthesized under modified conditions (Hertz et al., 2023; Ho & Salimans, 2021). However, DDIM inversion and classifier-free guidance often often interfere with each other, leading to a trade-off between preserving essential content and achieving faithful edits (Ju et al., 2024; Huberman-Spiegelglas et al., 2024; Kynkäänniemi et al., 2024; Cao et al., 2025). To mitigate this, Null-text inversion (Mokady et al., 2023) learns a null embedding to account for reconstruction discrepancies, while Dong et al. (2023) optimizes conditional embeddings to reduce information loss in unconditional guidance. Yet, both methods require costly per-sample optimization. Subsequent works (Ju et al., 2024; Xu et al., 2024; Miyake et al., 2025) improve efficiency by recording residual losses between conditional and unconditional embeddings and re-injecting them during editing, to stabilize edits and preserving content. Textual inversion (Gal et al., 2023) improves content preservation by disentangling single concepts: it learns new text embeddings from a few personalized images to represent objects in novel contexts. Extending this, Vinker et al. (2023) decomposes concept embeddings into sub-concepts via learned vectors injected into the latent space of T2I model, while Jin et al. (2024) performs multi-concept disentanglement using adjective-based prompts and contrastive optimization. Lyu et al. (2024) propose a one-dimensional non-invasive adapter that modulates concept semi-permeability in frozen diffusion models for concept erosion, and Yeganeh et al. (2025) address domain shift in medical imaging, supporting counterfactual-like edits such as aging or disease progression without explicit structural constraints. Despite these advances, generic T2I editing remains insufficient for causal counterfactual generation. Existing methods depend heavily on prompt manipulation and do not incorporate a learnable structural causal

model (SCM). They make no use of observed semantic attributes or a predefined causal graph, and therefore cannot enforce that edits follow correct causal dependencies. As a result, current T2I-based editing techniques struggle to simultaneously maintain causal faithfulness and identity preservation in counterfactual image generation.

As shown in our motivational study (Appendix B), we evaluate the adaptation of text-to-image diffusion models for counterfactual image generation. Our findings show that relying solely on a frozen diffusion backbone with prompt-tuning is insufficient, as it fails to jointly represent causal semantic attributes and image embeddings. As a result, the model struggles to achieve precise counterfactual reasoning and generation. This highlights the need for an adaptive mechanism within the diffusion model, enhanced with injected causal semantics. Following the standard formulation of counterfactual image generation (Pearl, 2009; De Sousa Ribeiro et al., 2023; Wu et al., 2025; Rasal et al., 2025), where a predefined causal graph is treated as the structural prior, our Causal-Adapter models the SCM directly on the observed semantic attributes. This enforces correct causal dependencies during intervention and enables the generation of counterfactual images that are both visually plausible and causally faithful. When the SCM module is omitted, the framework naturally reduces to a standard conditional generation setting for that attribute, as illustrated in Figure 1.

**Controllable Diffusion Models**   extend T2I diffusion frameworks by incorporating additional user-specified signals to guide generation (Huang et al., 2025). One approach is to train diffusion models from scratch with multi-conditional objectives (Huang et al., 2023; Puglisi et al., 2024), which achieves strong controllability and high-quality synthesis but at huge computational cost. More recently, adapter-based methods (Zhang et al., 2023; Li et al., 2023; Zhao et al., 2023; Mou et al., 2024; Li et al., 2025) have emerged as a scalable alternative. By attaching lightweight, trainable modules to a frozen Stable Diffusion backbone, these methods enable the model to incorporate auxiliary control signals such as segmentation masks or pose skeletons, significantly reducing training overhead while maintaining stability and fidelity. We adopt the same high-level recipe by treating causal semantic attributes as the auxiliary control signals, and employ an adapter encoder to explicitly learn causal interactions between high-level variables. These interactions are then injected into a a frozen diffusion backbone, and jointly optimized with partial textual embeddings. The design of Causal-Adapter introduces a dynamic and learnable prior, enabling the effective adaptation of a frozen diffusion network for realistic and faithful counterfactual image generation.

## B. Full Motivational Study Results [Back to Contents]

To assess the feasibility of using a pretrained text-to-image (T2I) model for counterfactual generation, we conduct a motivation study to answer the following three questions.

**1. Can existing T2I based editing methods perform counterfactual generation?** We begin by examining a representative text-to-image (T2I) editing method, Null-Textual Inversion (NTI) (Mokady et al., 2023). Our findings highlight two fundamental issues: (1) *Heavy reliance on prompt engineering and instability.* As shown in Figure 12a, the success of NTI-based edits is highly sensitive to prompt wording. For example, when editing age, the token "old" may fail while the synonym "aged" succeeds under one inversion prompt, yet the opposite occurs under another prompt. This indicates that editing success and visual quality depend not only on the choice of attribute word but also on the particular inversion prompt. Even minor wording changes that preserve semantic meaning can alter tokenization and cross-attention patterns, leading to inconsistent success rates and perceptual artifacts. Moreover, to preserve identity while achieving the desired edit, practitioners must manually try multiple prompt variants, requiring extensive hand-crafted effort. (2) *Weak counterfactual faithfulness.* Beyond instability, prompt-based editing fails to reliably reflect the intended intervention. In Figure 12b, using the word "man" to edit gender (woman case) or "old" to edit age (man case) yields different counterfactual characteristics depending on the inversion prompt: one "old" edit introduces glasses while another does not. Such variability arises purely from prompt formulation, introducing extraneous variance unrelated to the target attribute. This illustrates weak counterfactual faithfulness: the same intervention should yield coherent edits of the intended factor, without unintended changes in other attributes. Hence, purely text-driven control is inadequate for reliable counterfactual generation.

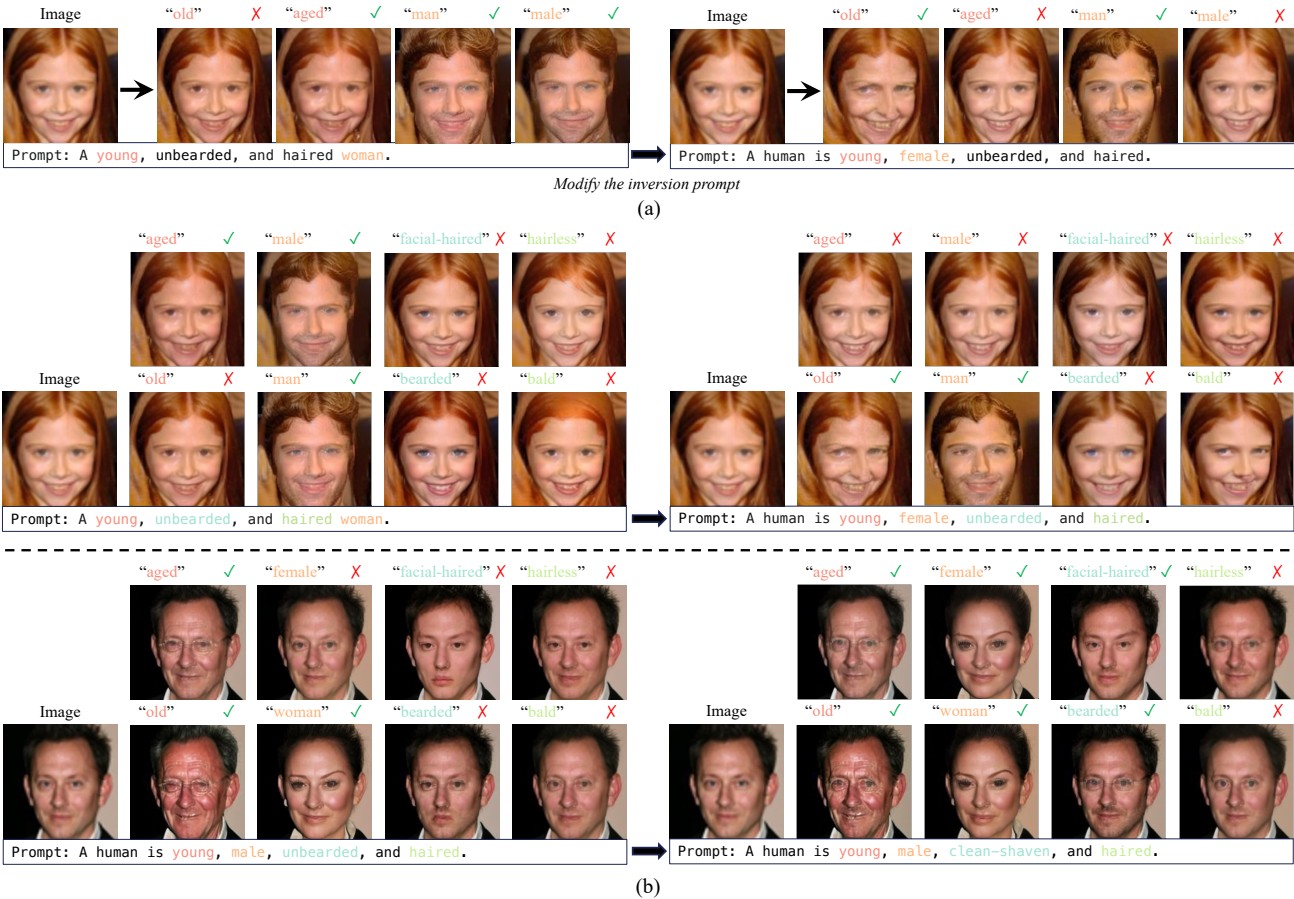

*Figure 12.* Null-Textual Inversion (NTI) relies heavily on prompt engineering, where minor word changes can determine editing success. (a) Illustration of our study: two inversion prompts are given for the same image, and attribute words are replaced with different synonyms. Successful edits are marked with ✓ and failures with ✗ (human evaluation). Results show that editing outcomes are highly sensitive to prompt wording and reveal weak counterfactual faithfulness. (b) Full results of this investigation.

**2. Can multi-concept prompt learning yield disentangled attribute control?** Prompt-learning approaches such as Textual Inversion (Gal et al., 2023), Inspiration Tree (Vinker et al., 2023) and Multi-Concept Prompt Learning (MCPL) framework (Jin et al., 2024) to learn conditional concept embeddings for attribute disentanglement. We adopt MCPL framework as our baseline. As shown in Figure 13, MCPL can perform certain edits that NTI fails to capture (e.g., editing baldness for men or adding beards for women) by equipping text embeddings learned from multiple samples. However, we observe two key limitations: (1) *Entangled edits and lack of faithfulness.* Edits often induce changes in unrelated attributes, heavily altering the background or unintended regions. This leads to a loss of fidelity and identity preservation, thus violating the faithfulness requirement of counterfactual generation. (2) *Per-sample optimization is still required.* Similar to NTI, prompt-learning approaches need separate fine-tuning for each image instance. This makes them impractical for scalable, real-world causal reasoning tasks. Our findings show that relying solely on a frozen diffusion backbone with prompt-tuning is insufficient, as it fails to jointly represent causal semantic attributes and image embeddings. As a result, the model struggles to achieve precise counterfactual reasoning and generation. This highlights the need for an adaptive mechanism within the diffusion model, enhanced with causal semantics.

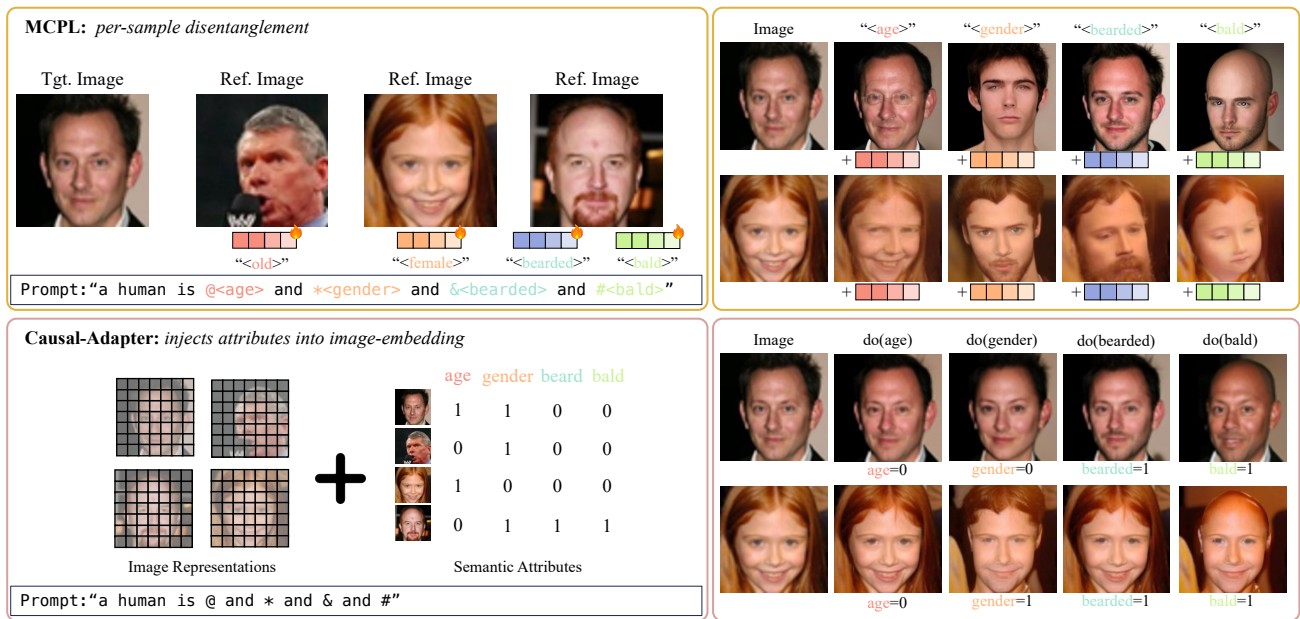

*Figure 13.* Multi-Concept Prompt Learning (MCPL) as a representative prompt-learning baseline. Each placeholder token (*e.g.*, @ for "young", * for "male") is initialized with a pretrained embedding and jointly optimized across multiple concepts. At test time, counterfactuals are generated by swapping embeddings with target embeddings (*e.g.*, "old", "female"). MCPL can achieve some edits missed by NTI (e.g., baldness, beard), but often entangles unrelated attributes, alters backgrounds, and requires per-instance optimization. In contrast, our vanilla Causal-Adapter injects causal attributes into image embeddings, supporting batch optimization and counterfactual generation via direct attribute interventions.

**3. Can existing T2I based methods do fine-grained editing?** In causal editing, interventions on a single factor (*e.g.*, increasing object size) should consistently induce predictable and proportional changes in the generated image. However, with current T2I based methods, numerical attributes must be mapped to discrete linguistic tokens, making it difficult to express edits along a continuous range. In practice, the primary way to adjust editing strength is by tuning the classifier-free guidance scale (Ho & Salimans, 2021). As shown in Figure 14, both NTI and MCPL fail to achieve fine-grained anatomical counterfactual editing of brain ventricular volume, even with different guidance scales. Their text-only control mechanisms cannot reliably translate numeric interventions into gradual visual changes. This limitation is particularly problematic in medical imaging domains, where precise, numerically controlled edits (*e.g.*, adjusting brain or ventricular volume in MRI scans) are essential for simulating disease progression.

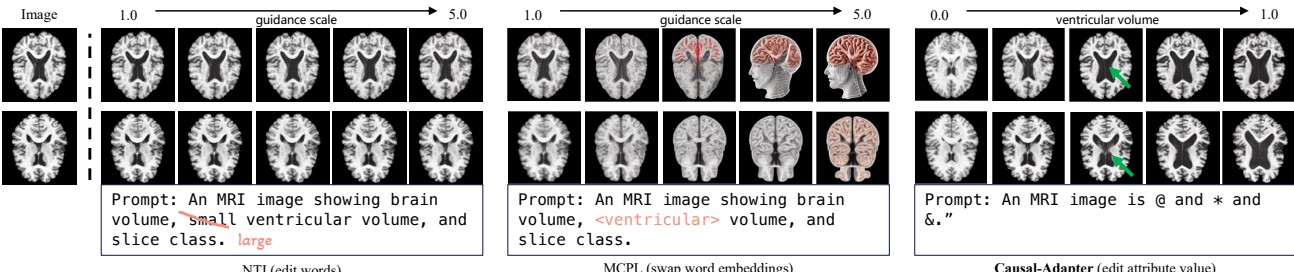

*Figure 14.* Fine-grained anatomical counterfactual editing of brain ventricular volume. NTI and MCPL cannot achieve fine-grained control with text-only prompts. In contrast, our Causal-Adapter produces sharp, localized interventions that smoothly adjust ventricular volume from small to large while preserving subject identity.

## C. Dataset and Implementation [Back to Contents]

### C.1. Implementation Details

We build our method on the backbone of Stable Diffusion v1.5 (Rombach et al., 2022), using the same hyperparameter settings as the original implementation. The pre-trained weights are trained at a resolution of $256 \times 256$. Our adapter network $\ddot{\epsilon}_\psi$ is designed as a half-copy of the diffusion U-Net backbone, consisting of the encoder and bottleneck layers.

Unless specified, experimental setups on CelebA and ADNI follow the benchmark protocol in (Melistas et al., 2024), and CelebA-HQ experiments follow the setup in (Rasal et al., 2025). To ensure fairness, we adopt the same causal mechanism (normalizing flow) as described in the benchmark implementation. Experimental setup on Pendulum follows the configuration in CausalDiffAE (Komanduri et al., 2024c). Since both approaches assume linear causal modeling, we use our constructed simple MLP-based causal mechanism.

We construct prompts in the form of "a human is @ and ..." or "an MRI image is @ and ...", where the placeholder tokens like "@" is aligned with semantic attributes in the embedding space via PAI. Prefix words (*e.g.* "a human is" and "an image is") are kept fixed across datasets, as we observed no significant performance differences when switching them.

During training, we apply the proposed CTC loss with a temperature $\tau = 0.2$ and scaling factor $\lambda = 0.0005$ for all datasets. At inference, all images are first generated at $256 \times 256$ resolution and then down-sampled via bicubic interpolation to match the resolution of the original benchmarks.

For counterfactual image generation, we employ DDIM inversion (Song et al., 2021). To best preserve identity, we set the inversion guidance scale to $1.0$ (no classifier-free guidance (CFG)). For the forward process, we use CFG: CelebA uses $\alpha = 3.0$ and CelebA-HQ uses $\alpha = 1.5$ for effective edits, while ADNI and Pendulum perform well without CFG. Optionally, we incorporate token-level attention guidance manipulation following (Ju et al., 2024), replacing only the intervened token attentions while keeping the others fixed.

All experiments are executed on a single NVIDIA A100 GPU. Training completes within one day, and generating a single counterfactual image takes approximately 5–7 seconds. Table 6 presents detailed training configurations.

*Table 6.* Experimental settings for our Causal-Adapter across three datasets.

| | CELEBA | ADNI | PENDULUM | CELEBA-HQ |
|---|---|---|---|---|
| TRAIN SET SIZE | 162,770 | 10,780 | 5,000 | 24000 |
| VALIDATION SET SIZE | 19,867 | 0 | 500 | 3000 |
| TEST SET SIZE | 19,962 | 2,240 | 2,000 | 3000 |
| RESOLUTION | $256 \times 256 \times 3$ | $256 \times 256 \times 3$ | $256 \times 256 \times 3$ | $256 \times 256 \times 3$ |
| DOWNSAMPLED RESOLUTION | $64 \times 64 \times 3$ | $192 \times 192 \times 1$ | $94 \times 94 \times 3$ | $64 \times 64 \times 3$ |
| BATCH SIZE | 40 | 40 | 40 | 40 |
| TRAINING STEPS | 200k | 100k | 50k | 100k |
| NUM OF ATTRIBUTES | 4 | 6 | 4 | 7 |
| PROMPT TEMPLATE | ``a human is @ and * and & and #'' | ``an MRI image is @ and * and &'' | ``an image is @ and * and & and #'' | ``a human is @ and * and & and # and ! and ? and %'' |
| CTC TEMPERATURE $\tau$ | 0.2 | 0.2 | 0.2 | 0.2 |
| CTC SCALE $\lambda$ | 0.0005 | 0.0005 | 0.0005 | 0.0005 |
| DDIM STEPS $T$ | 100 | 100 | 100 | 100 |
| GUIDANCE SCALE $\alpha$ | 3.0 | 1.0 | 1.0 | 1.5 |
| LEARNING RATE | | 1e-5 | | |
| OPTIMIZER | | AdamW (weight decay 1e-2) | | |
| LOSS | | MSE (noise prediction) + CTC (proposed) | | |

### C.2. Metrics

We detail the counterfactual evaluation metrics used in our experiments below.

**Effectiveness.** Effectiveness measures how successfully an intervention alters the intended attributes in counterfactual images. To quantitatively evaluate effectiveness, we leverage an anti-causal predictor trained on the observed data distribution for each parent variable of the image $x$ defined in the causal graph (Monteiro et al., 2023). For CelebA and ADNI, we train deep convolutional regressors as anti-causal predictors for continuous attributes (*e.g.*, brain volume, ventricular volume),

and deep convolutional classifiers for categorical attributes (*e.g.*, age, gender, beard, bald, slice class). Both models use a ResNet-18 backbone pretrained on ImageNet, following the implementation of (Melistas et al., 2024). For the Pendulum dataset, we train regressors for the four continuous attributes following the implementation of (Komanduri et al., 2024c).

**Composition.** If an attribute variable $y_i$ is forced to take the same value $\bar{y}_i$ that it would assume without intervention, the intervention should have no effect on any other variables. This corresponds to the *null intervention*, which leaves all variables unchanged and, in the generative setting, can reduce to a standard reconstruction task. To evaluate counterfactual generation under *null intervention*, we perform abduction via DDIM inversion and prediction via DDIM sampling, but skip the action step that edits the original attribute value. For evaluation, we report the MAE distance between the reconstructed and input images, as well as the Learned Perceptual Image Patch Similarity (LPIPS) (Zhang et al., 2018), which better reflects perceptual fidelity and content preservation.

**Realism.** Realism evaluates the perceptual quality of counterfactual images by measuring their similarity to real samples. We adopt the Fréchet Inception Distance (FID) (Heusel et al., 2017), which quantifies the similarity between the distribution of generated counterfactual images and the real dataset. Specifically, real and counterfactual samples are passed through an Inception v3 network (Szegedy et al., 2015) pretrained on ImageNet to extract high-level semantic feature representations, which are then used to compute the FID score.

**Minimality.** Minimality evaluates whether a counterfactual differs from the factual image only in the intervened parent attribute, ideally leaving all other attributes unaffected. Counterfactual Latent Divergence (CLD) quantifies this by measuring the "distance" between counterfactual and factual images in a latent space (Sanchez & Tsaftaris, 2022). Intuitively, CLD captures a trade-off: the counterfactual should move sufficiently away from the factual class, but not farther than real samples belonging to the counterfactual class. Following Melistas et al. (2024), we compute CLD using an unconditional VAE. Specifically, we measure the KL divergence between the latent distributions of real and counterfactual images. The metric is minimized when both probabilities remain low, reflecting the balance between departing from the factual class while remaining closer to the counterfactual class than unrelated real samples.

### C.3. Full Regularization Algorithm

In the following, we present the fully regularized Causal-Adapter in Algorithms 1–2. Below we summarize the training and inference algorithm in high-level description.

**Training:**

1. *Learn the causal mechanisms $F$:* Using the predefined adjacency matrix $A$ derived from the causal graph and semantic attributes $Y$, the model learns how attributes causally influence one another. This produces a standalone SCM that governs how attributes should update when interventions are applied.

2. *Train the conditional adapter:* Each semantic attribute $y_i$ is projected into a token embedding $v_i(y_i)$ through Prompt-Aligned Injection (PAI). With the diffusion model frozen, the adapter $\ddot{\epsilon}_\psi$ is trained to produce residuals $r_t$ conditioned on the attribute embeddings $V(Y)$. A diffusion loss $\mathcal{L}_{\text{DM}}$ ensures correct denoising behaviour, while a contrastive loss $\mathcal{L}_{\text{CTC}}$ encourages disentanglement and suppresses spurious attribute correlations.

**Inference (Counterfactual Reasoning):**

1. *Abduction:* The input image $x$ is inverted through the diffusion model (via DDIM inversion $H_\theta$) to obtain a latent trajectory $[z_0^\star, \ldots, z_T^\star]$ consistent with the observed image.

2. *Action:* A user-specified intervention $y_i'$ is applied to the semantic attributes. The learned causal mechanisms update all causally connected attributes to produce a new attribute set $\bar{y}_i'$ that respects the causal graph.

3. *Prediction:* Starting from the abducted latent $\bar{z}_T$, the model synthesizes a counterfactual image using the conditional adapter $\ddot{\epsilon}_\psi$ and the updated token embeddings $V\big(Y_{\text{do}(y_i = y_i')}\big)$, yielding the final counterfactual image $\bar{x}$.

---

**Algorithm 1** Causal-Adapter: Training

---

1: **Input:** Image $x$, semantic attributes $Y = \{y_i\}_{i=1}^K$, binary adjacency matrix $A \in \{0,1\}^{K \times K}$, frozen modules $\{\mathcal{E}, c_\phi, \epsilon_\theta\}$, projector $G = \{g_i\}_{i=1}^K$
2: **Output:** causal mechanisms $F = \{f_i\}_{i=1}^K$, updated placeholder embeddings $C = \{c_i\}_{i=1}^K$, updated projector $G = \{g_i\}_{i=1}^K$, causal adapter $\ddot{\epsilon}_\psi$.
3: # Train causal mechanisms $F$
4: $\bar{y}_i := f_i(A_i \odot Y; \omega_i) + u_i, \quad u_i \sim \mathcal{N}(0, \sigma_i^2),$
5: $F := \arg\min_F \mathcal{L}_{\text{NLL}}(y_i, \bar{y}_i)$
6: # Construct attribute injected token embeddings
7: **initialize** placeholder embeddings $C = \{c_i\}_{i=1}^K$,
8: $v_i(y_i) = c_i + g_i(y_i), i = \{1, \cdots, K\}$
9: $V(Y) = [v_1(y_1), \ldots, v_K(y_K)]^\top$
10: # Train conditional adapter
11: **for** $t = 1$ to $T$ **do**
12:     Encode latent: $z_t = \mathcal{E}(x, t)$
13:     Compute residual from causal adapter: $r_t = \ddot{\epsilon}_\psi(z_t, t, V(Y))$
14:     Update parameters: $\psi, G, C := \arg\min_{\psi, G, C} (\mathcal{L}_{\text{DM}} + \mathcal{L}_{\text{CTC}})$
15: **end for**
16: **Return** $F = \{f_i\}_{i=1}^K, \ddot{\epsilon}_\psi, V$

---

---

**Algorithm 2** Causal-Adapter: Inference

---

1: **Input:** Image $x$, semantic attributes $Y$, frozen modules $\{\mathcal{E}, c_\phi, \epsilon_\theta\}$, trained causal mechanisms $F = \{f_i\}_{i=1}^K$, learned placeholder embedding $C = \{c_i\}_{i=1}^K$, trained causal adapter $\ddot{\epsilon}_\psi$, DDIM Inversion operator $H_\theta$ and its generative inverse $H_\theta^{-1}$
2: **Output:** Counterfactual image $\bar{x}$
3: $v_i(y_i) = c_i + g_i(y_i), \quad i = \{1, \cdots, K\}$
4: $V(Y) = [v_1(y_1), \ldots, v_K(y_K)]^\top$
5: # Abduction { infer inversed latent noise
6: $z_0^\star = \mathcal{E}(x, 0)$
7: **for** $t = 0$ to $T - 1$ **do**
8:     $r_t = \ddot{\epsilon}_\psi(z_t^\star, t, V(Y))$
9:     $z_{t+1}^\star = H_\theta(z_t^\star, r_t, V(Y), t)$
10: **end for**
11: # Action { apply intervention with $do(y_i = y_i')$
12: $\bar{y}_i' := f_i(A_i \odot Y_{do(y_i = y_i')}; \omega_i) + u_i, \quad u_i \sim \mathcal{N}(0, \sigma_i^2), \quad i = \{1, \cdots, K\}$
13: $v_i(\bar{y}_i') = c_i + g_i(\bar{y}_i'), \quad i = \{1, \cdots, K\}$
14: $V(Y_{do(y_i = y_i')}) = [v_1(\bar{y}_1'), \ldots, v_K(\bar{y}_K')]^\top$
15: # Prediction { generate counterfactual
16: **initialize** $\bar{z}_T \leftarrow z_T^\star$
17: **for** $t = T$ to $1$ **do**
18:     $\bar{r}_t = \ddot{\epsilon}_\psi(z_t^\star, t, V(Y_{do(y_i = y_i')}))$
19:     $\bar{z}_{t-1} = H_\theta^{-1}(\bar{z}_t, \bar{r}_t, V(Y_{do(y_i = y_i')}), t)$
20: **end for**
    $\bar{x} = \text{Decode}(\bar{z}_0)$
21: **Return** $\bar{x}$

---

## D. Extended Ablation Results [Back to Contents]

We further investigate the impact of guidance scale (Appendix D.1), DDIM steps ( Appendix D.2),, and attention guidance on counterfactual generation(Appendix D.3), and provide additional qualitative evidence (Appendix D.4). To reduce computational cost, these experiments are conducted on the CelebA validation set using 400 random samples.

### D.1. Effect of Guidance Scale

We study the influence of the classifier-free guidance scale $\alpha$ on intervention effectiveness, realism, and minimality, as summarized in Table 7 and Figure 15. Increasing $\alpha$ consistently improves intervention effectiveness across attributes, with all variants showing stronger F1-scores as interventions become more easily separable by the anti-causal predictor (classifier). For instance, the fully regularized model (Table 7 rows 11-15) improves Age F1 from 42.0 at $\alpha = 1.0$ to 63.4 at $\alpha = 5.0$, with similar improvement for beard and bald interventions. These results suggest that aligning semantic and spatial features and enforcing disentanglement allow more effective editing, particularly at higher guidance scales. This effectiveness improvement comes at a cost: both FID, reflecting reduced realism and causal minimality (Figure 15). Thus, we adopt $\alpha = 3.0$ as the default setting, where intervention effectiveness is already strong across all variants (Table 7 rows 3,8,13) while FID and CLD remain relatively low. Further increasing $\alpha$ beyond this point will leads to marginal improvements in effectiveness but rapidly increases FID and CLD. Qualitative example is shown in Figure 17.

*Table 7.* Influence of guidance scale on counterfactual effectiveness. $\alpha$ denotes the classifier-free guidance scale. "Ours" refers to the plain Causal-Adapter, "Ours$^\star$" to the regularized Causal-Adapter (+PAI), and "Ours$^{\star\star}$" to the fully regularized Causal-Adapter (+PAI+CTC).

| | Method | Age (a) F1 ↑ | | | | Gender (g) F1 ↑ | | | | Beard (br) F1 ↑ | | | | Bald (bl) F1 ↑ | | | |
|---|---|---|---|---|---|---|---|---|---|---|---|---|---|---|---|---|---|
| | | $do(a)$ | $do(g)$ | $do(br)$ | $do(bl)$ | $do(a)$ | $do(g)$ | $do(br)$ | $do(bl)$ | $do(a)$ | $do(g)$ | $do(br)$ | $do(bl)$ | $do(a)$ | $do(g)$ | $do(br)$ | $do(bl)$ |
| 1. | Ours($\alpha = 1.0$) | 38.0 | 79.9 | 86.0 | 77.8 | 96.1 | 97.1 | 65.1 | 86.6 | 95.4 | 75 | 32.2 | 93.7 | 100 | 39.7 | 66.7 | 22.9 |
| 2. | Ours($\alpha = 2.0$) | 38.1 | 80.2 | 95.8 | 75.5 | 95.5 | 96.7 | 61 | 82.3 | 95 | 78 | 39.2 | 93 | 100 | 50 | 85.7 | 32 |
| 3. | Ours($\alpha = 3.0$) | 38.2 | 81.1 | 85.3 | 73.9 | 95.3 | 95.6 | 59.7 | 77.8 | 93.9 | 81.3 | 42.1 | 92.2 | 100 | 61.3 | 85.7 | 39.8 |
| 4. | Ours($\alpha = 4.0$) | 37.2 | 80.1 | 84.8 | 72.6 | 93.7 | 95.2 | 58.7 | 75.4 | 92.8 | 82.4 | 45.2 | 91.3 | 100 | 66.7 | 75 | 50.2 |
| 5. | Ours($\alpha = 5.0$) | 37.1 | 80.7 | 83.1 | 71 | 92.8 | 94.8 | 57.8 | 71.5 | 92.3 | 84.1 | 43.4 | 90.1 | 80 | 67 | 85.7 | 54.9 |
| 6. | Ours$^\star$($\alpha = 1.0$) | 37.5 | 78.1 | 87.6 | 70.8 | 96.1 | 96.8 | 71.7 | 86.1 | 95.1 | 77.4 | 29 | 93.3 | 100 | 44.2 | 85.7 | 22.5 |
| 7. | Ours$^\star$($\alpha = 2.0$) | 46.8 | 85.4 | 93 | 84.3 | 99 | 99.8 | 72.8 | 95.5 | 98.6 | 93.7 | 43.8 | 97 | 50 | 84.8 | 50 | 47.2 |
| 8. | Ours$^\star$($\alpha = 3.0$) | 48.5 | 90.8 | 92.7 | 89.4 | 99 | 100 | 72.1 | 97 | 99.4 | 97.4 | 50.7 | 97.3 | 50 | 91.8 | 66.7 | 55.8 |
| 9. | Ours$^\star$($\alpha = 4.0$) | 48.8 | 90.2 | 92.2 | 90.6 | 99.7 | 100 | 71.1 | 97.5 | 99.7 | 98.6 | 53 | 98.6 | 50 | 94.7 | 54.5 | 57.7 |
| 10. | Ours$^\star$($\alpha = 5.0$) | 48.6 | 90 | 91.6 | 91.8 | 100 | 100 | 70.5 | 97.5 | 100 | 99.1 | 54.1 | 99 | 66.7 | 94.3 | 40 | 58.2 |
| 11. | Ours$^{\star\star}$($\alpha = 1.0$) | 42.0 | 80.2 | 86.0 | 73.6 | 96.2 | 97.7 | 67.8 | 81.5 | 94.9 | 78.0 | 31.5 | 92.9 | 1 | 45.9 | 75.0 | 29.4 |
| 12. | Ours$^{\star\star}$($\alpha = 2.0$) | 51.9 | 88.0 | 93.4 | 84.9 | 98.1 | 99.6 | 72.3 | 90.1 | 98.8 | 91.0 | 46.4 | 97.0 | 80.0 | 84.2 | 75.0 | 55.1 |
| 13. | Ours$^{\star\star}$($\alpha = 3.0$) | 58.8 | 91.3 | 94.4 | 89.4 | 100 | 99.8 | 74.5 | 91.1 | 99.7 | 96.4 | 53.4 | 97.8 | 80.0 | 92.4 | 66.7 | 59.8 |
| 14. | Ours$^{\star\star}$($\alpha = 4.0$) | 61.3 | 89.9 | 94.7 | 91.5 | 99.7 | 100 | 77.4 | 92.2 | 100 | 97.2 | 55.6 | 97.6 | 75.0 | 92.4 | 66.7 | 59.8 |
| 15. | Ours$^{\star\star}$($\alpha = 5.0$) | 63.4 | 90.7 | 95.2 | 91.4 | 99.7 | 100 | 79.0 | 93.5 | 100 | 98.0 | 57.0 | 97.8 | 75.0 | 92.4 | 66.7 | 63.0 |

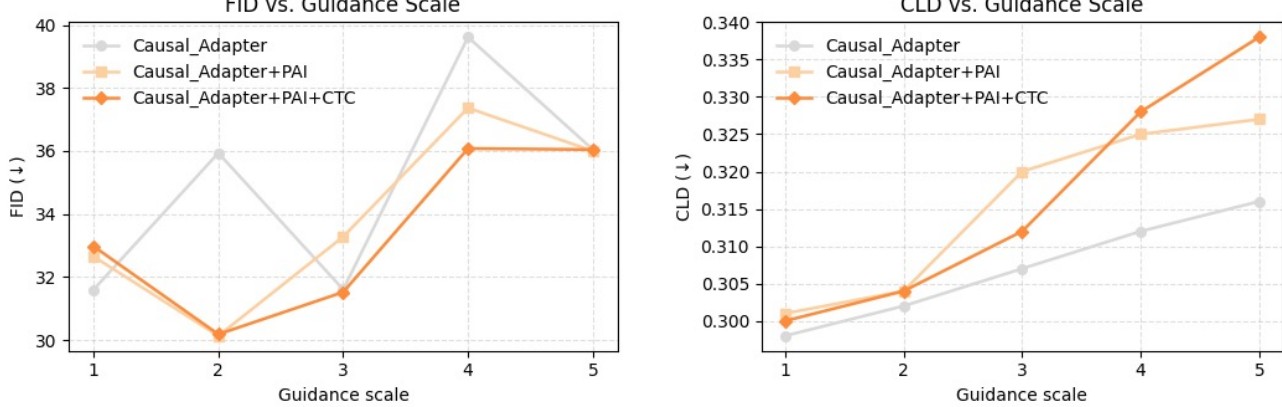

*Figure 15.* Impact of guidance scale on FID and CLD across three Causal-Adapter variants. Note that FID values here are higher than in the main manuscript due to the smaller evaluation set, which shifts the distribution mean and variance. The implied relative trends in image quality remain consistent.

## D.2. Effect of DDIM Steps

We further evaluate the effect of DDIM steps on counterfactual effectiveness using our fully regularized model (Table 8). As $T$ increases, intervention effectiveness remains stable, indicating that performance is insensitive to the number of denoising steps. Realism and minimality also show negligible variation. The main drawback of larger $T$ lies in increased inference time: generating one counterfactual takes 5–7 seconds at $T = 100$, about 20 seconds at $T = 200$, and nearly 40 seconds at $T = 500$. Balancing efficiency and accuracy, we adopt $T = 100$ as the default setting for all experiments.

*Table 8.* Influence of DDIM steps under $\alpha = 3.0$ on counterfactual effectiveness. $T$ denote the used steps.

| | Method | Age (a) F1 ↑ | | | | Gender (g) F1 ↑ | | | | Beard (br) F1 ↑ | | | | Bald (bl) F1 ↑ | | | |
|---|---|---|---|---|---|---|---|---|---|---|---|---|---|---|---|---|---|
| | | $do(a)$ | $do(g)$ | $do(br)$ | $do(bl)$ | $do(a)$ | $do(g)$ | $do(br)$ | $do(bl)$ | $do(a)$ | $do(g)$ | $do(br)$ | $do(bl)$ | $do(a)$ | $do(g)$ | $do(br)$ | $do(bl)$ |
| 1. | Ours**$(T = 50)$ | 57.1 | 90.5 | 93.8 | 89.4 | 99.7 | 99.6 | 75.7 | 91.6 | 99.7 | 96.1 | 49.1 | 98 | 80.0 | 89.4 | 75.0 | 60 |
| 2. | Ours**$(T = 100)$ | 58.8 | 91.3 | 94.4 | 89.4 | 100 | 99.8 | 74.5 | 91.1 | 99.7 | 96.4 | 53.4 | 97.8 | 80.0 | 92.4 | 66.7 | 61.8 |
| 3. | Ours**$(T = 200)$ | 60.5 | 91.7 | 94.9 | 89.6 | 100 | 99.8 | 74.1 | 91.4 | 99.7 | 96.4 | 54.3 | 97.4 | 80.0 | 91.9 | 66.7 | 61.3 |
| 4. | Ours**$(T = 500)$ | 60.7 | 91.6 | 94.7 | 90.0 | 100 | 99.8 | 73.8 | 91.9 | 99.7 | 96.1 | 54.8 | 98.0 | 85.7 | 91.9 | 66.7 | 61.3 |

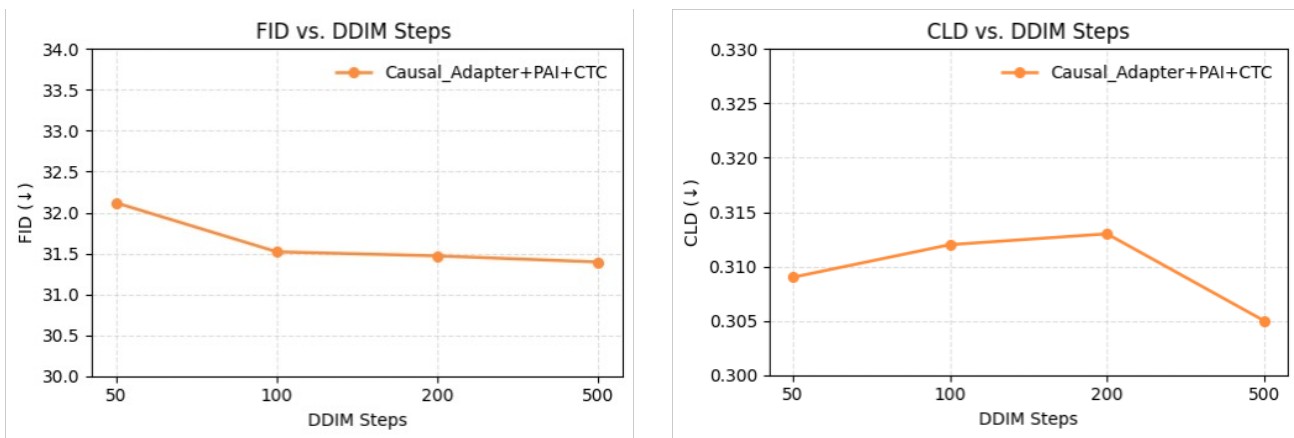

*Figure 16.* Impact of DDIM steps on FID and CLD

**D.3. Investigation of Attention Guidance**

Attention guidance (AG) has been proposed as an external mechanism to enforce localized edits. While our primary contribution lies in Causal-Adapter and its regularizers, we further examine whether AG can complement different variants of our model. As shown in Tables 9–10, incorporating AG sometimes reduces intervention effectiveness, particularly for the plain adapter (Table 9 rows 1,3). For example, Age F1 drops sharply from 38.2 to 28.1 when switching from global to local editing, reflecting misaligned attention maps that disrupt counterfactual consistency. In contrast, the regularized variants exhibit only mild decreases (Table 9 rows 2,5), indicating that aligning semantic and spatial features in diffusion latents via PAI is a prerequisite for stable local editing. Our fully regularized model (Table 9 rows 3,6) not only maintains strong intervention effectiveness but also achieves the lowest FID (31.216) with AG, highlighting that PAI and CTC help produce more precise attention maps. By comparison, applying AG to the plain adapter actually worsens FID and CLD (Table 10 row 4), suggesting that noisy or misaligned attention can harm editing quality. Qualitative evidence is provided in Figures 18–19.

We emphasize that while AG can be beneficial for identity preservation in human faces (CelebA), in other domains such as ADNI or Pendulum, our model already produces accurate, faithful, and identity-preserving counterfactuals without AG. This underscores that our contribution lies not in AG itself, but in designing Causal-Adapter such that causal attributes are naturally aligned with token embeddings, enabling both disentanglement and, if desired, effective integration with AG for localized editing.

*Table 9.* Effectiveness of Causal-Adapter variants with and without attention guidance.

| Method | Age (a) F1 ↑ | | | | Gender (g) F1 ↑ | | | | Beard (br) F1 ↑ | | | | Bald (bl) F1 ↑ | | | |
|---|---|---|---|---|---|---|---|---|---|---|---|---|---|---|---|---|
| | $do(a)$ | $do(g)$ | $do(br)$ | $do(bl)$ | $do(a)$ | $do(g)$ | $do(br)$ | $do(bl)$ | $do(a)$ | $do(g)$ | $do(br)$ | $do(bl)$ | $do(a)$ | $do(g)$ | $do(br)$ | $do(bl)$ |
| *without attention guidance for global editing* | | | | | | | | | | | | | | | | |
| 1. Ours | 38.2 | 81.1 | 85.3 | 73.9 | 95.3 | 95.6 | 59.7 | 77.8 | 93.9 | 81.3 | 42.1 | 92.2 | 100 | 61.3 | 85.7 | 39.8 |
| 2. Ours* | 48.5 | 90.8 | 92.7 | 89.4 | 99 | 100 | 72.1 | 97 | 99.4 | 97.4 | 50.7 | 97.3 | 50 | 91.8 | 66.7 | 55.8 |
| 3. Ours** | 58.8 | 91.3 | 94.4 | 89.4 | 100 | 99.8 | 74.5 | 91.1 | 99.7 | 96.4 | 53.4 | 97.8 | 80.0 | 92.4 | 66.7 | 61.8 |
| *with attention guidance for local editing* | | | | | | | | | | | | | | | | |
| 4. Ours | 28.1 | 79.4 | 83.2 | 63.3 | 94.7 | 93.4 | 62.4 | 78.1 | 96.5 | 80.6 | 40.4 | 89.4 | 66.7 | 70.6 | 33.3 | 34.5 |
| 5. Ours* | 49.4 | 88.8 | 91.2 | 73.9 | 98.8 | 96.4 | 71.1 | 92.8 | 99.5 | 96.1 | 47.4 | 94.6 | 22.2 | 76.2 | 40 | 52.4 |
| 6. Ours** | 57.1 | 88.7 | 94 | 77 | 99.4 | 100 | 73.3 | 90.4 | 99.7 | 95.6 | 49.4 | 95.1 | 66.7 | 80.7 | 50 | 54.4 |

*Table 10.* Realism and minimality of Causal-Adapter variants with and without attention guidance

| Method | Realism | Minimality |
|---|---|---|
| | FID ↓ | CLD ↓ |
| *without attention guidance for global editing* | | |
| 1. Ours | 31.604 | 0.307 |
| 2. Ours* | 33.278 | 0.320 |
| 3. Ours** | 31.52 | 0.312 |
| *with attention guidance for localized editing* | | |
| 4. Ours | 33.849 | 0.311 |
| 5. Ours** | 31.885 | 0.299 |
| 6. Ours** | 31.216 | 0.300 |

## D.4. Qualitative Evidence

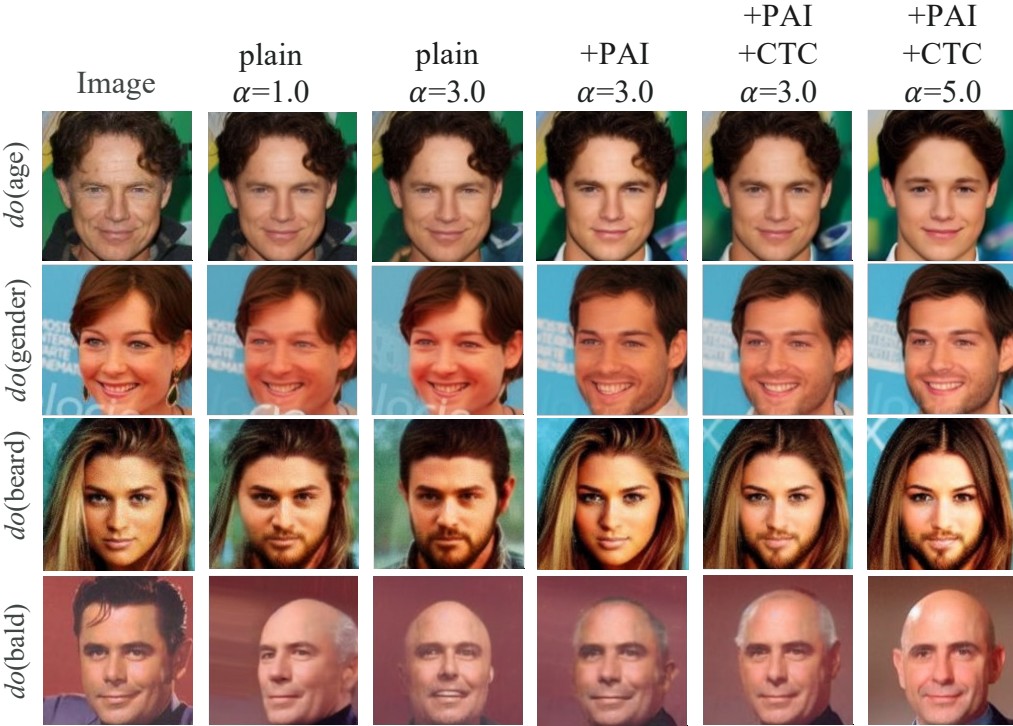

*Figure 17.* Counterfactuals from Causal-Adapter variants under different guidance scales. The plain variant shows weak effectiveness at $\alpha = 1.0$ and 3.0. Adding PAI strengthens the editing signal, while further incorporating CTC yields the most effective edits with strong identity preservation. At $\alpha = 5.0$, the counterfactuals become slightly over-edited.

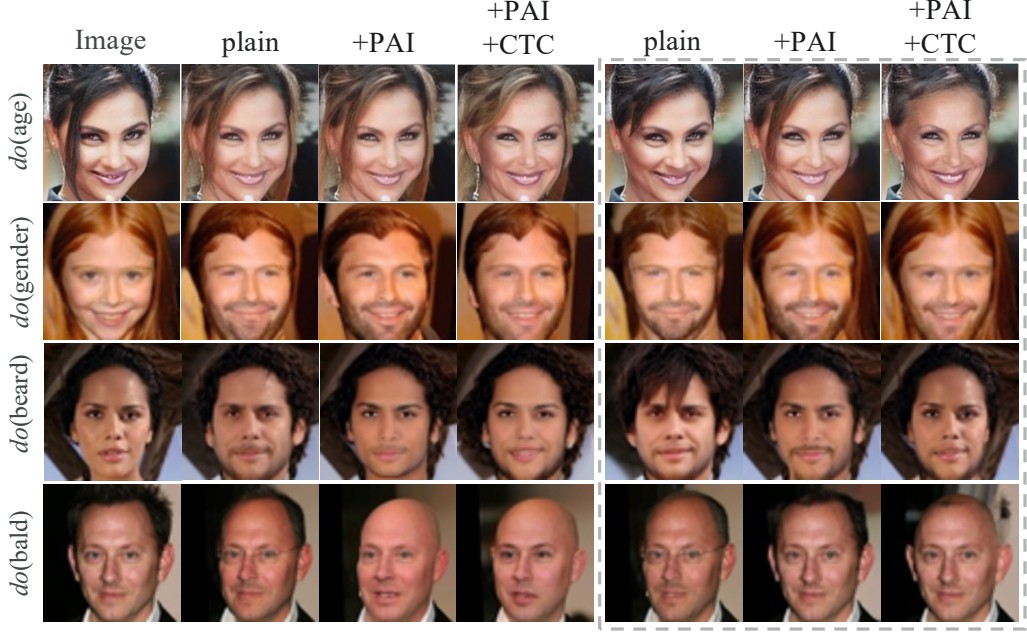

*Figure 18.* Full ablation visualizations with optional attention guidance (AG). Causal-Adapter with the two regularizers reduces spurious correlations, AG can enhance identity preservation through localized editing. Dotted boxes indicate results with AG-based localized edits.

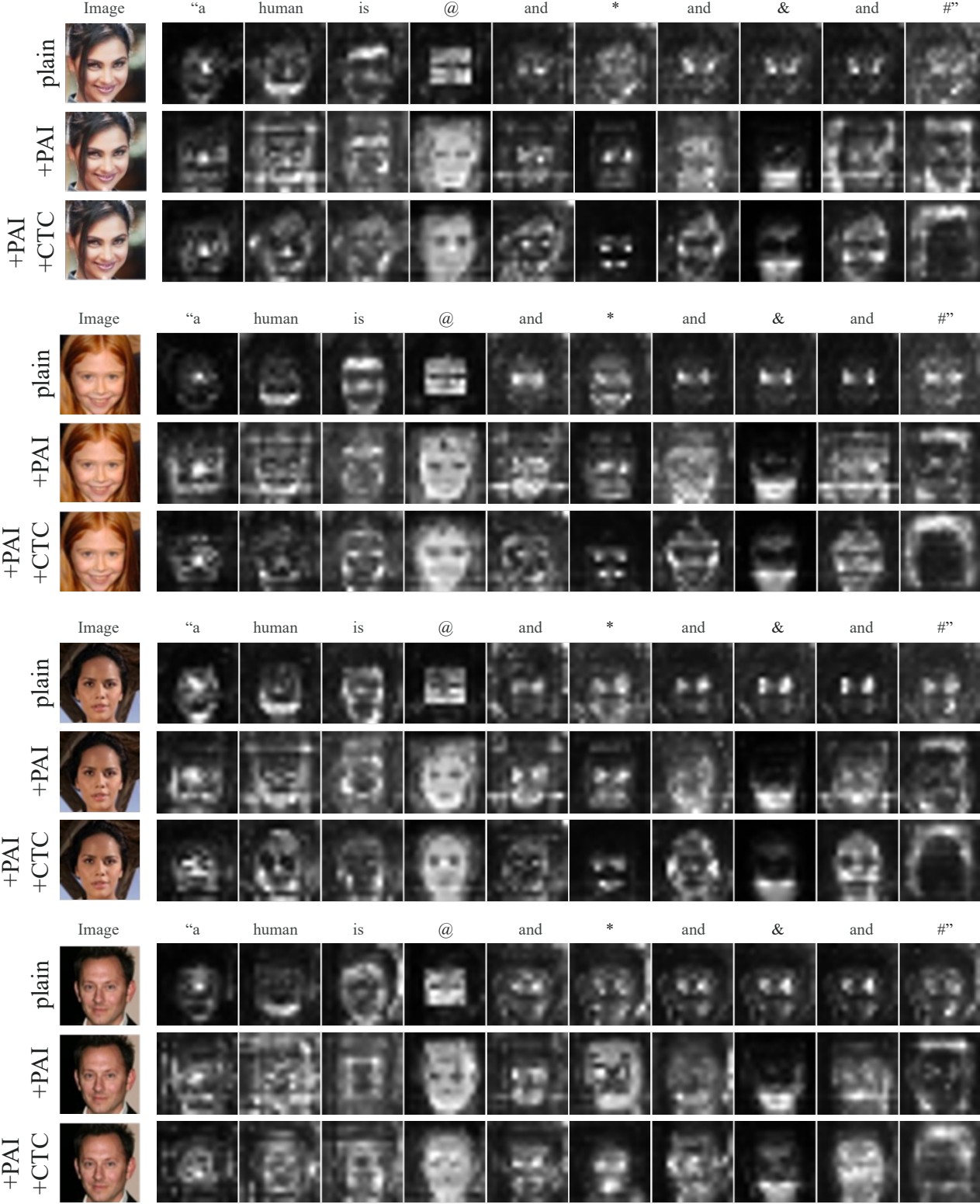

*Figure 19.* Average cross-attention maps from Causal-Adapter variants. Tokens denote attributes: "@" for age, "*" for gender, "&" for beard, and "#" for bald. The plain adapter fails to align semantics with spatial features, producing poor maps. Adding PAI improves alignment but some tokens (e.g., "*", "#") remain entangled. With both PAI and CTC, token embedding disentanglement is enforced, and attentions are clearly localized to the semantic regions.

# E. Additional Results [Back to Contents]

## E.1. Pendulum

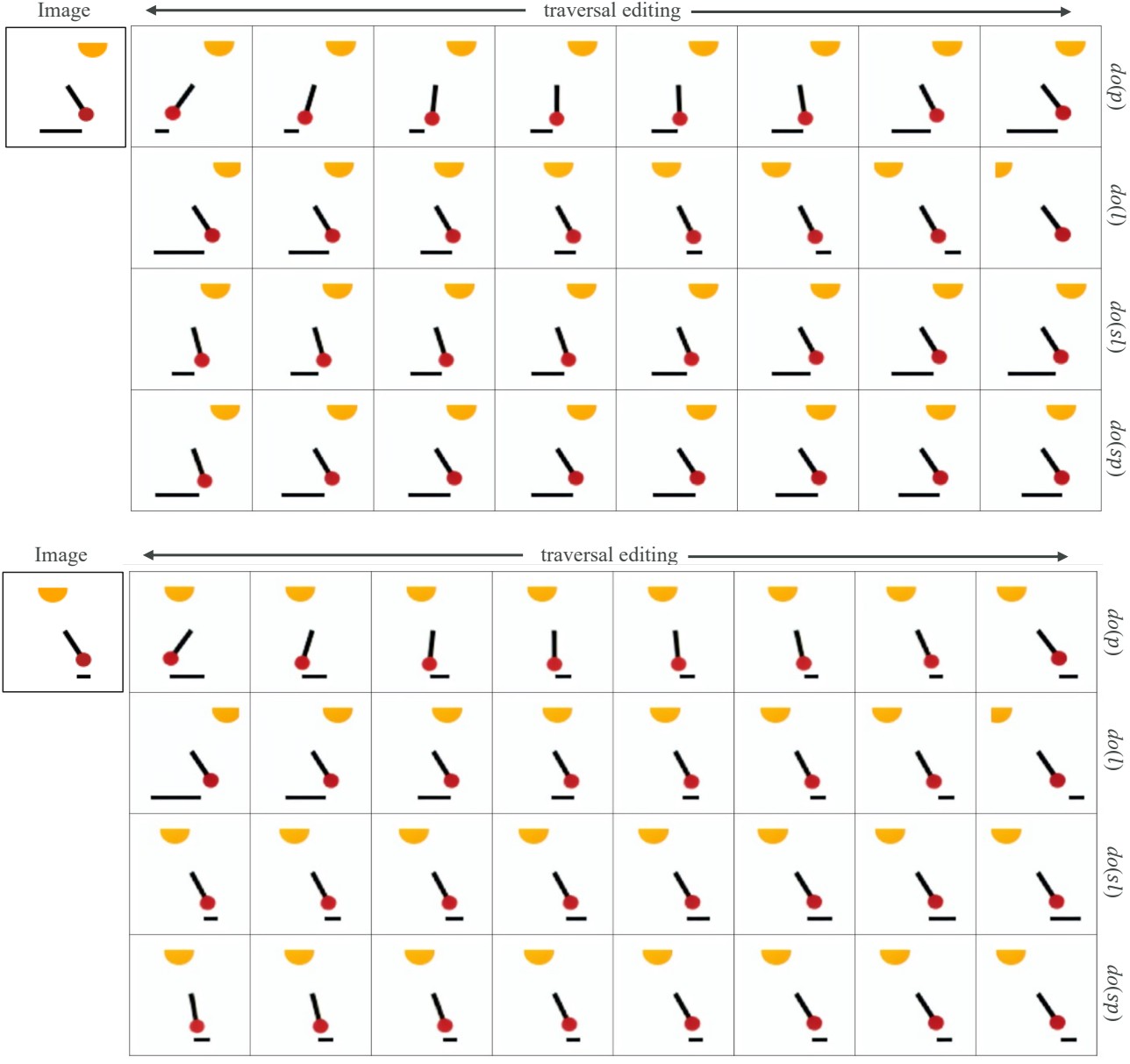

*Figure 20.* Pendulum counterfactuals from Causal-Adapter. $p$ for pendulum, $l$ for light, $sl$ for shadow length and $sp$ for shadow position. Traversal editing across four attributes demonstrates that our method produces high-quality, fine-grained generations of attribute variations.

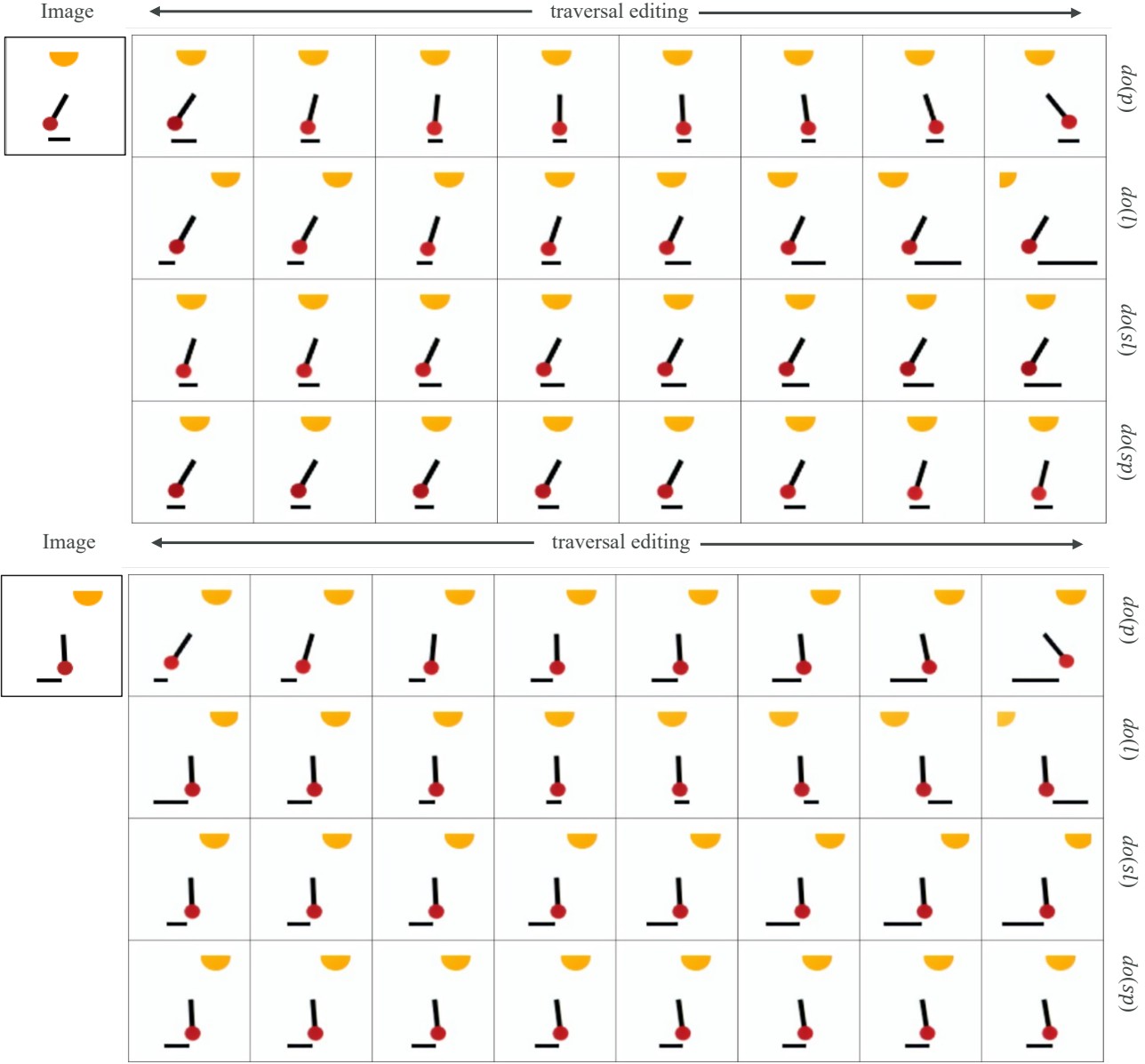

*Figure 21.* Pendulum counterfactuals from Causal-Adapter. $p$ for pendulum, $l$ for light, $sl$ for shadow length and $sp$ for shadow position.

## E.2. CelebA

We report the full CelebA results with the corresponding means and standard deviations in Tables 11–13. Compared with VAE, HVAE, and GAN, our method achieves best performance across most interventions, including up to an $86\%$ reduction in LPIPS (composition, from 0.122 to 0.017) and an $79\%$ reduction in FID (realism, from 27.861 to 8.152). Besides, our method delivers significant improvements under $do(a)$ and $do(br)$ on F1 score of bald attribute, demonstrating causal faithfulness in representing baldness across diverse individuals. As illustrated in Figure 22, Causal-Adapter can successfully add baldness to both male and female under $do(bl)$, and under interventions on causal parents such as $do(a)$, it can jointly edit age and baldness, whereas baselines fail to maintain causal consistency.

Table 13 additionally reports the LPIPS distance between each counterfactual and its original image, measuring identity preservation under different attribute interventions.

*Table 11.* Intervention effectiveness on CelebA test set. Age and Gender F1 under four interventions.

| | Method | Age $(a)$ **F1** $\uparrow$ | | | | Gender $(g)$ **F1** $\uparrow$ | | | |
|---|---|---|---|---|---|---|---|---|---|
| | | $do(a)$ | $do(g)$ | $do(br)$ | $do(bl)$ | $do(a)$ | $do(g)$ | $do(br)$ | $do(bl)$ |
| 1. | VAE | $35.0_{0.04}$ | $78.2_{0.02}$ | $81.6_{0.02}$ | $81.9_{0.02}$ | $97.7_{0.01}$ | $90.9_{0.02}$ | $95.9_{0.02}$ | $\mathbf{97.3}_{0.01}$ |
| 2. | HVAE | $\mathbf{65.4}_{0.10}$ | $89.3_{0.04}$ | $90.8_{0.03}$ | $\mathbf{89.9}_{0.03}$ | $98.8_{0.02}$ | $94.9_{0.03}$ | $\mathbf{99.4}_{0.01}$ | $95.0_{0.03}$ |
| 3. | GAN | $41.3_{0.04}$ | $71.0_{0.02}$ | $81.8_{0.02}$ | $79.9_{0.01}$ | $95.2_{0.01}$ | $98.2_{0.01}$ | $92.0_{0.01}$ | $96.1_{0.01}$ |
| 4. | Ours | $58.5_{0.14}$ | $\mathbf{89.9}_{0.35}$ | $94.0_{0.07}$ | $89.4_{0.00}$ | $\mathbf{99.6}_{0.00}$ | $\mathbf{99.9}_{0.00}$ | $74.5_{0.07}$ | $92.5_{0.21}$ |
| | *with attention guidance for localized editing* | | | | | | | | |
| 5. | Ours (AG) | $57.1_{0.14}$ | $89.5_{1.76}$ | $\mathbf{94.5}_{0.00}$ | $81.5_{0.63}$ | $\mathbf{99.6}_{0.00}$ | $99.7_{0.00}$ | $73.8_{0.56}$ | $89.2_{0.14}$ |

*Table 12.* Intervention effectiveness on CelebA test set. Beard and Bald F1 under four interventions.

| | Method | Beard $(br)$ **F1** $\uparrow$ | | | | Bald $(bl)$ **F1** $\uparrow$ | | | |
|---|---|---|---|---|---|---|---|---|---|
| | | $do(a)$ | $do(g)$ | $do(br)$ | $do(bl)$ | $do(a)$ | $do(g)$ | $do(br)$ | $do(bl)$ |
| 1. | VAE | $94.4_{0.01}$ | $82.8_{0.03}$ | $29.6_{0.05}$ | $94.5_{0.02}$ | $2.3_{0.03}$ | $49.6_{0.05}$ | $4.5_{0.04}$ | $41.2_{0.03}$ |
| 2. | HVAE | $95.2_{0.03}$ | $95.1_{0.03}$ | $44.1_{0.11}$ | $91.6_{0.04}$ | $2.0_{0.05}$ | $86.0_{0.05}$ | $4.5_{0.07}$ | $\mathbf{61.1}_{0.04}$ |
| 3. | GAN | $90.8_{0.01}$ | $83.8_{0.02}$ | $23.3_{0.03}$ | $90.7_{0.01}$ | $2.1_{0.02}$ | $82.0_{0.02}$ | $5.5_{0.02}$ | $49.2_{0.02}$ |
| 4. | Ours | $\mathbf{99.7}_{0.00}$ | $96.1_{0.00}$ | $\mathbf{52.1}_{0.21}$ | $\mathbf{98.1}_{0.07}$ | $\mathbf{59.2}_{0.91}$ | $\mathbf{91.2}_{0.28}$ | $\mathbf{74.5}_{2.75}$ | $58.8_{0.35}$ |
| | *with attention guidance for localized editing* | | | | | | | | |
| 5. | Ours (AG) | $99.7_{0.07}$ | $\mathbf{96.8}_{0.00}$ | $48.2_{0.14}$ | $96.6_{0.07}$ | $38.7_{1.90}$ | $78.4_{0.49}$ | $47.7_{1.76}$ | $51.4_{0.49}$ |

*Table 13.* Identity preservation, realism, and minimality on the CelebA test set. VAE, HVAE, and GAN are reported as single-run results from prior work, while our methods report mean$_{\text{std}}$ over three random seeds. "–" indicates that LPIPS was not reported for the corresponding method.

| | Method | Identity Preservation (LPIPS $\downarrow$) | | | | Composition | | Realism | Minimality |
|---|---|---|---|---|---|---|---|---|---|
| | | Age | Gender | Bearded | Bald | MAE $\downarrow$ | LPIPS $\downarrow$ | FID $\downarrow$ | CLD $\downarrow$ |
| 1. | VAE | – | – | – | – | 18.695 | 0.282 | 59.393 | $\mathbf{0.299}$ |
| 2. | HVAE | – | – | – | – | 7.143 | 0.122 | 35.712 | 0.305 |
| 3. | GAN | – | – | – | – | 60.120 | 0.276 | 27.861 | 0.304 |
| 4. | Ours | $0.087_{0.0066}$ | $0.157_{0.0067}$ | $0.0797_{0.0060}$ | $0.157_{0.0073}$ | $\mathbf{3.535}_{0.000}$ | $\mathbf{0.017}_{0.000}$ | $8.152_{0.365}$ | $0.310_{0.001}$ |
| | *Ours with attention guidance for improved realism and minimality* | | | | | | | | |
| 5. | Ours (AG) | $\mathbf{0.061}_{0.0171}$ | $\mathbf{0.072}_{0.0178}$ | $\mathbf{0.0330}_{0.0088}$ | $\mathbf{0.109}_{0.0192}$ | – | – | $\mathbf{5.213}_{0.173}$ | $0.301_{0.001}$ |

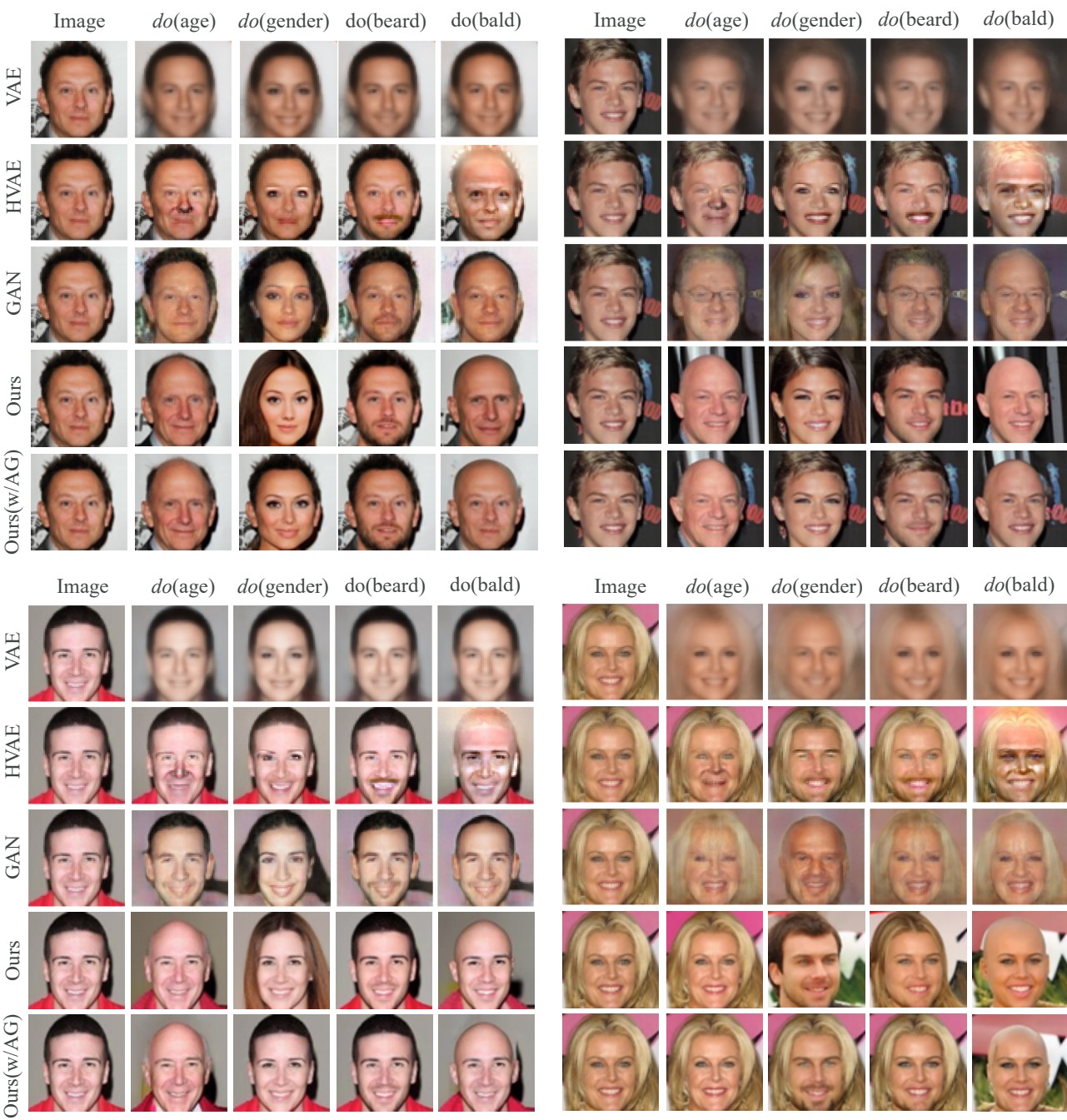

*Figure 22.* Additional counterfactual results on the CelebA dataset (with edit samples selected in a non-cherrypicked manner following (Melistas et al., 2024)). Our Causal-Adapter effectively disentangles target attributes compared with prior methods and achieve faithful counterfactual generations.

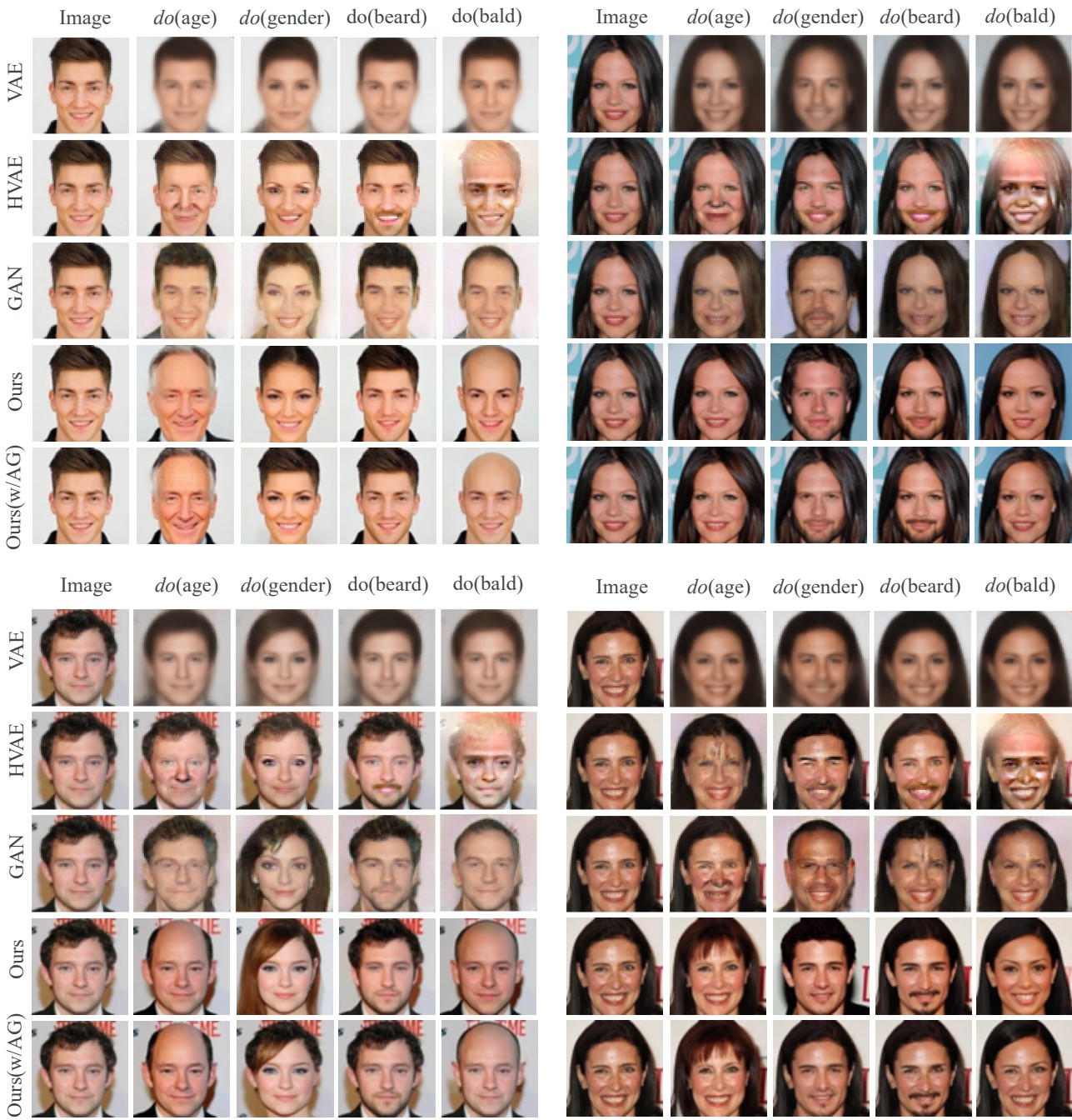

*Figure 23.* Additional counterfactual results on the CelebA dataset (with edit samples selected in a non-cherrypicked manner following (Melistas et al., 2024)).

### E.3. ADNI

We intervene on three generative conditioning attributes (Brain Volume, Ventricular Volume, Slice) and report the results in Table 14. Our approach achieves best performance in intervention effectiveness and minimality, while also delivering strong realism. Qualitative results in Figure 24-25 further show that our model produces sharp and localized interventional changes consistent with the causal graph. For example, edits to ApoE, Age, or Sex appropriately influence Brain and Ventricular Volumes. In particular, fine-grained edits to Ventricular Volume visibly adjust the ventricle region while faithfully preserving subject identity.

*Table 14.* Effectiveness on ADNI test set, evaluated with MAE/F1 from pretrained regressors/classifiers (std. dev. reported). Composition, realism, and minimality metrics are also included.

| | Method | Brain volume (b) MAE ↓ | | | Ventricular volume (v) MAE ↓ | | | Slice (s) F1 ↑ | | | Composition | | Realism | Minimality |
|---|---|---|---|---|---|---|---|---|---|---|---|---|---|---|
| | | $do(b)$ | $do(v)$ | $do(s)$ | $do(b)$ | $do(v)$ | $do(s)$ | $do(b)$ | $do(v)$ | $do(s)$ | MAE ↓ | LPIPS↓ | FID↓ | CLD↓ |
| 1. | VAE | $0.17_{0.03}$ | $0.15_{0.06}$ | $0.15_{0.06}$ | $0.08_{0.05}$ | $0.20_{0.04}$ | $0.08_{0.05}$ | $0.52_{0.15}$ | $0.48_{0.15}$ | $0.46_{0.10}$ | 18.88 | 0.306 | 278.245 | 0.352 |
| 2. | HVAE | $0.09_{0.03}$ | $0.12_{0.06}$ | $0.13_{0.06}$ | $0.06_{0.04}$ | $0.04_{0.01}$ | $0.06_{0.04}$ | $0.38_{0.15}$ | $0.41_{0.16}$ | $0.41_{0.11}$ | **3.38** | 0.101 | 74.696 | 0.347 |
| 3. | GAN | $0.17_{0.02}$ | $0.16_{0.07}$ | $0.16_{0.06}$ | $0.12_{0.02}$ | $0.22_{0.03}$ | $0.12_{0.03}$ | $0.14_{0.03}$ | $0.16_{0.03}$ | $0.05_{0.02}$ | 24.26 | 0.268 | 113.749 | 0.353 |
| 4. | Ours | $\textbf{0.09}_{0.01}$ | $\textbf{0.11}_{0.03}$ | $\textbf{0.11}_{0.03}$ | $\textbf{0.03}_{0.01}$ | $\textbf{0.03}_{0.01}$ | $\textbf{0.03}_{0.01}$ | $0.53_{0.09}$ | $0.55_{0.09}$ | $\textbf{0.48}_{0.06}$ | 3.54 | **0.035** | 9.130 | 0.346 |
| *with attention guidance for localized editing* | | | | | | | | | | | | | | |
| 5. | Ours | $0.10_{0.01}$ | $0.14_{0.04}$ | $0.14_{0.06}$ | $0.10_{0.02}$ | $0.04_{0.01}$ | $0.10_{0.02}$ | $\textbf{0.55}_{0.08}$ | $\textbf{0.57}_{0.08}$ | $0.46_{0.08}$ | - | - | **9.066** | **0.332** |

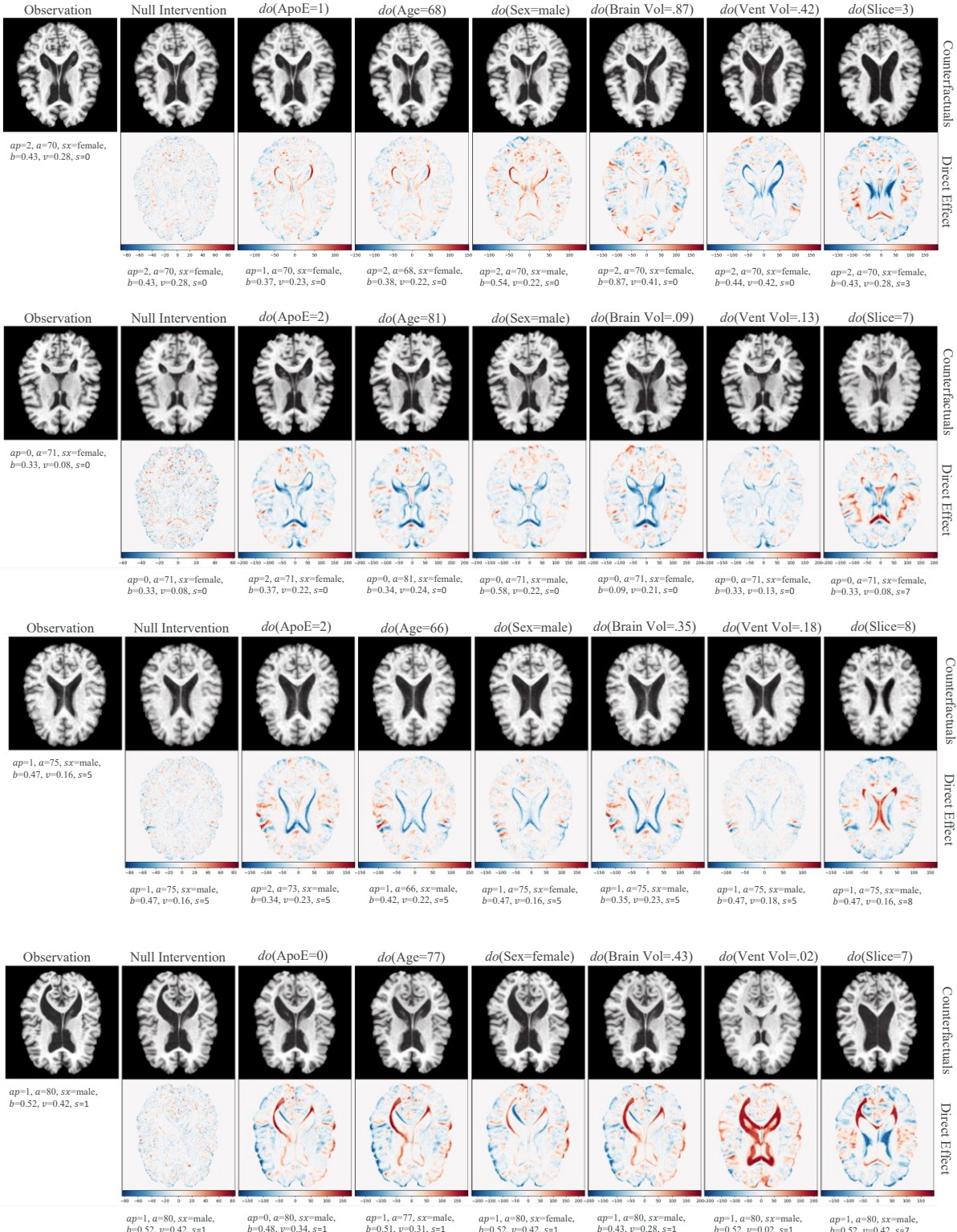

*Figure 24.* Additional counterfactual results from random interventions on each attribute in the ADNI dataset (non-cherrypicked). We observe localized changes consistent with the performed interventions and the assumed causal graph. Importantly, the identity of the original observation is well preserved, demonstrating the effectiveness of Causal-Adapter.

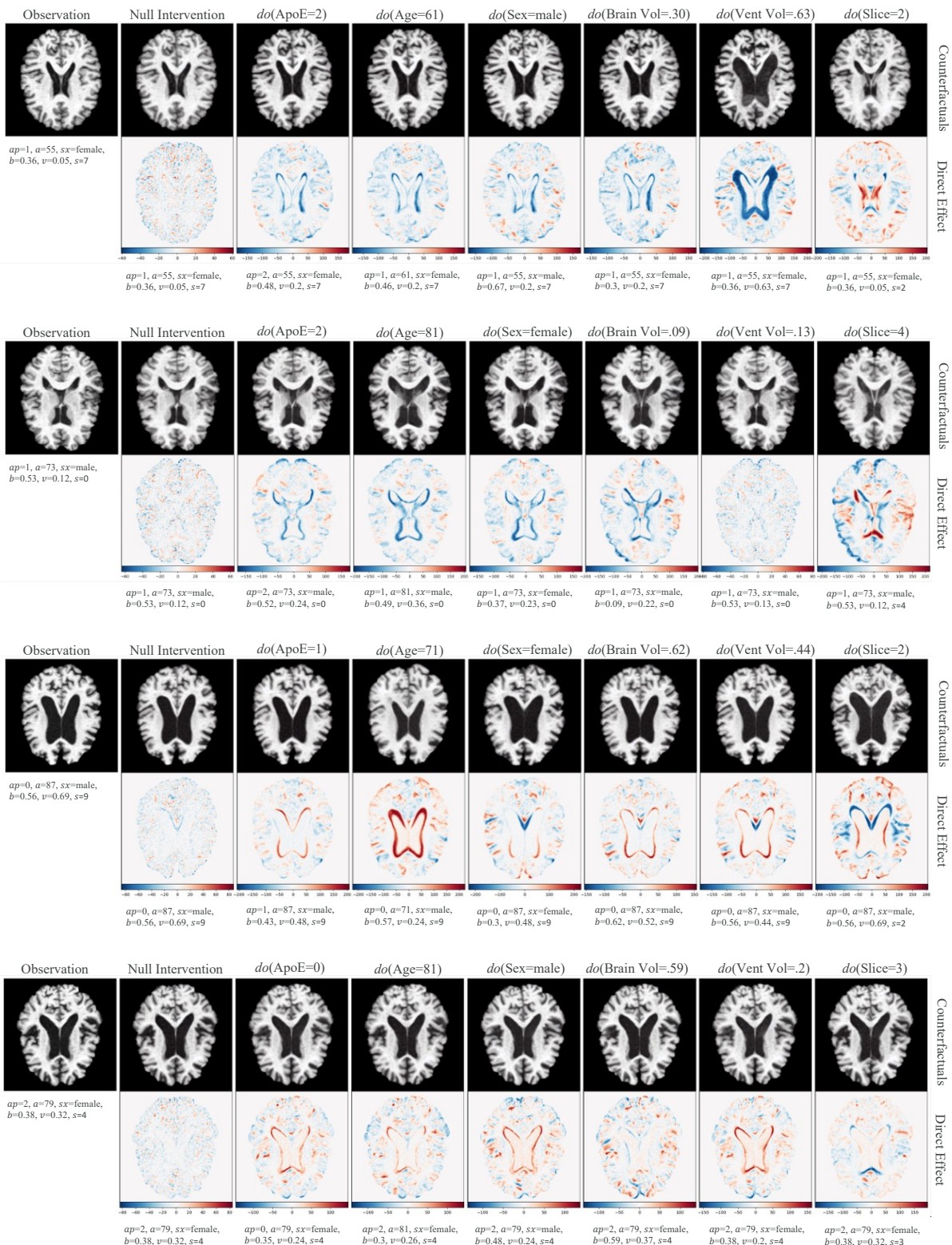

*Figure 25.* Additional counterfactual results from random interventions on each attribute in the ADNI dataset

## E.4. CelebA-HQ

We compare Causal-Adapter with recent state-of-the-art counterfactual methods including VCI (Wu et al., 2025), HVAE (De Sousa Ribeiro et al., 2023), and DiffCounter (Rasal et al., 2025) on high-resolution human face dataset CelebA-HQ. We follow the same settings in DiffCounter for fair comparison, focusing on three categorical variables (glasses, smile, mouth-open). The quantitative results are presented in Table 15, with an additional reversibility metric that measures how well the generated counterfactuals can be recovered back to the original observations. Our Causal-Adapter (row 4) achieves intervention effectiveness comparable to DiffCounter, while substantially improving both reversibility and identity preservation. For example, under the smile intervention, our method reduces LPIPS by 57% (from 0.66 to 0.028), and for glasses reversal, it reduces $L_1$ by 73% (from 0.185 to 0.049). The reversibility analysis empirically indicates that our model can recover counterfactuals back to their original observations (Appendix F), suggesting potential for achieving counterfactual identifiability (Ribeiro et al., 2025).

*Table 15.* Soundness of CelebA-HQ counterfactual images generated by the proposed Causal-Adapter. Effectiveness is evaluated using F1-scores from pre-trained classifiers for eyeglasses ($g$) and smiling ($s$), while reversibility (Rev.) and compositional consistency (Comp.) are measured using $L_1$. Identity preservation (IDP) is assessed using LPIPS.

| | | EYEGLASSES INTERVENTION ($do(g)$) | | | | SMILING INTERVENTION ($do(s)$) | | | | NULL |
| | | EFFECTIVENESS | | REV. | IDP | EFFECTIVENESS | | REV. | IDP | COMP. |
| | METHOD | F1($s$) ↑ | F1($g$) ↑ | $L_1$ ↓ | LPIPS ↓ | F1($s$) ↑ | F1($g$) ↑ | $L_1$ ↓ | LPIPS ↓ | $L_1$ ↓ |
|---|---|---|---|---|---|---|---|---|---|---|
| 1. | VCI | 97.84 | 3.39 | - | - | 33.81 | **99.85** | - | - | - |
| 2. | HVAE | 90.05 | 65.31 | - | - | 75.33 | 95.82 | - | - | - |
| 3. | DiffCounter | **99.09** | 96.86 | 0.185 | 0.096 | **94.93** | 99.45 | 0.183 | 0.066 | 0.130 |
| 4. | Ours | 96.89 | **99.26** | **0.049** | 0.084 | 94.15 | 99.15 | **0.028** | **0.028** | 0.010 |
| 5. | Ours (DiT) | 98.19 | 97.39 | 0.086 | **0.060** | 94.71 | 99.71 | 0.089 | 0.035 | **0.001** |

**Additional counterfactual and reversal results.** Supplementary counterfactual and reversal examples generated by Causal-Adapter are provided in Figure 26–27. These visualizations demonstrate faithful interventions and strong identity preservation across diverse attribute edits.

**Stress test with compositional interventions.** Following the settings of Rasal et al. (2025), we concatenated four confounding attributes (Male, Wearing Lipstick, Bald, Wearing Hat) during training to mitigate cross-attribute spurious correlations. Based on the disentangled attributes, we perform a compositional stress test in which we select five non-overlapping attributes (smile, mouth-open, gender, glasses, hat) and progressively apply multiple interventions, i.e., $do(\text{smile})$, $do(\text{smile, mouth})$, etc. Results in Figure 28–29 show that Causal-Adapter produces plausible counterfactuals under progressively complex intervention sets while maintaining strong identity consistency.

**Generalization test (SD3 backbone).** To further evaluate the generalizability of Causal-Adapter, we apply it to Stable Diffusion 3 (SD3) (Esser et al., 2024), a diffusion backbone based on the Diffusion Transformer (DiT) architecture. Unlike SD 1.5, SD3 employs three text encoders, including two CLIP encoders and one T5 encoder. Accordingly, PAI injects attribute information into all three token-embedding streams. For a dataset with $N$ attributes, we learn $N$ attribute tokens for each encoder, resulting in $3N$ injected embeddings that guide the SD3 MMDiT backbone through the adapter.

Following the SD3 training recipe, we replace the standard diffusion loss with the flow-matching loss $\mathcal{L}_{\text{FM}}$. To maintain semantic alignment across the three heterogeneous embedding spaces, we apply CTC to the injected tokens of each encoder and optimize the following objective:

$$\mathcal{L}_{\text{total}} = \mathcal{L}_{\text{FM}} + \lambda \sum_{i=1}^{3} \mathcal{L}_{\text{CTC}_i}, \tag{9}$$

where $\mathcal{L}_{\text{CTC}_i}$ denotes the contrastive token loss for the injected embeddings of the $i$-th encoder.

During inference, we adopt FlowEdit (Kulikov et al., 2025), an editing method designed for flow-matching models. We construct the source condition by injecting the original attributes of the input image into the text embeddings, and construct

the target condition by injecting the SCM-intervened attributes. We use 50 inversion and denoising steps with a target guidance scale of 1.5 to generate counterfactual images.

As shown in Figure 30–32, Causal-Adapter successfully adapts to the SD3 backbone and produces causally faithful counterfactuals. These results demonstrate that our method is not tied to DDIM inversion or a specific diffusion architecture, but can naturally extend to different T2I generative families and editing paradigms. Although other inversion techniques, such as null-text inversion (Mokady et al., 2023), can also be integrated, they require per-sample optimization and are therefore less suitable for large-scale benchmarks.

E.4.1. QUALITATIVE RESULTS WITH THE SD1.5 BACKBONE

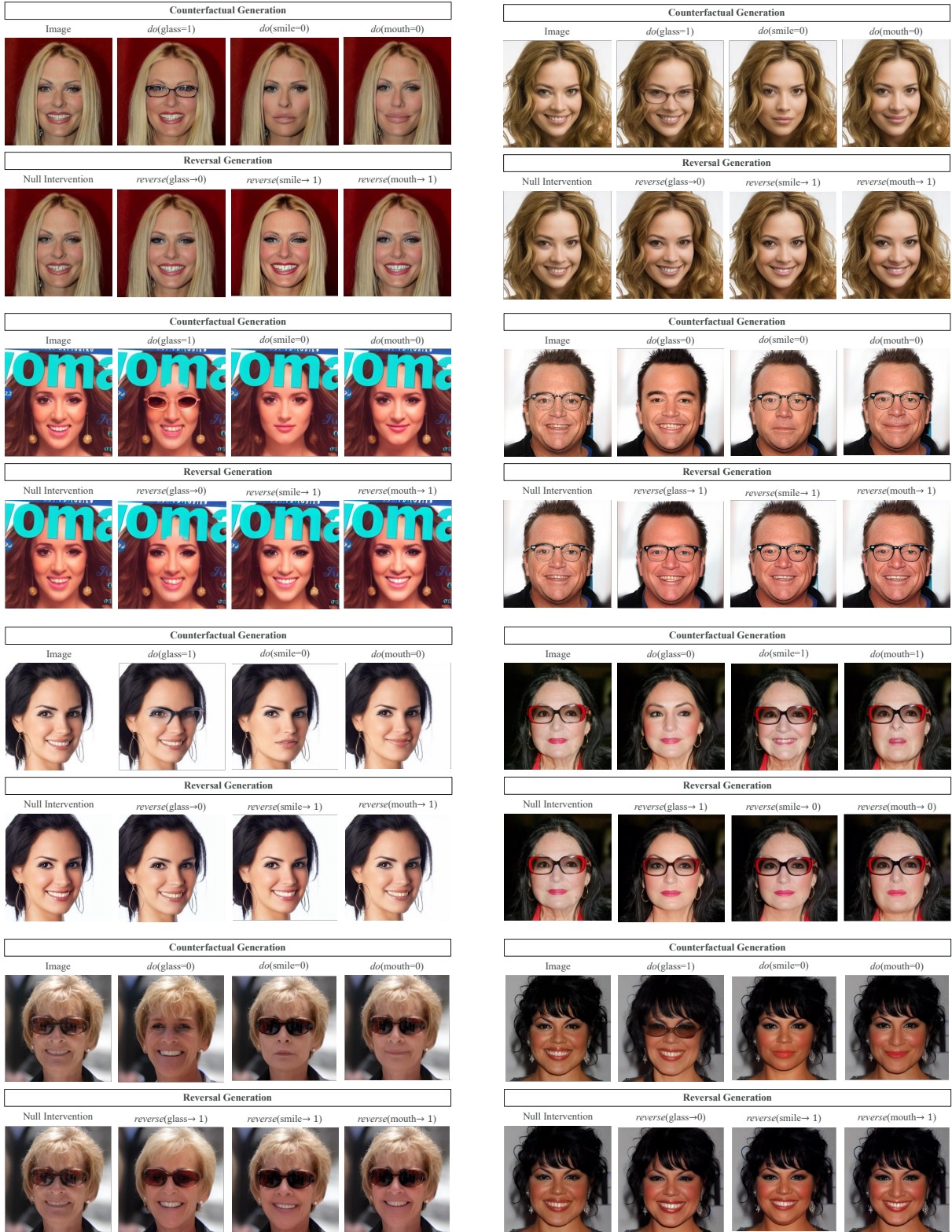

*Figure 26.* Additional counterfactual and reversal results (256×256) on CelebA-HQ.

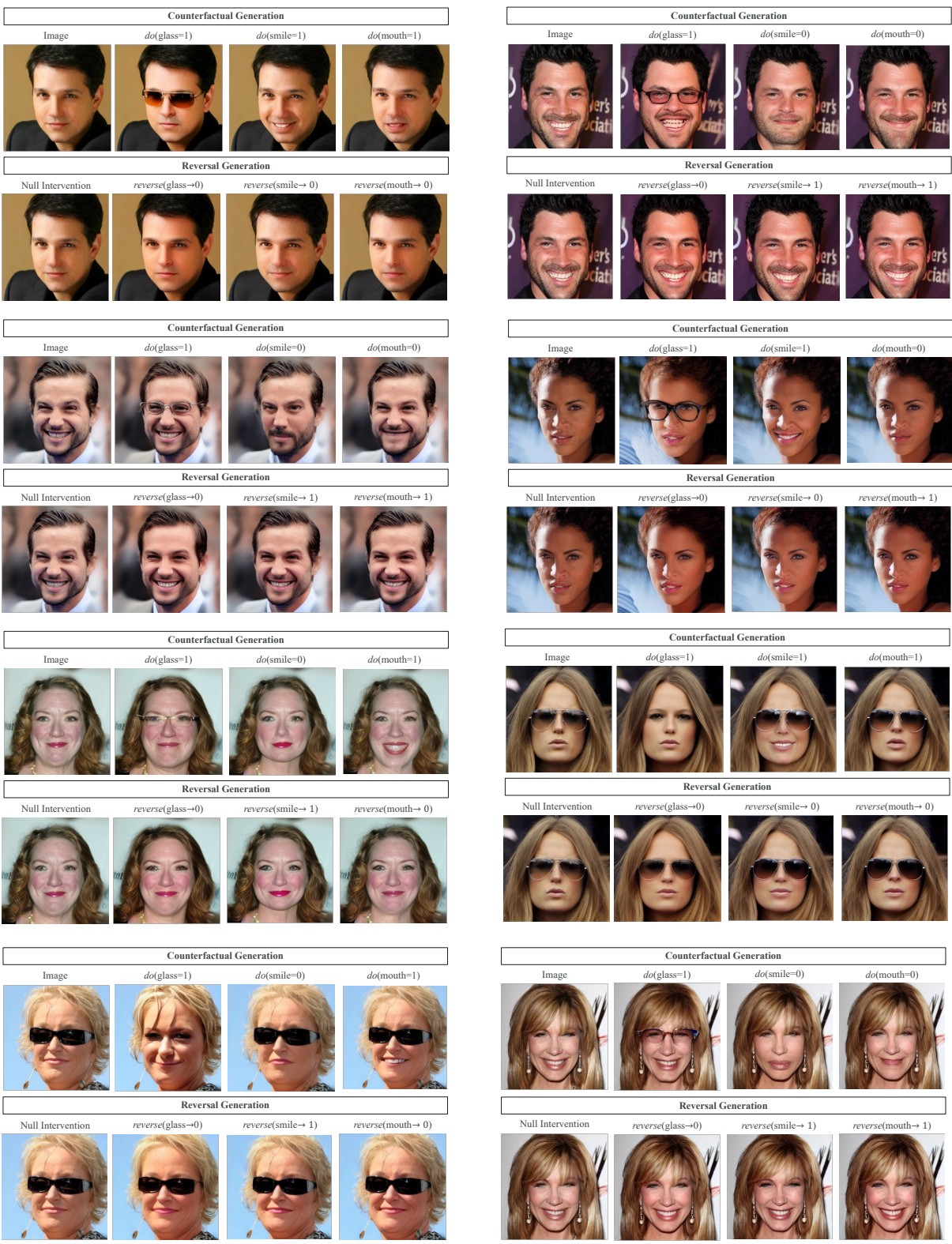

*Figure 27.* Additional counterfactual and reversal results (256×256) on CelebA-HQ.

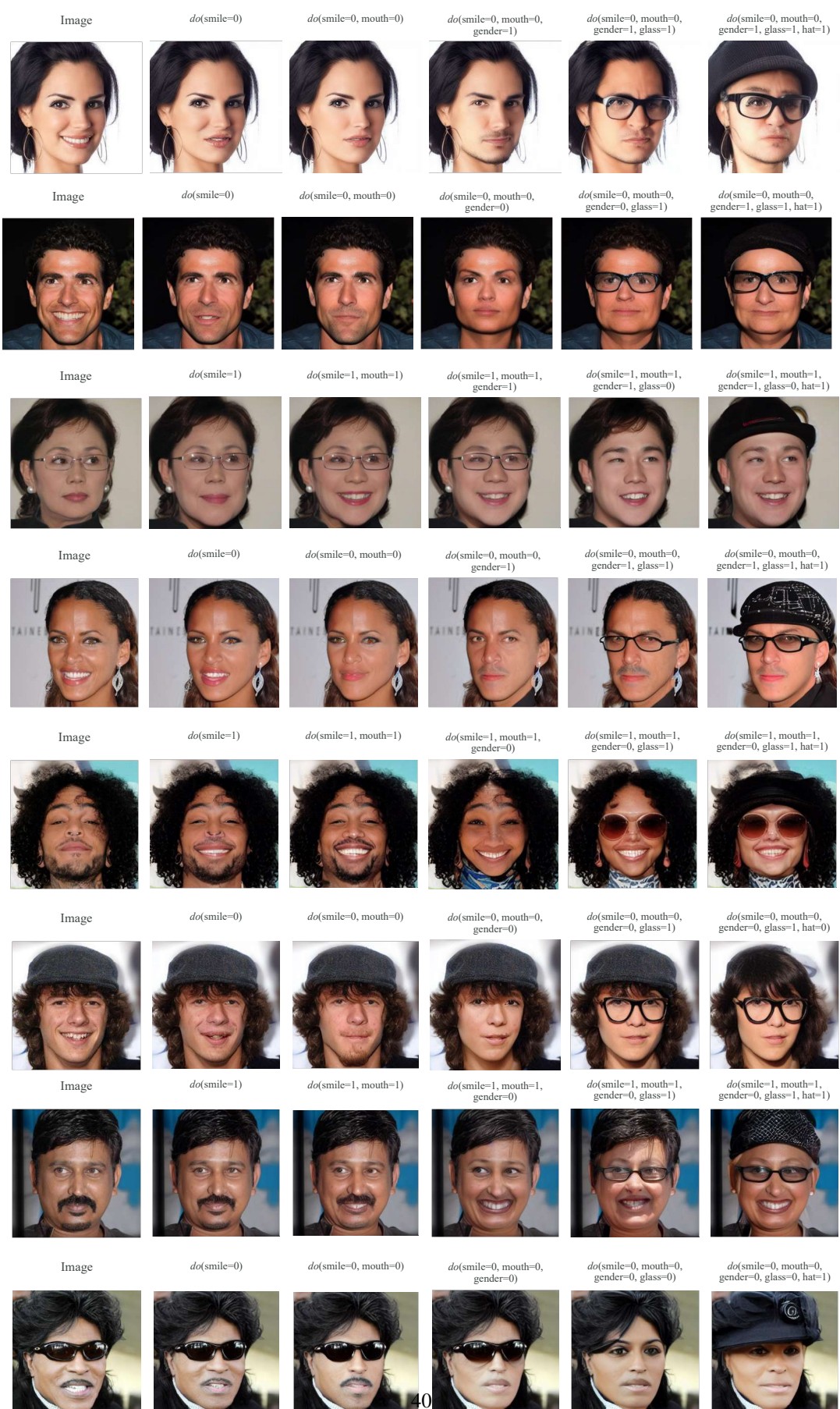

*Figure 28.* Stress test with compositional interventions: five attributes (smile, mouth-open, gender, glasses, hat) are progressively intervened. For example, the second column applies $do(\text{smile})$; the third column applies $do(\text{smile}, \text{mouth})$; subsequent columns add further interventions in sequence.

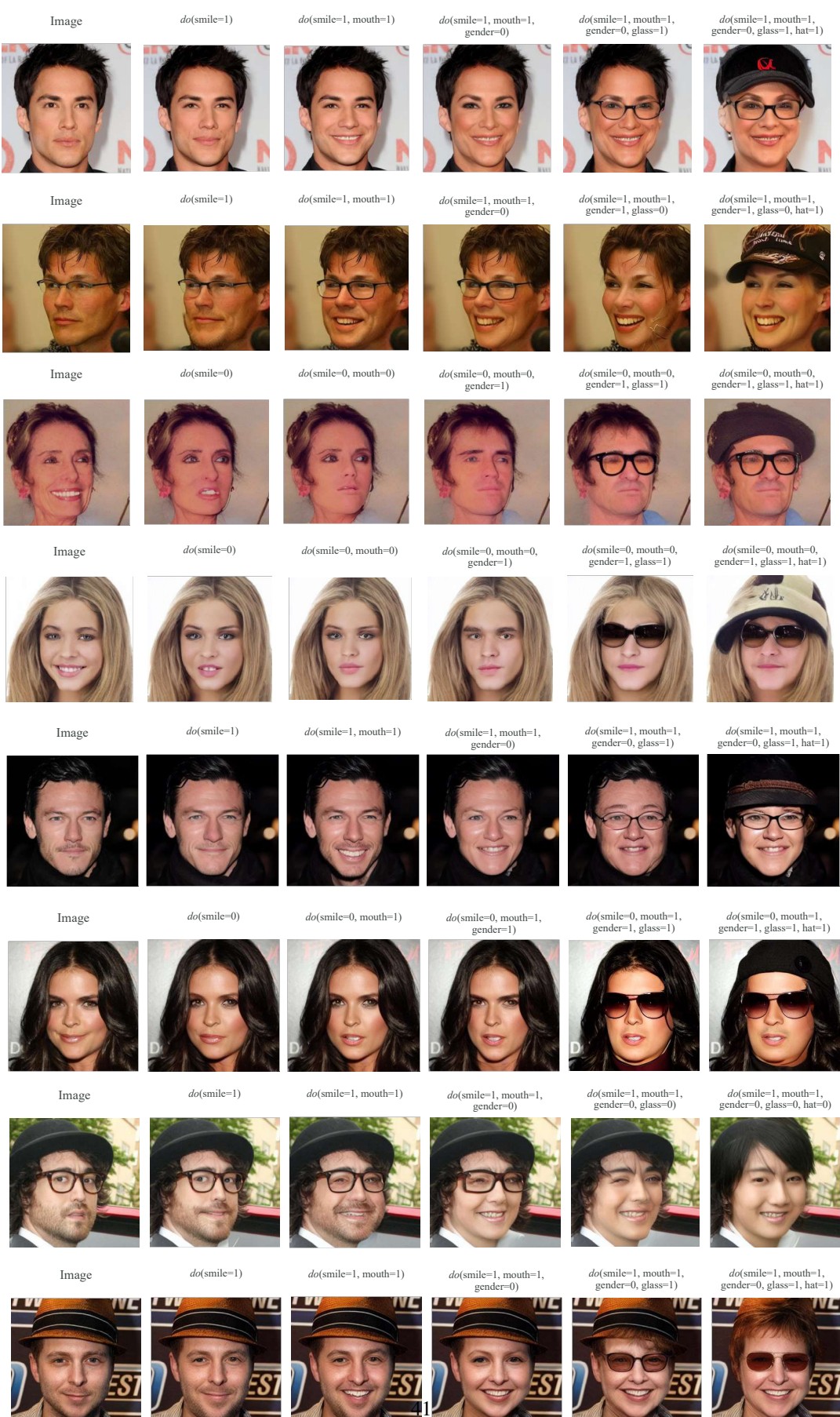

*Figure 29.* Stress test with compositional interventions: five attributes (smile, mouth-open, gender, glasses, hat) are progressively intervened. For example, the second column applies $do(\text{smile})$; the third column applies $do(\text{smile}, \text{mouth})$; subsequent columns add further interventions in sequence.

E.4.2. QUALITATIVE RESULTS WITH THE SD3 BACKBONE

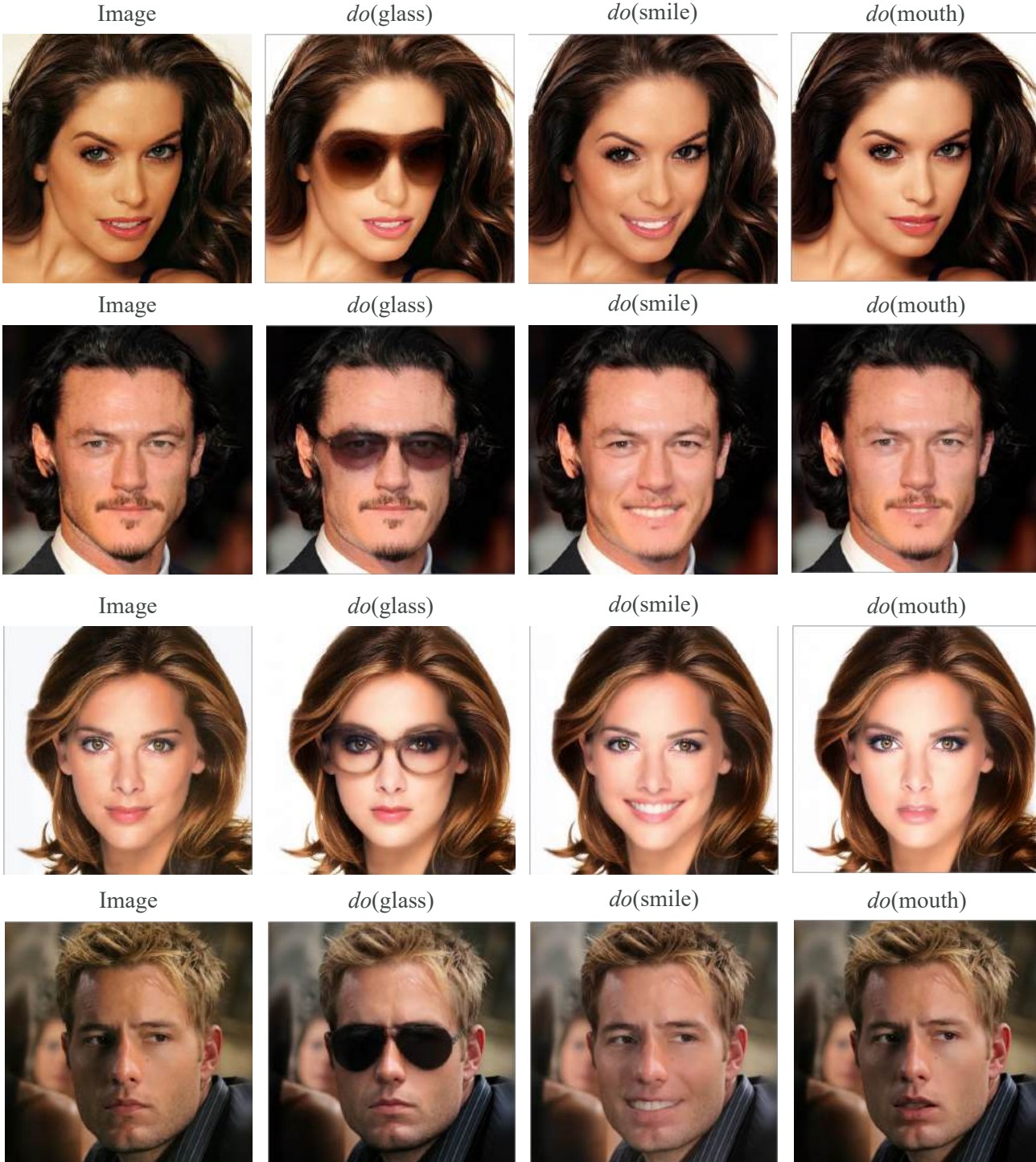

*Figure 30.* Additional counterfactuals (512×512) generated by using SD3 Backbone.

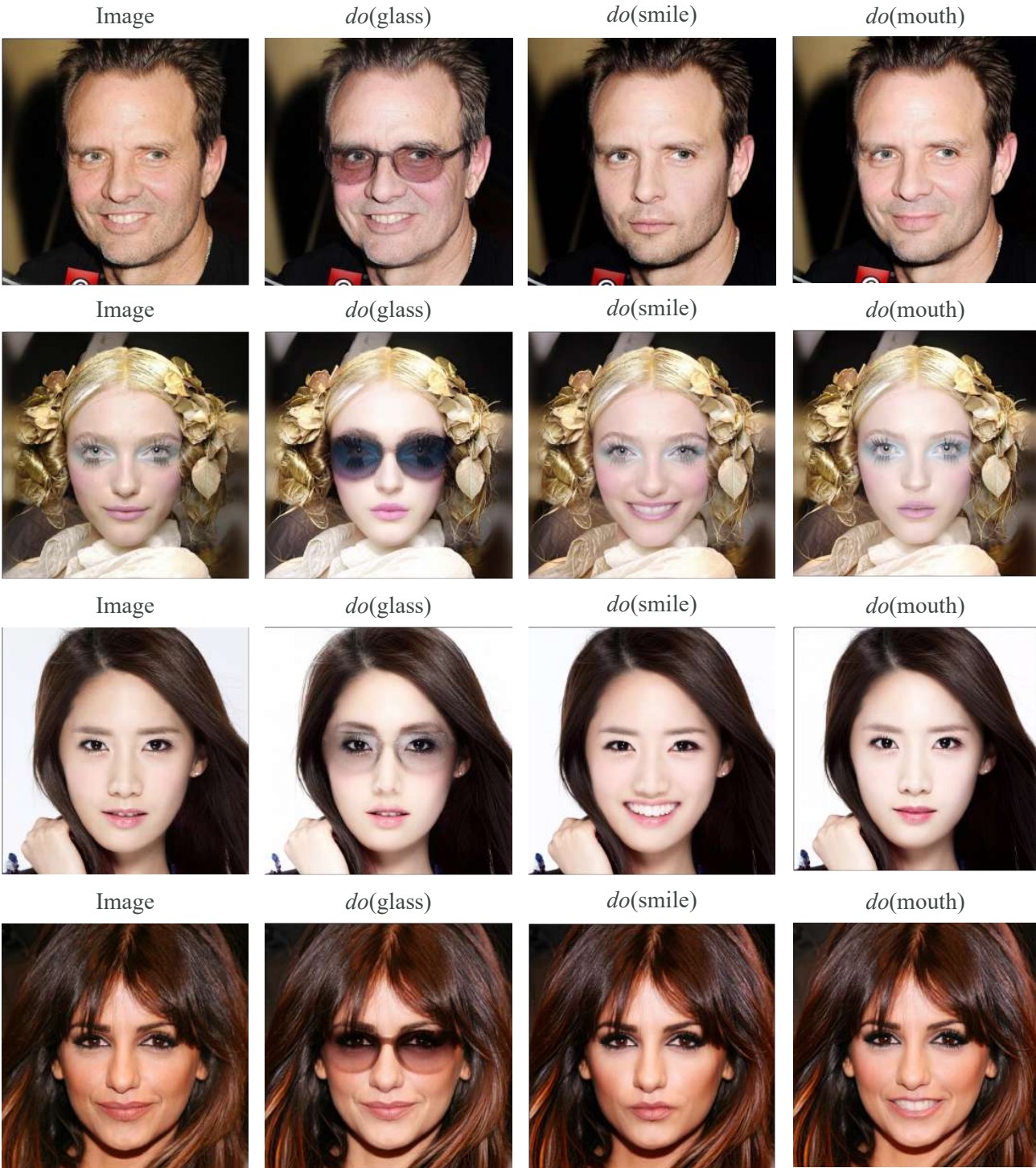

*Figure 31.* Additional counterfactuals (512×512) generated by using SD3 Backbone.

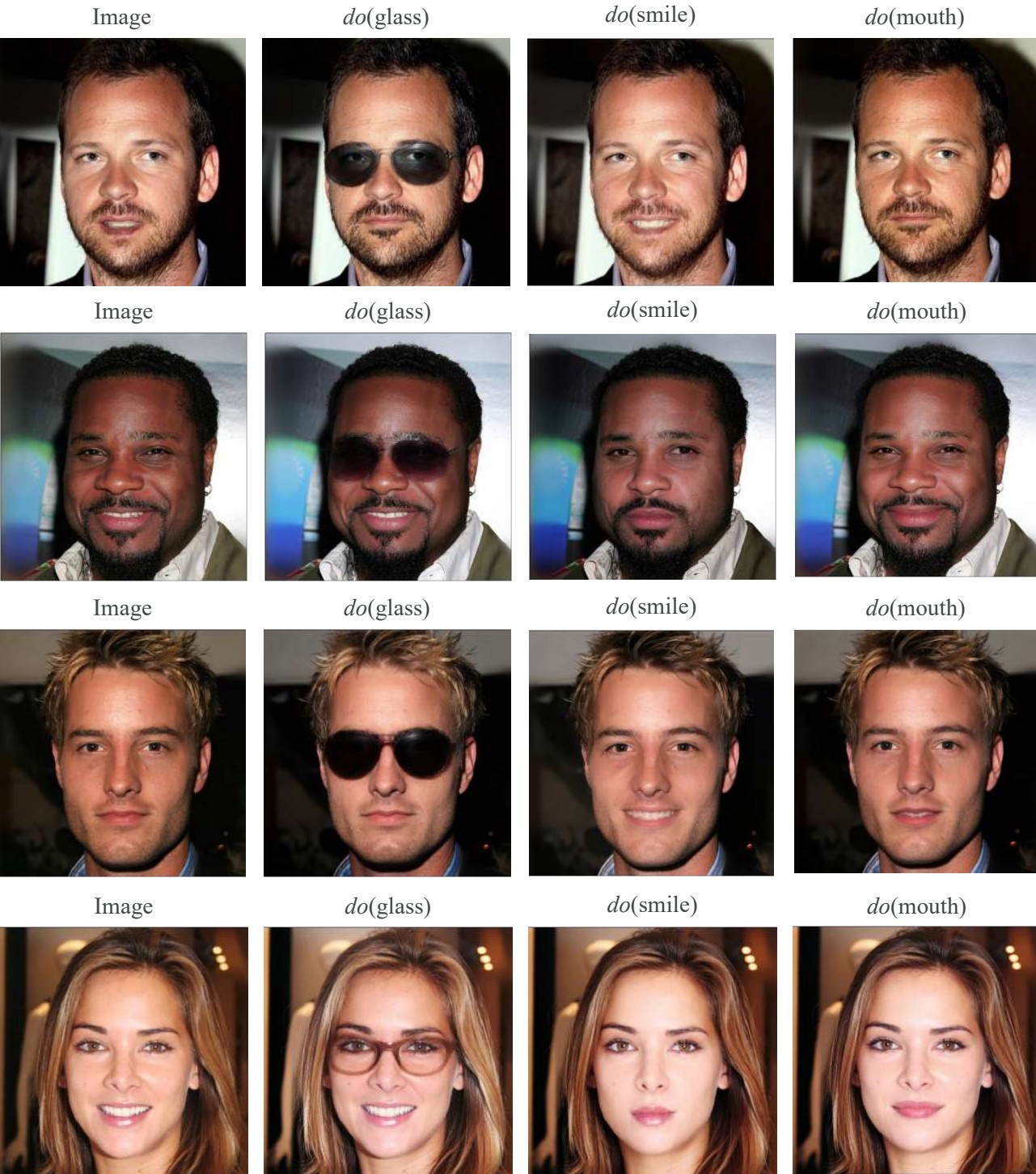

*Figure 32.* Additional counterfactuals (512×512) generated by using SD3 Backbone.

### E.5. Attention Maps

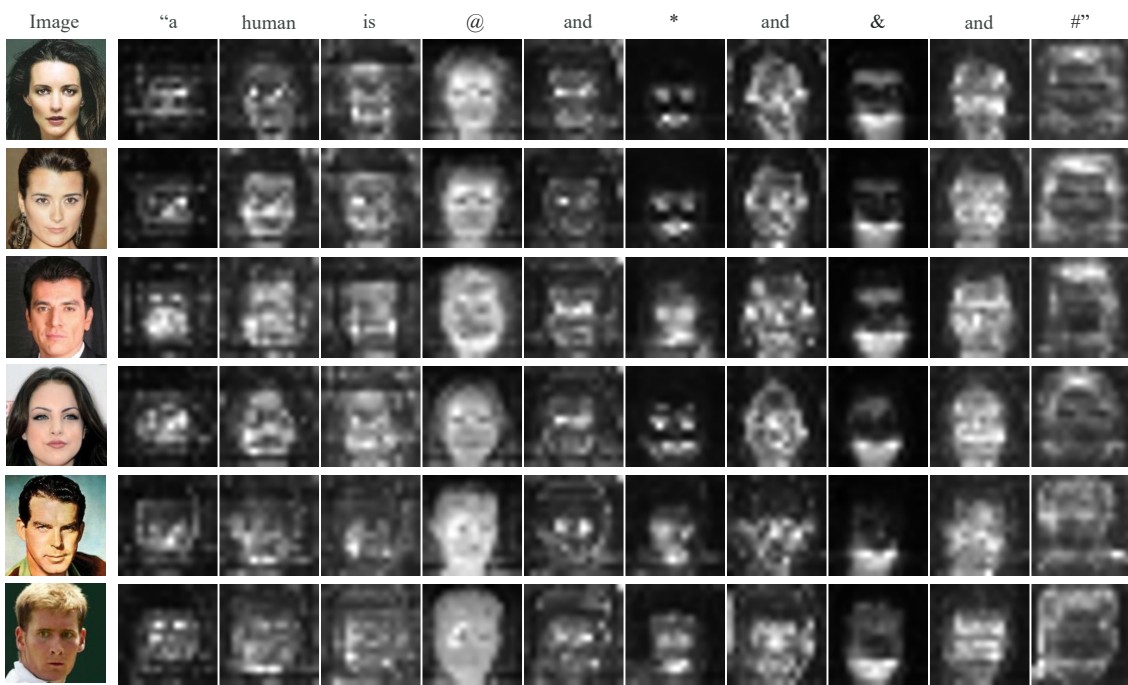

*Figure 33.* Average cross-attention maps from Causal-Adapter on CelebA dataset. Token denote attributes: "@" for age, "*" for gender, "&" for beard, and "#" for bald.

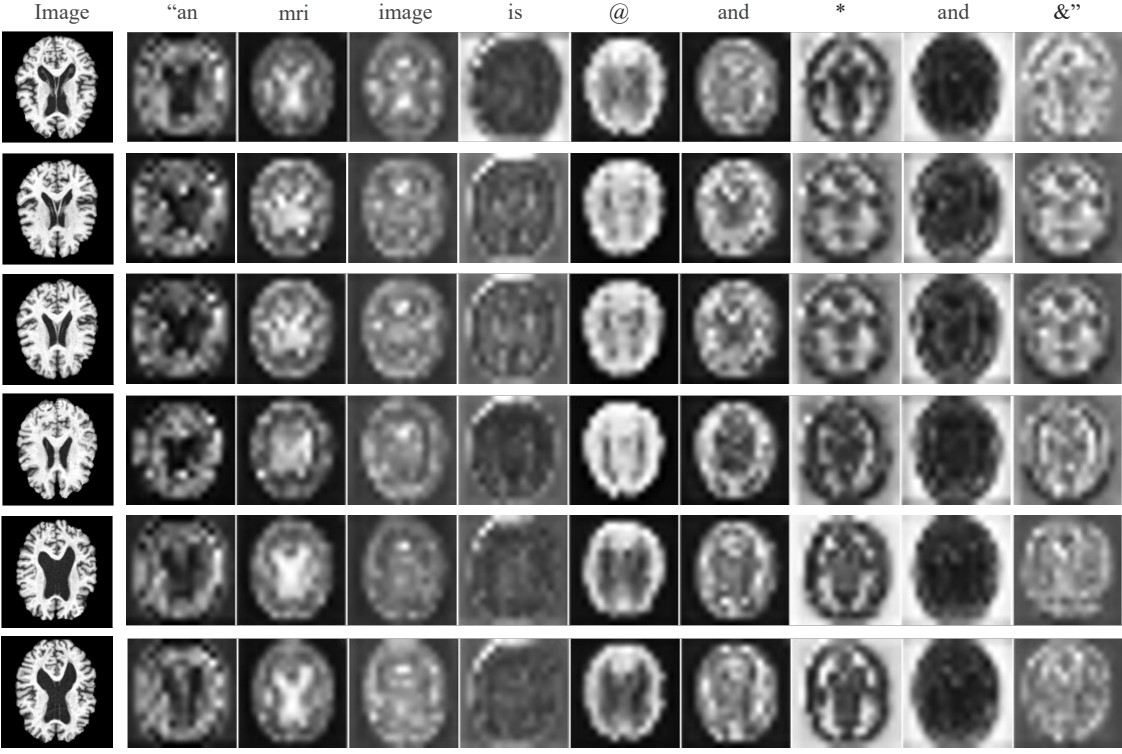

*Figure 34.* Average cross-attention maps from Causal-Adapter on ADNI dataset. Token denote attributes: "@" for brain volume, "*" for ventricular volume, "&" for slice.

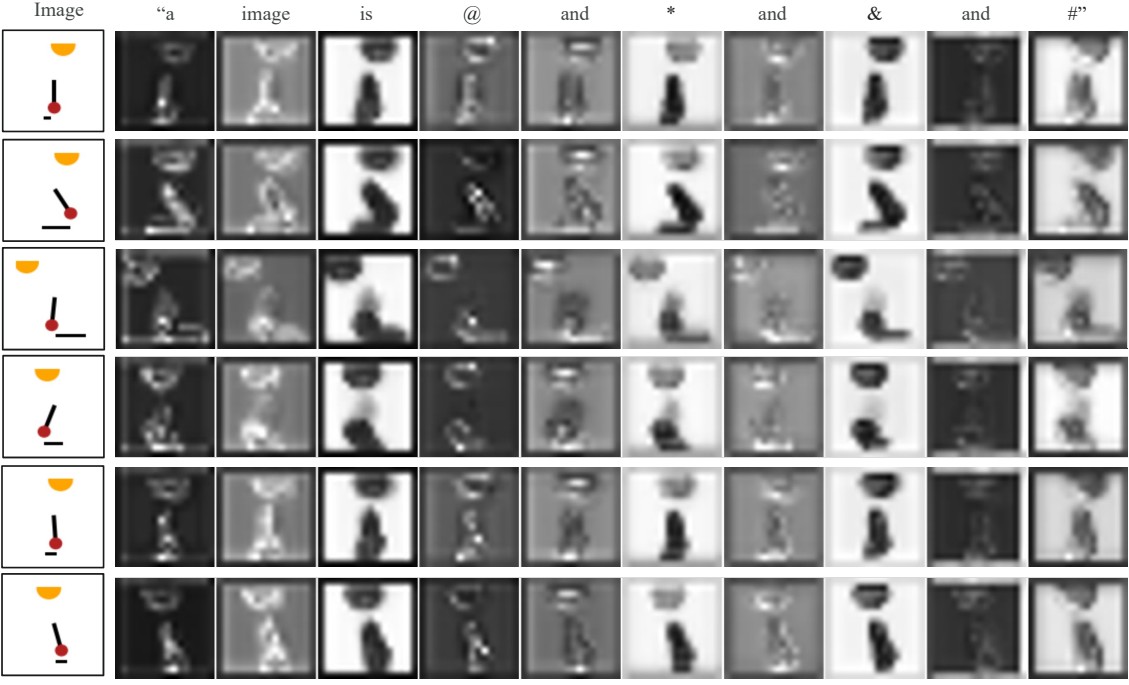

*Figure 35.* Average cross-attention maps from Causal-Adapter on Pendulum dataset. Token denote attributes: "@" for pendulum, "*" for light, "&" for shadow length, and "#" for shadow position.

### E.6. Comparison with Commercial Foundation Models

To further assess the practical effectiveness of Causal-Adapter, we compare it with recent commercial foundation editing models, including Nano Banana 2, Qwen-Image-Edit, and Flux.1-Kontext-Dev. We provide qualitative comparisons on Pendulum, ADNI, and CelebA in Figures 36–38. Since these foundation models do not expose explicit attribute-level causal controls, editing is performed by updating their text prompts. In contrast, Causal-Adapter performs counterfactual editing by intervening on attributes through the SCM and injecting the resulting causal conditions into the text-embedding space.

We also provide quantitative comparisons on a CelebA subset. Tables 16 and 17 report classifier-based accuracy for the target and related attributes, together with LPIPS for visual similarity. Compared with large-scale commercial editing models trained with complex pipelines, Causal-Adapter remains simple and efficient while achieving competitive performance under interventions on age, gender, and beard. These results are consistent with our main findings: recent foundation editing models can produce visually plausible edits, but they still struggle with fine-grained causal interventions, especially for continuous or causally entangled attributes. They often modify visually salient regions rather than faithfully following the intended causal intervention, which limits their suitability for counterfactual image generation.

*Table 16.* Quantitative comparison with commercial foundation editing models on CelebA under age and gender interventions. We report classifier accuracy for Age, Gender, Beard, and Bald, together with LPIPS for visual similarity. Higher accuracy is better, while lower LPIPS is better.

| | Method | Age Acc. ↑ | | Gender Acc. ↑ | | Beard Acc. ↑ | | Bald Acc. ↑ | | LPIPS ↓ | |
|---|---|---|---|---|---|---|---|---|---|---|---|
| | | $do(a)$ | $do(g)$ | $do(a)$ | $do(g)$ | $do(a)$ | $do(g)$ | $do(a)$ | $do(g)$ | $do(a)$ | $do(g)$ |
| 1. | Ours | **72** | 90 | **100** | **100** | **100** | 96 | 90 | 92 | 0.0725 | 0.0526 |
| 2. | Qwen-Image-Edit | 62 | 86 | 84 | 90 | 96 | 96 | 86 | **98** | 0.1496 | 0.1579 |
| 3. | Flux.1-Kontext-Dev | 34 | 84 | 98 | 46 | 96 | 88 | 92 | 90 | **0.0560** | **0.0480** |
| 4. | Nano Banana 2 | 44 | **94** | 98 | 68 | 94 | **100** | 92 | 98 | 0.0750 | 0.0820 |

*Table 17.* Quantitative comparison with commercial foundation editing models on CelebA under beard and bald interventions. We report classifier accuracy for Age, Gender, Beard, and Bald, together with LPIPS for visual similarity. Higher accuracy is better, while lower LPIPS is better.

| | Method | Age Acc. ↑ | | Gender Acc. ↑ | | Beard Acc. ↑ | | Bald Acc. ↑ | | LPIPS ↓ | |
|---|---|---|---|---|---|---|---|---|---|---|---|
| | | $do(br)$ | $do(bl)$ | $do(br)$ | $do(bl)$ | $do(br)$ | $do(bl)$ | $do(br)$ | $do(bl)$ | $do(br)$ | $do(bl)$ |
| 1. | Ours | **98** | **70** | 48 | **86** | 80 | 92 | **100** | 40 | 0.0282 | 0.1087 |
| 2. | Qwen-Image-Edit | 84 | 56 | 36 | 82 | **92** | 96 | 98 | **94** | 0.1326 | 0.1892 |
| 3. | Flux.1-Kontext-Dev | 78 | 60 | 44 | 80 | 88 | 92 | 98 | 76 | 0.0665 | **0.0976** |
| 4. | Nano Banana 2 | 92 | 66 | **50** | **86** | 86 | 96 | **100** | 94 | **0.0215** | 0.1200 |

```
Prompt: Edit the reference image of a simple causal pendulum scene.

Scene semantics:
– orange top object: light source
– black and red middle object: pendulum
– black bottom bar: shadow
Current state:
– pendulum angle: –2 degrees
– light position: 0.2
Intervention:
– change only the pendulum angle to [TARGET_ANGLE] degrees
– keep the light position fixed at [TARGET_LIGHT_POSITION]
Causal constraint:
– the shadow position and shadow length must be updated only as a consequence of the pendulum angle change

Requirements:
1. Preserve the same scene, colors, object identities, and simple 2D style.
2. Rotate the pendulum to the target angle when [TARGET_ANGLE] changed.
3. Move the light source horizontally only when [TARGET_LIGHT_POSITION] changed.
4. Adjust the shadow position and length to remain physically and causally consistent.
5. Do not change any other elements unnecessarily.
6. Do not add text, labels, or extra objects.
```

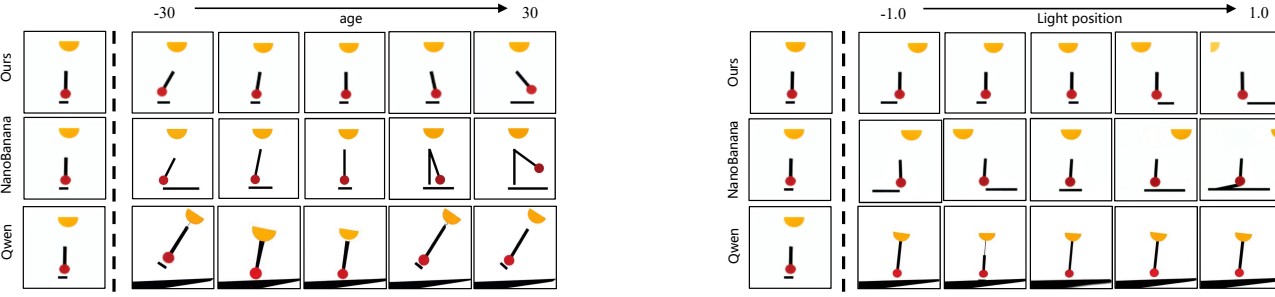

*Figure 36.* Qualitative comparison with commercial foundation editing models on the Pendulum dataset.

```
Prompt: The reference image is an axial brain MRI of an Alzheimer's disease patient.
Current attributes:
– age: 89
– ventricular volume: 0.51
Generate a realistic counterfactual MRI of the same subject with:
– target age: [TARGET_AGE]
– target ventricular volume: [TARGET_VENT_VOLUME]
– Preserve the same identity, slice position, orientation, field of view, grayscale MRI appearance, and overall anatomical
  layout.
Causal constraint:
– Age → brain volume → ventricular volume
Only introduce medically plausible structural changes related to the transition from the current state to the target
state.
Keep non–intervened regions unchanged as much as possible.
Do not add text, labels, lesions, tumors, or unrealistic artifacts.
Output a single high–quality grayscale MRI image.
```

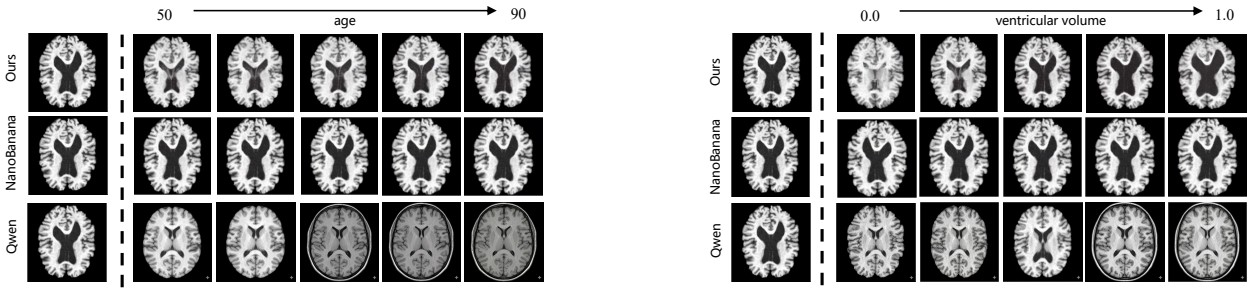

*Figure 37.* Qualitative comparison with commercial foundation editing models on ADNI.

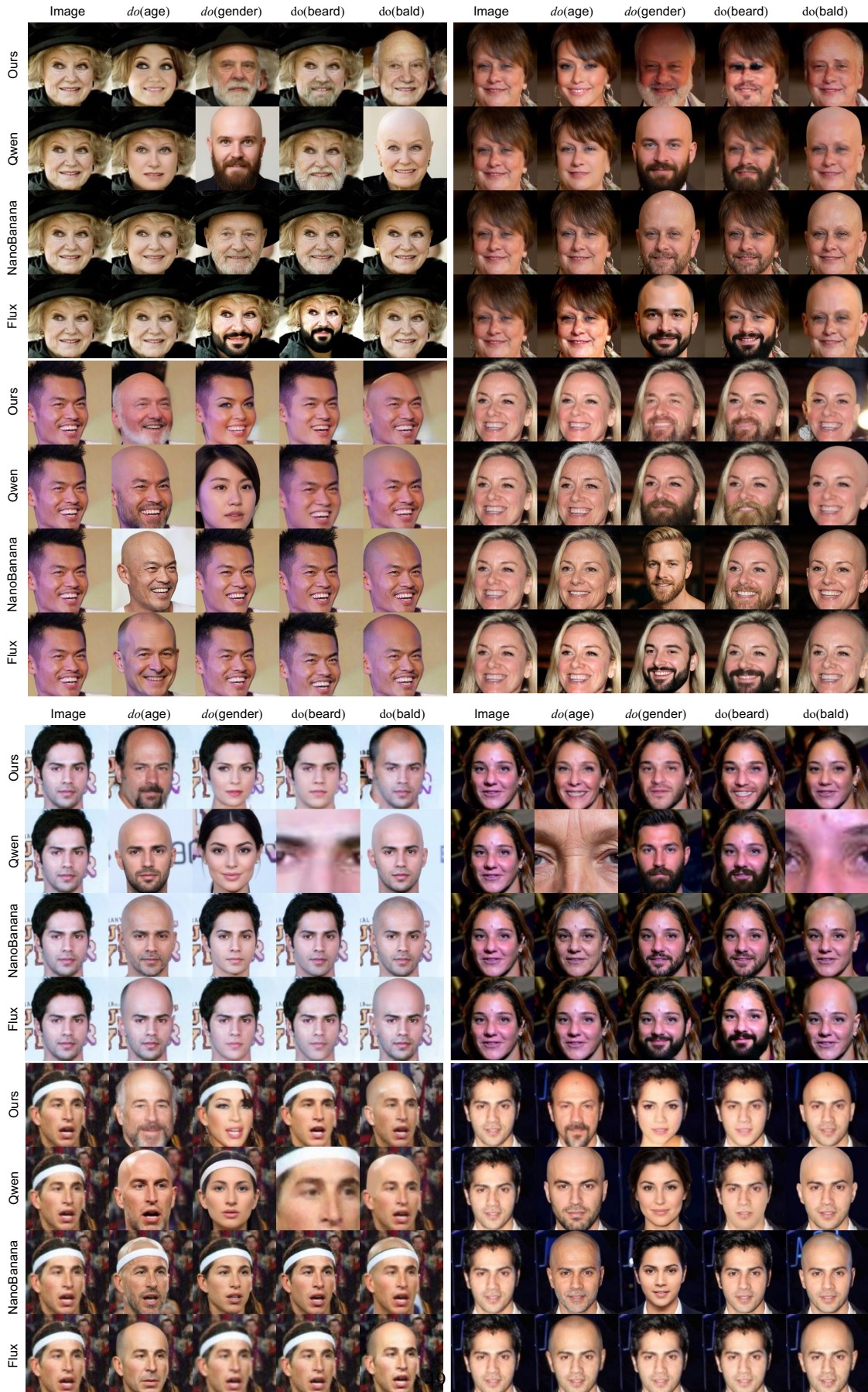

*Figure 38.* Qualitative comparison with commercial foundation editing models on CelebA.

### E.7. Generalization Beyond Aligned Images

To further examine whether Causal-Adapter can generalize beyond tightly aligned datasets such as CelebA, ADNI, and Pendulum, we evaluate it on full-body human images with larger variations in pose, background, scale, and composition. Specifically, we use a public human-body image dataset[1], randomly select 100 samples, and manually annotate three attributes: age, gender, and beard. Since constructing a comprehensive causal graph for open-domain human images is challenging, we perform disentangled editing without using an SCM in this experiment. We compare Causal-Adapter with Qwen-Image-Edit, Nano Banana 2, and Flux.1-Kontext-Dev.

As shown in Figure 39, commercial foundation editing models can generate visually plausible results, but they often struggle to perform precise attribute-level interventions while preserving the overall image structure. For example, Qwen-Image-Edit tends to over-edit the image and sometimes introduces partial cropping artifacts, while Flux.1-Kontext-Dev often fails to reliably edit age and gender. In contrast, Causal-Adapter produces clear changes under age and gender interventions while better preserving pose, clothing layout, and background. These results suggest that the proposed framework retains a degree of disentangled editing ability beyond highly aligned face datasets.

This experiment is not intended to claim fully open-domain causal generation, as reliable causal editing in general scenes would require richer attribute annotations and more comprehensive causal knowledge. Nevertheless, the results indicate that Causal-Adapter can potentially scale to more complex domains when sufficiently labeled attributes and training samples are available. They also support our main claim that the framework can still function as a disentangled editing method even without an explicit causal graph, while the SCM can be incorporated when reliable causal structure is available.

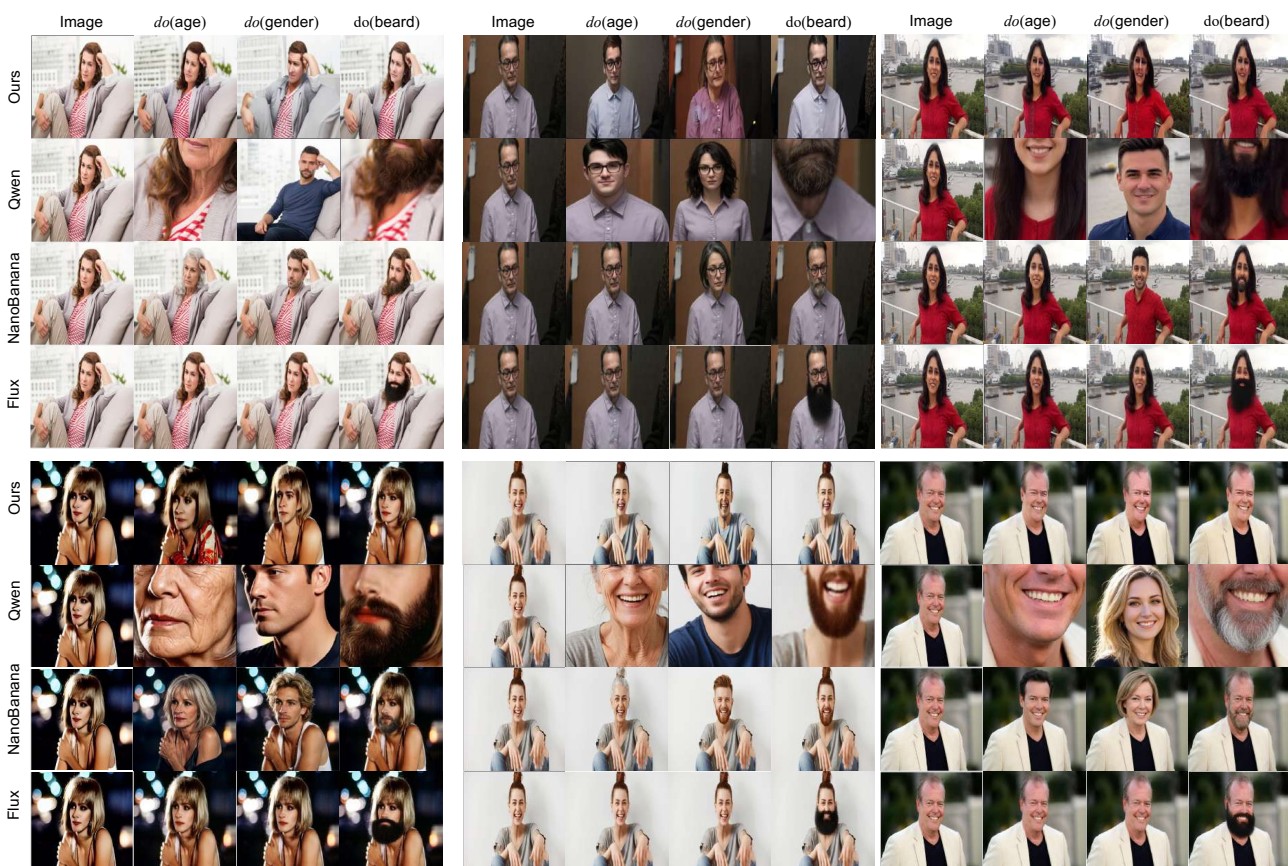

*Figure 39.* Qualitative comparison with commercial foundation editing models on full-body human images. Causal-Adapter performs disentangled editing without using an explicit causal graph in this experiment.

---

[1]https://www.kaggle.com/datasets/snmahsa/human-images-dataset-men-and-women/data

# F. Counterfactual Identifiability [Back to Contents]

The abduction step in counterfactual generation implicitly assumes that the frozen diffusion model provides a counterfactually valid inverse mapping that DDIM inversion can approximately recover the exogenous noise consistent with the true data-generating process. This relates to the broader challenge of counterfactual identifiability in high-dimensional generative models (Ribeiro et al., 2025; Komanduri et al., 2024a; Nasr-Esfahany et al., 2023), where recovering latent exogenous variables from observations is generally non-trivial and often non-identifiable without additional assumptions.

To empirically assess identifiability, we perform a reversibility analysis, following the principle that counterfactual outputs should be reconstructable from the observed image distribution. In our CelebA-HQ experiments, we compare Causal-Adapter with DiffCounter and observe substantially stronger reversibility under our approach. Representative examples for three interventions (glasses, smile, mouth) are shown in Figure 26–27. These results suggest that our adapter improves the model's ability to produce counterfactuals that remain consistent with the underlying observational manifold.

However, we note that strong empirical recovery does not imply a formal identifiability guarantee. DDIM inversion is approximate, and imperfect reconstruction may cause information loss, especially under complex or compound interventions. Thus, even our Causal-Adapter "tames" causal priors into the frozen T2I backbone and yields practically robust counterfactual behavior, exact theoretical identifiability remains an open challenge.

Achieving formal counterfactual identifiability would require improved inverse operators or diffusion models explicitly designed to recover exogenous noise, may potentially leveraging recent progress in flow matching. We regard this as an important direction for future work, complementary to our empirical findings.

## G. Relaxing the Assumption of a Known Causal Graph [Back to Contents]

Existing counterfactual generation frameworks typically assume that a pre-defined causal graph is provided as part of the input (Pearl, 2009; Pawlowski et al., 2020; Yang et al., 2021; De Sousa Ribeiro et al., 2023; Wu et al., 2025; Komanduri et al., 2024a;c; Rasal et al., 2025; Xia et al., 2025). This assumption enables active interventions and facilitates comparable evaluation across methods, but it can be restrictive in real-world scenarios where the causal graph may be misspecified, partially known, or entirely unavailable. To examine this issue, we conduct two complementary studies: (1) a causal discovery study that explores whether Causal-Adapter can learn the causal structure from data, and (2) a graph ablation study that evaluates the effect of using no graph, an incorrect graph, and the correct graph.

**Causal Discovery as a Mitigation Study.** Causal-Adapter is modular and can be extended with minimal modification to jointly learn the causal graph. Instead of fixing the adjacency matrix $A$, we treat $A$ as a learnable parameter initialized to zeros. Following differentiable DAG-based methods such as NOTEARS (Zheng et al., 2018) and DAGMA (Bello et al., 2022), we impose an acyclicity constraint, such as a log-determinant penalty, directly on $A$. This requires no additional networks or architectural changes, but only introduces an extra structural loss term.

As shown in Figure 40, this extension enables Causal-Adapter to perform causal structure learning and achieve competitive or superior performance compared with state-of-the-art differentiable causal discovery methods, including SDCD (Nazaret et al., 2024). Across three benchmark settings, our method recovers more true edges than competing baselines. The learning dynamics of $A$ are further visualized in Figure 41, showing stable convergence toward the ground-truth graph and demonstrating interpretable behavior. Nevertheless, consistent with prior studies (Nazaret et al., 2024; Olko et al., 2025), recovering the full causal graph from purely observational data, such as CelebA or ADNI, remains fundamentally challenging due to the lack of interventional signals. These results suggest that Causal-Adapter has the potential to integrate causal discovery and counterfactual generation within a single, simple, and efficient framework.

**Ablation on Causal Graph Specification.** We further evaluate the robustness of Causal-Adapter under different causal graph specifications. Specifically, we consider three settings: (1) without SCM, (2) with an incorrect causal graph, denoted as SCM1, and (3) with the correct causal graph, denoted as SCM2. This study examines whether the framework can still perform disentangled editing without an explicit graph, and whether the correct graph improves graph-faithful counterfactual generation.

For the Pendulum dataset, qualitative results are shown in Figure 42, and quantitative results are reported in Table 18. Without SCM, interventions can still modify shadow-related variables because pendulum, light, and shadow attributes are strongly correlated in the training data. However, these changes are primarily correlation-driven rather than graph-faithful: intervening on one factor may spuriously affect another non-intervened factor, such as changing the light while also moving the pendulum. With SCM1, the incorrectly specified causal edge leads to an inconsistent propagation pattern and produces a different shadow evolution behavior. In contrast, with the correct graph SCM2, the intervention effects propagate more consistently with the underlying physical mechanisms.

*Table 18.* Ablation on causal graph specification for the Pendulum dataset. We report MAE under interventions on pendulum $do(p)$ and light $do(l)$. Lower values indicate better intervention effectiveness.

| | Method | Pendulum ($p$) MAE ↓ | | Light ($l$) MAE ↓ | | Shadow Length ($sl$) MAE ↓ | | Shadow Position ($sp$) MAE ↓ | |
|---|---|---|---|---|---|---|---|---|---|
| | | $do(p)$ | $do(l)$ | $do(p)$ | $do(l)$ | $do(p)$ | $do(l)$ | $do(p)$ | $do(l)$ |
| 1. | w/o SCM | 0.159 | 0.183 | 0.060 | 0.173 | 0.143 | 0.235 | 0.086 | 0.155 |
| 2. | w/ SCM1 | **0.011** | 0.040 | 0.048 | 0.062 | 0.039 | 0.104 | 0.031 | **0.033** |
| 3. | w/ SCM2 | 0.014 | **0.035** | **0.045** | **0.041** | **0.028** | **0.051** | **0.030** | 0.033 |

We also perform a graph ablation study on ADNI. The qualitative results are shown in Figure 43, and the quantitative results are reported in Table 19. In the non-causal setting, intervening on brain volume rarely affects the ventricular region. With SCM1, where the edge between brain volume and ventricular volume is incorrectly specified, interventions on parent factors no longer clearly influence ventricular volume and contour. By contrast, the correct graph better preserves the expected propagation pattern between brain and ventricular structures.

Overall, these studies show that our design deliberately separates SCM modeling from conditional attribute learning and disentanglement. This allows the framework to adapt to different causal graphs at inference time by replacing the SCM,

*Table 19.* Ablation on causal graph specification for ADNI. We report MAE for brain and ventricular volume under interventions on brain volume $do(b)$, ventricular volume $do(v)$, and sex $do(s)$. Lower values indicate better intervention effectiveness.

| | Method | **Brain ($b$) MAE** ↓ | | | **Ventricle ($v$) MAE** ↓ | | |
|---|---|---|---|---|---|---|---|
| | | $do(b)$ | $do(v)$ | $do(s)$ | $do(b)$ | $do(v)$ | $do(s)$ |
| 1. | w/o SCM | 0.142 | **0.110** | 0.110 | **0.028** | **0.030** | 0.030 |
| 2. | w/ SCM1 | 0.145 | 0.144 | **0.030** | 0.032 | 0.032 | **0.029** |
| 3. | w/ SCM2 | **0.090** | **0.110** | 0.110 | 0.030 | **0.030** | 0.030 |

without retraining the diffusion backbone. When no causal graph is used, Causal-Adapter can still function as a disentangled editing method. When a reliable causal graph is available, the generated counterfactuals better match the underlying data-generating structure and achieve stronger quantitative performance.

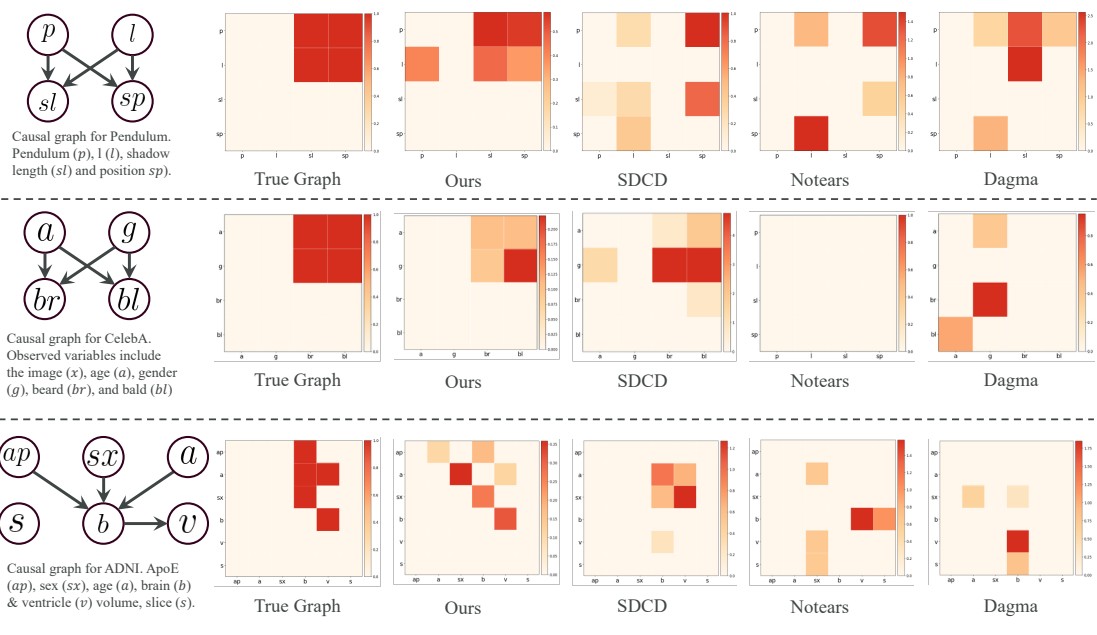

*Figure 40.* Causal discovery performance of Causal-Adapter compared with SDCD, NOTEARS, and DAGMA across three benchmarks. Using the predefined graph as ground truth, our method recovers more true edges than competing methods, demonstrating the potential to unify causal discovery and counterfactual generation within a single framework.

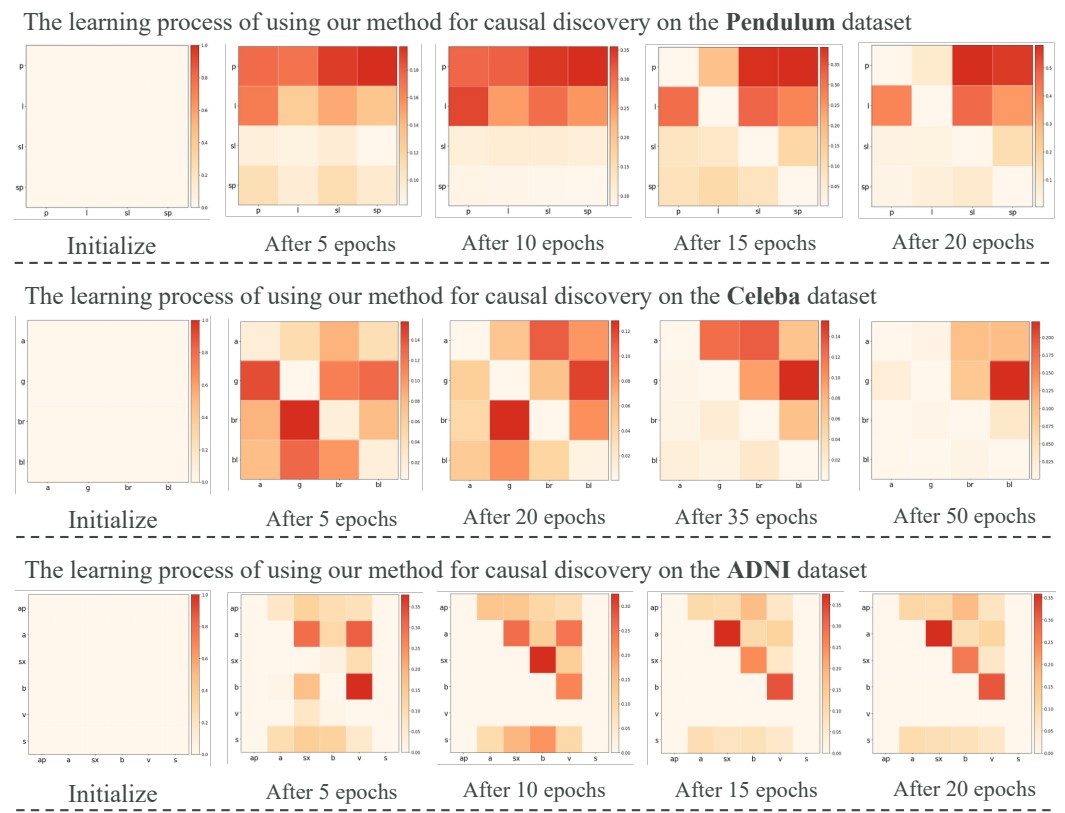

*Figure 41.* Learning trajectory of the adjacency matrix $A$. As training progresses, the learned graph progressively converges toward the ground-truth structure. A fixed threshold of 0.1 is applied across benchmarks for fair comparison.

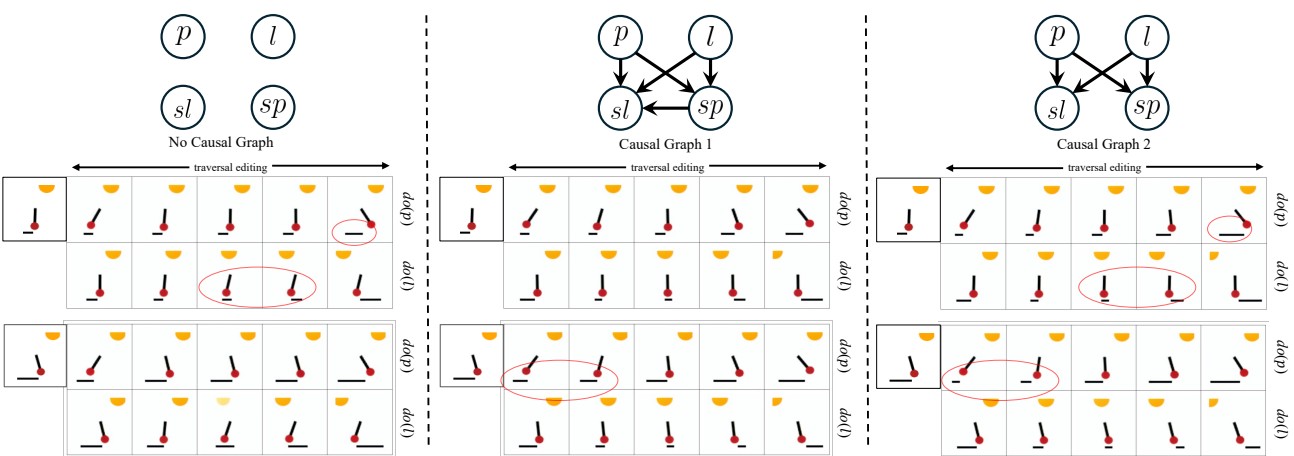

*Figure 42.* Pendulum counterfactuals under three graph settings: no causal graph, an incorrect causal graph (SCM1), and the correct causal graph (SCM2).

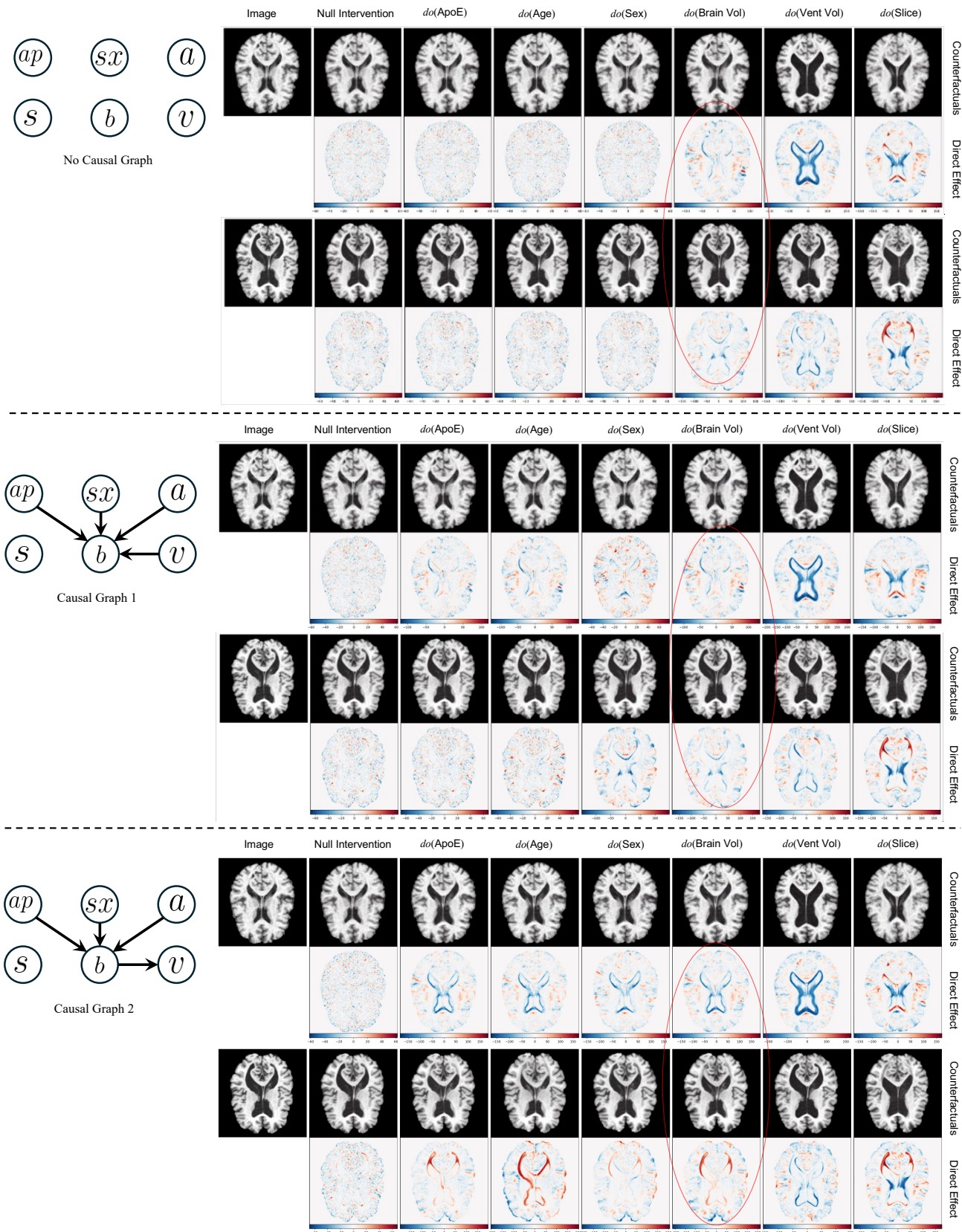

*Figure 43.* ADNI counterfactuals under three graph settings: no causal graph, an incorrect causal graph (SCM1), and the correct causal graph (SCM2).

