# OpenReview forum: "Causal-Adapter: Taming Text-to-Image Diffusion for Faithful Counterfactual Generation"
_ICML.cc/2026/Conference — ICML 2026 regular_

### Official Review · Reviewer_iauX · 2026-03-10

**Soundness:** 3
**Presentation:** 4
**Significance:** 4
**Originality:** 3
**Overall Recommendation:** 4
**Confidence:** 4

**Summary:**

This paper introduces Causal-Adapter, a lightweight and modular adapter that equips a frozen text-to-image diffusion model (e.g., Stable Diffusion v1.5) with explicit SCM capability for counterfactual image generation. The method models causal mechanisms over semantic attributes using a nonlinear additive noise model, injects these attributes into learnable textual token embeddings via PAI, and enforces disentanglement with a CTC loss. At inference, it follows the classic abduction–action–prediction procedure with DDIM inversion and optional attention guidance. Experiments on Pendulum, CelebA, CelebA-HQ, and ADNI demonstrate state-of-the-art performance in effectiveness (up to 91% MAE reduction), realism (up to 87% FID reduction), composition, and minimality, while preserving identity and enabling continuous attribute control. The approach is training-efficient since only the adapter and token embeddings are optimized, and compatible with different backbones (including DiT).

**Compliance With Llm Reviewing Policy:**

Affirmed.

**Ethical Review Concerns:**

Recommendation: Weak accept

The authors’ rebuttal successfully addressed my concerns. Specifically, their clarifications regarding causal graph robustness and the trade-offs in guidance scales were convincing. The paper’s strengths—particularly its clear motivation, the novel PAI+CTC framework, and its practical efficiency without full-model fine-tuning—are significant. The comprehensive evaluation and strong empirical results demonstrate its value to the community. I believe the technical contributions outweigh the minor limitations, and I recommend the paper for acceptance.

**Final Justification:**

Recommendation: Week Accept

The authors’ rebuttal successfully addressed my concerns. Specifically, their clarifications regarding causal graph robustness and the trade-offs in guidance scales were convincing. The paper’s strengths—particularly its clear motivation, the novel PAI+CTC framework, and its practical efficiency without full-model fine-tuning—are significant. The comprehensive evaluation and strong empirical results demonstrate its value to the community. I believe the technical contributions outweigh the minor limitations, and I recommend the paper for acceptance.

**Key Questions For Authors:**

1.How sensitive is performance to inaccuracies in the causal graph? Could you show results with a noisy or partially incorrect graph?

2.PAI projects continuous numerical attributes into the textual embedding space. Does the model struggle to represent highly granular, continuous numerical interventions (e.g., exactly 1.5x brain volume) given the discrete nature of the base model's pre-trained textual latent space?

3.Could you quantify the computational overhead introduced by the CTC loss during training? Does the requirement of generating positive/negative pairs across the batch significantly increase memory requirements?

**Limitations:**

yes

**Strengths And Weaknesses:**

Strengths:

1. Clear motivation and strong empirical grounding. The detailed motivational study convincingly shows the limitations of pure prompt-based or multi-concept T2I editing (NTI, MCPL) for counterfactual tasks. The paper then delivers a clean, principled solution.

2. Novel technical contributions. PAI + CTC is a simple yet effective way to align numeric causal attributes with diffusion latents and reduce spurious correlations. The adapter design keeps the backbone frozen, making the method highly practical and transferable.

3. Comprehensive evaluation. Strong quantitative results across four diverse datasets with four standard metrics. Qualitative results and ablations are thorough; attention guidance further improves localized editing.

4. Practical advantages. No full-model fine-tuning, batch training support, continuous attribute control, reversibility, and identity preservation are all superior to prior diffusion-SCM or VAE/GAN baselines. The method generalizes to DiT backbones.

5. Well-written and reproducible. Clear figures, detailed appendices, and public-style benchmarking make the work easy to follow and reproduce.

Weaknesses:

1. Assumption of a known causal graph: Like many causal generative models, the core framework relies heavily on a pre-defined causal graph. While Appendix G provides a brief mitigation study involving causal structure learning, the primary methodology might struggle in complex, real-world domains where the causal graph is misspecified or filled with unobserved confounders.

2. Guidance Scale Trade-offs: The ablation study on CFG reveals that while higher guidance scales (alpha = 5.0) improve intervention effectiveness, they can simultaneously degrade image realism (FID) and minimality (CLD), leading to slightly over-edited images.

---

> ### Author Rebuttal · Authors · 2026-03-30
>
> __Q1__: "__Known-graph assumption__"
>
> __R1__: Please refer our response to Reviewer-DeSP Q3, the same diffusion backbone can still perform disentangled editing without SCM, while different SCMs can be swapped at inference time to study how causal graph specification changes the propagation pattern.
>
> We also conduct an additional test on a real-world full-body human dataset; please refer to Reviewer-JH51 Q3-R3. These results suggest that our method has practical potential and remains reasonably robust under more complex images.
>
> __Q2__: "__Guidance-scale trade-off__"
>
> __R2__: We agree that higher guidance scales improve intervention effectiveness but can also reduce realism and minimality, resulting in slightly over-edited images. This trade-off is common in diffusion-based editing, since stronger guidance enforces the target condition more aggressively. In our experiments, as specified in Supplementary Sec. C.1, we use a guidance scale of 1.0 for Pendulum and ADNI, while for CelebA and CelebA-HQ we use a moderate guidance scale to balance edit strength and realism.
>
> __Q3__: "__Continuous control via PAI__"
>
> __R3__:  To test sensitivity to fine-grained and out-of-range interventions, we performed an out-of-range traversal on [ADNI](https://anonymous.4open.science/r/Rebuttal_materials-3763/pdfs/ADNI_traverse.pdf), which shows smooth variation for ventricular volume. In particular, although the training range is normalized to (0,1), editing remains smooth when extrapolated to (-1,2). As an additional check, the traversal results on [CelebA](https://anonymous.4open.science/r/Rebuttal_materials-3763/pdfs/traverse_celeba.pdf) also show smooth and controllable variation, even in the binary-attribute setting.
>
> These results suggest that, despite relying on a pretrained text encoder with discrete tokens, the model can still realize fine-grained continuous control because the intervention is performed on the attribute value and then injected into the token embedding, rather than being limited to token-level discretization.
>
> __Q4__: __CTC training overhead__"
>
> __R4__: In our framework, the positive/negative pairs in CTC do not require generating additional images or extra diffusion passes; they are constructed only from the placeholder token embeddings within the current batch. Concretely, positives are formed from the same placeholder token across different samples, while negatives are formed from different placeholder tokens across samples. Therefore, CTC only adds a lightweight similarity computation on token embeddings.
>
> In our measurement, the total FLOPs are 120.25 GFLOPs both with and without CTC, and the peak forward VRAM is 5.64 GB. This indicates that the extra computational and memory overhead from CTC is negligible relative to the overall diffusion backbone cost. We will clarify this point more explicitly in the revised manuscript.

---

> > ### Author Rebuttal · Reviewer_iauX · 2026-04-03
> >
> > The paper presents a well-motivated and principled approach (PAI + CTC) to handle counterfactual T2I editing. The motivational study is particularly strong, clearly identifying the failure modes of existing prompt-based methods.

---

> > > ### Author Response · Authors · 2026-04-05
> > >
> > > Dear Reviewer iauX,
> > >
> > > We sincerely thank you for acknowledging that our response has fully addressed your concerns, particularly regarding the additional graph ablation studies and the technical clarifications on guidance scale, PAI, and CTC. We greatly appreciate the time and effort you have devoted to reviewing our work, as well as your constructive feedback, which has helped improve the clarity and rigor of our manuscript.
> > >
> > > We are especially encouraged that you found the paper to be well motivated and principled. If you feel that the outcome of this discussion is consistent with your positive assessment and that your concerns have been fully resolved, we would be very grateful if you would kindly consider upgrading your score in the reviewer console to a higher recommendation. This would better reflect your positive feedback and align with the key strengths of our proposed Causal-Adapter recognized across all reviewers, including _clear motivation and strong empirical grounding_, _novel technical contributions_, _rigorous evaluation_, _practical efficiency and real-world utility_, as well as _clear presentation and reproducibility_.

---

### Official Review · Reviewer_DeSP · 2026-03-12

**Soundness:** 3
**Presentation:** 3
**Significance:** 3
**Originality:** 3
**Overall Recommendation:** 4
**Confidence:** 3

**Summary:**

This paper presents Causal-Adapter, a modular framework for counterfactual image generation built on top of a frozen text-to-image diffusion model. The method combines an explicit structural causal model over semantic attributes with a trainable adapter that injects causal signals into the diffusion backbone. Two additional components, Prompt-Aligned Injection (PAI) and Conditioned Token Contrast (CTC), are introduced to improve semantic alignment and attribute disentanglement. The paper evaluates the approach on Pendulum, CelebA, ADNI, and CelebA-HQ, and reports strong results on intervention effectiveness, realism, composition, and minimality.

**Compliance With Llm Reviewing Policy:**

Affirmed.

**Final Justification:**

My concerns have been adequately addressed. I raised my score to 4.

**Key Questions For Authors:**

1. In the SCM component, how are exogenous variables inferred for a factual instance before intervention? Are they abducted and reused, or sampled from a generic prior?
2. Why does the proposed CTC loss not collapse embeddings corresponding to different values of the same continuous attribute?
3. Can the authors include a strong non-causal baseline with the same diffusion backbone and comparable adapter capacity, but without SCM-based propagation?

**Limitations:**

yes

**Strengths And Weaknesses:**

Strengths
1. The paper addresses an important problem: generating visually plausible edits that are also causally consistent is a meaningful and timely direction.
2. The overall design is practical. Adapting a frozen diffusion backbone via a lightweight module is much more scalable than retraining a full generative model per dataset.
3. The empirical scope is broad, spanning synthetic, natural-image, medical, and high-resolution benchmarks.

Weaknesses
1. The paper’s central counterfactual claim is not fully justified. While the image latent is inverted through DDIM, it is unclear whether the SCM-level exogenous variables are actually abducted for each factual instance and then held fixed under intervention. As written, the method appears closer to interventional generation than to individualized counterfactual reasoning.
2. The CTC objective seems conceptually problematic for continuous attributes. Positive pairs are defined as the same attribute token across different samples, regardless of attribute value, which appears to encourage collapse across distinct values (e.g., different ages or ventricle volumes). The paper does not sufficiently explain why this would still preserve fine-grained control.
3. The experiments do not clearly isolate the benefit of the causal mechanism itself. On real datasets, there is no strong backbone-matched non-causal baseline, so it remains unclear how much of the gain comes from explicit causal modeling versus the pretrained diffusion prior and adapter-based conditioning.

---

> ### Author Rebuttal · Authors · 2026-03-30
>
> __Q1__: "__Justification on counterfactual claim__"
>
> __R1__: The exogenous variable $u_i$ in the equation is implemented as a bias term in the MLP, initialized from a Gaussian distribution. During inference, the intervened attributes are obtained from the frozen MLP. At the image level, instance-specific information is preserved through DDIM inversion, and the edited image is generated conditioned on both the image latents and the SCM outputs. Thus, instance-specific information is maintained in the diffusion latent space and the SCM-level exogenous variables are abducted and held fixed under intervention.
>
> __Q2__: "__Why CTC are not collapsing on different continuous values.__"
>
> __R2__: The positive pairs in CTC are not meant to collapse different continuous values of the same attribute. Rather, because the text encoder is frozen, after PAI injects sample-specific attribute values into the placeholder tokens, the embeddings may become entangled across attributes. The role of the positive pairs is to keep the same placeholder token semantically consistent across samples, i.e., to ensure that `@` still represents the same factor across the batch even after value-conditioned injection. __Fine-grained control is preserved because the actual continuous variation is still encoded by the injected value itself, while CTC mainly stabilizes token semantics and enfore disentanglement__. This argument is also supported empirically. The traversal results on [CelebA](https://anonymous.4open.science/r/Rebuttal_materials-3763/pdfs/traverse_celeba.pdf) show smooth fine-grained control even in the binary-attribute setting, and the out-of-range traversal on [ADNI](https://anonymous.4open.science/r/Rebuttal_materials-3763/pdfs/ADNI_traverse.pdf) shows the smooth variation for ventricular volume. In particular, although the training range is normalized to (0,1), editing remains smooth when extrapolated to (-1,2), which further suggests that CTC does not collapse within-attribute variation.
>
> __Q3__: "__Additional non-causal baseline experiments.__"
>
> __R3__: We have performed the requested ablation under three settings: (1) without SCM, (2) with an incorrect causal graph, and (3) with the correct causal graph.
>
> For Pendulum, the qualitative results are shown in [pendulum-dist](https://anonymous.4open.science/r/Rebuttal_materials-3763/pdfs/non-causal-pendulum.pdf). Without SCM, interventions still change the shadow-related variables, since pendulum/light and shadow are strongly correlated in the training data. The key issue, however, is that these changes are correlation-driven rather than graph-faithful: intervening on one factor can also spuriously modify another non-intervened factor (e.g., changing light also moves the pendulum). For SCM1, the additionally introduced incorrect edge propagates the effect from shadow position to shadow length, resulting in a different shadow evolution pattern. By contrast, with the correct graph SCM2, the propagation pattern becomes more consistent with the underlying physical law. Quantitative results:
>
> __Intervention on pendulum (`do(p)`) and light (`do(l)`), measured by MAE__
>
> |Method|do(p)-pend|do(p)-light|do(p)-shadow_l|do(p)-shadow_p|do(l)-pend|do(l)-light|do(l)-shadow_l|do(l)-shadow_p|
> |-|-|-|-|-|-|-|-|-|
> |Ours(w/o SCM)|0.159|0.060|0.143|0.086|0.183|0.173|0.235|0.155|
> |Ours(w/ SCM1)|0.018|0.048|0.039|0.031|0.040|0.062|0.104|0.033|
> |Ours(w/ SCM2)|0.014|0.045|0.028|0.030|0.035|0.041|0.051|0.033|
>
> Another test is performed on ADNI, with qualitative results shown in [ADNI](https://anonymous.4open.science/r/Rebuttal_materials-3763/pdfs/non-causal-ADNI.pdf). In the non-causal setting, intervention on brain volume rarely affects the ventricular region. In causal graph 1, because the edge brain volume->ventricular volume is inverted, interventions on parent factors (e.g., APOE, age, sex, brain volume) no longer clearly influence ventricular volume and contour. By contrast, the correct graph better preserves the expected propagation pattern. Quantitative results:
>
> __Intervention on brain (`do(b)`) and ventricle (`do(v)`), measured by MAE (brain, vent) and F1 (slide)__
>
> |Method|do(b)-brain|do(b)-vent|do(b)-slice|do(v)-brain|do(v)-vent|do(v)-slice|
> |-|-|-|-|-|-|-|
> |Ours (w/o SCM)|0.142|0.028|0.500|0.110|0.030|0.550|
> |Ours (w/ SCM1)|0.142|0.028|0.500|0.142|0.032|0.527|
> |Ours (w/ SCM2)|0.090|0.030|0.530|0.110|0.030|0.550|
>
> Our design separates SCM modeling from conditional disentanglement. This allows the framework to adapt to different causal graphs at inference time by swapping the trained SCM, without retraining the diffusion backbone. Together with the CelebA results discussed in our response to Reviewer-JH51 Q2-R2, __these experiments show that without a causal graph the model can still function as a disentangled editing method, whereas with the correct causal graph the generated counterfactuals better match the underlying data-generating structure and achieve better quantitative performance__.

---

> > ### Author Rebuttal · Reviewer_DeSP · 2026-04-01
> >
> > I raised my score to 4.

---

> > > ### Author Response · Authors · 2026-04-05
> > >
> > > Dear Reviewer DeSP,
> > >
> > > First of all, we sincerely thank you for acknowledging our rebuttal in time, particularly for addressing your concerns around counterfactual claim, CTC loss dynamics and additional ablation study. We appreciate your time and efforts, along with your constructive feedback which helped improve the clarity and solidity of our manuscript.
> > >
> > > If you feel that the outcome of this discussion process is consistent with your positive assessment, we would be very grateful if you would kindly consider upgrading score through reviewer console to a higher recommendation, to better align with your positive feedback and with the common strengths of our proposed Causal-Adapter recognized across all reviewers, including _the soundness of research question and solution_, _practical efficiency and real-world utility_, _the significance, scalability, and originality of our method_, as well as _the rigorous evaluation and clear presentation_.

---

### Official Review · Reviewer_Jh51 · 2026-03-15

**Soundness:** 3
**Presentation:** 3
**Significance:** 3
**Originality:** 3
**Overall Recommendation:** 4
**Confidence:** 2

**Summary:**

The paper proposes "Causal-Adapter," a modular framework that integrates an explicit Structural Causal Model (SCM) with a frozen text-to-image diffusion model for counterfactual image generation. The authors utilize a trainable adapter alongside Prompt-Aligned Injection (PAI) to map causal scalar values into text embeddings, and apply a Conditioned Token Contrastive (CTC) loss to enforce disentanglement. The method aims to perform faithful do-interventions on specific attributes (e.g., Age) while correctly updating causally linked attributes (e.g., Baldness) and preserving independent identity features.

**Compliance With Llm Reviewing Policy:**

Affirmed.

**Final Justification:**

My biggest concerns are in (1) missing baselines; (2) missing casual graph ablation listed in weakness 1 and 2 are addressed in the rebuttal. Raised my score to 4.

**Key Questions For Authors:**

See weakness section for details. My biggest concerns are in (1) missing baselines; (2) missing casual graph ablation listed in weakness 1 and 2.  Without those results, it is hard to convince me to vote for an acceptance.

**Limitations:**

Yes

**Strengths And Weaknesses:**

## Strengths
* **Insightful Attention Analysis:** The cross-attention visualizations, particularly in Figure 3c, are great. They clearly and directly demonstrate the individual and combined contributions of the PAI and CTC modules in aligning and disentangling semantic concepts within the diffusion latent space.
* **Simple, Efficient and Modular Design:** Modeling the SCM as prompt embeddings and injecting them into a frozen diffusion backbone via an adapter is a practical and efficient.

## Weaknesses & Major Concerns

**1. Outdated and Insufficient Baselines**
The paper evaluates its text-to-image editing capabilities primarily against Null-Textual Inversion (NTI) from 2023. The paper is missing critical comparisons to latest image editing frameworks e.g., Flux-Kontext, Qwen-Image-Edit, Nano Banana 2. The paper also misses recent causal image editing work such as  CausalCtrl (Cao et. al, MM'25).



**2. Faithfulness vs. Causal Graph Ablation**
The authors claim the model strictly obeys the explicit causal graph (the adjacency matrix A). However, it is unclear if the generated images genuinely follow the SCM math or simply default to the diffusion model's inherent data distribution (e.g., the strong statistical prior that "Old" correlates with "Bald").
The paper lacks an ablation with a *misspecified* causal graph. For example, if the valid edge from "Age" to "Baldness" is deleted, an intervention of `do(Age=Old)` should mathematically yield an SCM output where baldness does not increase. If the model still generates a bald person, the pre-trained statistical prior is overpowering the explicit SCM, meaning the attributes are not truly disentangled.  If such an ablation is not provided, then it is hard to justify why not just using a localized editing model or improve such a model?


**3. Lack of Open-Domain Generalization**. Evaluations are restricted to narrow, highly aligned datasets (CelebA, ADNI, Pendulum)  where attribute disentanglement is inherently easier due to minimal structural variation. It remains unclear if the Causal-Adapter and CTC loss can scale to complex, unaligned open-domain images without altering overall composition. Furthermore, because the framework relies on a predefined causal graph, the authors should address whether this requirement strictly bottlenecks the method to narrow domains, given that constructing a comprehensive causal graph for general, open-world images is practically intractable. Or simpler, can the method be evaluated in full body human generation cases instead of cropped faces only where variations are less restrictive?

**4. Details in FlowEdit support**
In Appendix E.4, the authors claim the framework generalizes to Stable Diffusion 3 (SD3) by replacing DDIM inversion with FlowEdit. More detailed regarding training, model design, and inference should be provided.

**5. Lack of Clarity on SCM Mechanisms and GT Ablation**
The explanation of how $f_i$ (the MLP mechanism) and $u_i$ (exogenous noise) interact with the diffusion pipeline is somewhat opaque in the main text. Furthermore, in Table 1 (row 6), the "w/ GT" baseline needs clearer mathematical formalization.

---

> ### Author Rebuttal · Authors · 2026-03-30
>
> __Q1__: "__Comparisons to latest image editing frameworks__"
>
> __R1__: Thank you for this helpful suggestion. For comparison against frontier image editing models, please refer our response to reviewer-GcCz Q2-R2.
>
> The causal graph in CausalCtrl is used to define the image-level diffusion process. Our method focuses on pixel attribute-level causal modeling for counterfactual generation. Thus CausalCtrl is related but not directly equivalent (CausalCtrl doesnot opensource the code for reproducing). We will add this discussion during revision.
>
> __Q2__: "__Ablation study on mis-specified causal graph.__"
>
> __R2__: Thanks for the suggestion. We compare our full model against an ablation variant, which does not incorporate the structural causal model (w/o SCM).
>
> We reported both qualitative results [celeba-disentangle](https://anonymous.4open.science/r/Rebuttal_materials-3763/pdfs/non-causal-celeba.pdf) and quantitative results:
>
> __Intervention on Age (do(a)) and Gender (do(g)), measured with accuracy from four classifiers (-Age, -Gen, -Br and -Bald)__
>
> |Method|do(a)-Age|do(a)-Gen|do(a)-Br|do(a)-Bald|do(g)-Age|do(g)-Gen|do(g)-Br|do(g)-Bald|
> |-|-|-|-|-|-|-|-|-|
> |Ours(w/o SCM)|60|100|100|100|94|80|100|100|
> |Ours(w/ SCM)|72|100|100|90|90|100|96|92|
>
> We observe that non-causal model only changes target attribute, while causal version matches the intended causal propagation. These results show that adding causal mechanism improves the intervention target attribution (e.g., `do(a)` on Age and `do(g)` on Gender), while also changing causal related attributes under intervention following SCM. This supports that causal mechanism is not simply adding stronger local edits, but is affecting the propagation pattern of the intervention.
>
>
> __Q3__: "__Open-Domain Generalization__"
>
> __R3__: We used a related human body dataset from [link](https://www.kaggle.com/datasets/snmahsa/human-images-dataset-men-and-women/data), randomly selected 100 samples, and manually annotated three attributes: young, gender, and beard. We then performed disentangled editing without using any causal graph, and compared our method with Qwen, NanoBanana, and Flux. The results are shown in [human-body](https://anonymous.4open.science/r/Rebuttal_materials-3763/pdfs/human_body.pdf).
>
> Qwen often over-edits and sometimes introduces partial cropping artifacts; Flux fails to edit age and gender. Our method shows changes under age and gender interventions, suggesting that it retains disentanglement ability beyond tightly aligned face datasets, while also preserving pose and background relatively well.
>
> These results suggest that the method has the potential to extend to more complex domains when sufficiently labeled attributes and training samples are available. We will add this discussion during revision.
>
> __Q4__: "__Details in FlowEdit support__"
>
> __R4__: The technical details are as follows:
>
> 1. Model Design: SD3 utilizes two CLIP and one T5 text encoders. PAI separately injects attribute information into all three token-embedding streams. For a dataset with $N$ attributes, we learn $N$ tokens per encoder, resulting in $3N$ injected embeddings that guide the SD3 MMDiT backbone via an adapter.
> 2. Training Objectives: We replace the standard LDM loss with a Flow-Matching loss ($L_{FM}$) following the SD3 recipe. To ensure semantic alignment across the three distinct embedding spaces, we use a joint objective:$$L_{total} = L_{FM} + \lambda \sum_{i=1}^{3} L_{CTC_i}$$where $L_{CTC_i}$ is the Contrastive Token Loss for each encoder's injected embeddings.
> 3. Inference via FlowEdit: Original FlowEdit requires source prompt embedding and target prompt embedding. We construct the source input by injecting the original attributes of the input image into the textual embeddings, and the target input by injecting the SCM-intervened attributes into the textual embeddings. We use 50 inversion and denoising steps, with a target guidance scale of 1.5 for generate counterfactuals.
>
> __Q5__: "__Clarity on SCM Mechanisms and GT Ablation__"
>
> __R5__: Our framework has two stages. First, the SCM operates in semantic attribute space and computes the intervened attribute vector according to the causal graph. Second, the adapter takes together with the DDIM-inverted latent of the factual image to generate the edited image. In our current formulation, $u_i$ serves as the bias term for mechanism learning inside the SCM, rather than being separately injected into the diffusion backbone. Implementation details are in Supplementary Sec. C.3.
>
> "w/ GT" indicates "w/ GT SCM", meaning we replace the learned SCM with the ground-truth structural equations of the synthetic Pendulum dataset. That is, the intervened attributes are computed directly from the oracle equations in [CausalDiffAE](https://arxiv.org/abs/2404.17735) page 11 and then fed into the same adapter for image generation. We will add this information during revision.

---

> > ### Author Rebuttal · Reviewer_Jh51 · 2026-04-06
> >
> > My biggest concerns are in (1) missing baselines; (2) missing casual graph ablation listed in weakness 1 and 2 are addressed in the rebuttal. Raised my score to 4.

---

### Official Review · Reviewer_GcCz · 2026-03-16

**Soundness:** 3
**Presentation:** 3
**Significance:** 2
**Originality:** 2
**Overall Recommendation:** 4
**Confidence:** 2

**Summary:**

The paper proposes Causal-Adapter, a framework designed to endow frozen diffusion models with causal reasoning capabilities through SCM-based modeling and dual regularization (PAI and CTC). This architecture successfully mitigates attribute entanglement and spurious correlations, ensuring that counterfactual generations are not only high-quality and identity-consistent but also faithfully reflect the underlying causal mechanisms.

**Compliance With Llm Reviewing Policy:**

Affirmed.

**Key Questions For Authors:**

Do the authors intend to release the source code? The absence of the implementation during the review process hinders the verification of reproducibility. Providing a partial codebase or sample data within the supplementary materials would significantly bolster the transparency and credibility of the findings.

**Limitations:**

see weaknesses

**Strengths And Weaknesses:**

Strengthes:
1. Plug-and-play adapters enable causal interventions on frozen diffusion backbones, obviating costly full-parameter fine-tuning
2. By integrating Structural Causal Models (SCM), the method ensures that attribute interventions follow a logical causal structure for consistent effect propagation

Weaknesses:
1. The evaluation is constrained to spatially fixed, single-object datasets, failing to demonstrate the robustness of the PAI module's spatial localization and attribute disentanglement in complex natural scenes characterized by multiple objects, occlusions, and stochastic layouts.
2. Given the increasing reasoning power of foundation models, please pinpoint a scenario where non-causal SOTAs fundamentally collapse regardless of prompt engineering
3. What are the qualitative and quantitative improvements when Causal-Adapter is applied to current SOTA backbones compared to the base models alone?

---

> ### Author Rebuttal · Authors · 2026-03-30
>
> __Q1__: "__Complex scenes editing__"
>
> __R1__: As suggested by Reviewer-JH51. We used a related real-world human body dataset ([link](https://www.kaggle.com/datasets/snmahsa/human-images-dataset-men-and-women/data)), randomly selected 100 samples, and manually annotated three attributes: young, gender, and beard. We then performed disentangled editing without using any causal graph, and compared our method with Qwen-Image-Edit, Nano Banana 2, and Flux1.Kontext.
>
> The results comparison are shown on link: [human-body](https://anonymous.4open.science/r/Rebuttal_materials-3763/pdfs/human_body.pdf). Qwen often over-edits and sometimes introduces partial cropping artifacts; Flux fails to edit age and gender. Our method shows changes under age and gender interventions, suggesting that it retains disentanglement ability beyond tightly aligned face datasets, while also preserving pose and background relatively well.
>
> These results suggest that the method has the potential to extend to more complex domains when sufficiently labeled attributes and training samples are available. We will add this discussion into the revised manuscript.
>
> __Q2__: "__Compare with non-causal SOTAs__"
>
> __R2__: Thanks for the suggestion. To address this, we additionally evaluated foundation models Nano Banana 2, Qwen-Image-Edit, and Flux1.Kontext. You can find result comparison on following links: [Pendulum](https://anonymous.4open.science/r/Rebuttal_materials-3763/pdfs/foundation_model_comparison_pendulum.pdf), [ADNI](https://anonymous.4open.science/r/Rebuttal_materials-3763/pdfs/foundation_model_comparison_ADNI.pdf), [CelebA-1](https://anonymous.4open.science/r/Rebuttal_materials-3763/pdfs/foundation_model_comparison_celeba1.pdf), and [CelebA-2](https://anonymous.4open.science/r/Rebuttal_materials-3763/pdfs/foundation_model_comparison_celeba2.pdf).
>
> We also provide quantitative comparison result on a CelebA subset:
>
> __Intervention on Age (do(a)) and Gender (do(g)), measured with accuracy from four classifiers (-Age, -Gen, -Br and -Bald) and LPIPS for visual similarity.__
>
> |Method|do(a)-Age|do(a)-Gen|do(a)-Br|do(a)-Bald|do(a)-LPIPS↓|do(g)-Age|do(g)-Gen|do(g)-Br|do(g)-Bald|do(g)-LPIPS↓|
> |-|-|-|-|-|-|-|-|-|-|-|
> |Ours|72|100|100|90|0.0725|90|100|96|92|0.0526|
> |Qwen-Image-Edit|62|84|96|86|0.1496|86|90|96|98|0.1579|
> |Flux.1-Kontext-Dev|34|98|96|92|0.0560|84|46|88|90|0.0480|
> |Nano Banana 2|44|98|94|92|0.0750|94|68|100|98|0.0820|
>
> __Intervention on Beard (do(br)) and Bald (do(bl)), measured with accuracy from four classifiers (Age, Gen, Br and Bald) and LPIPS for visual similarity.__
>
> |Method|do(br)-Age|do(br)-Gen|do(br)-Br|do(br)-Bald|do(br)-LPIPS↓|do(bald)-Age|do(bald)-Gen|do(bald)-Br|do(bald)-Bald|do(bald)-LPIPS↓|
> |-|-|-|-|-|-|-|-|-|-|-|
> |Ours|98|48|80|100|0.0282|70|86|92|40|0.1087|
> |Qwen-Image-Edit|84|36|92|98|0.1326|56|82|96|94|0.1892|
> |Flux.1-Kontext-Dev|78|44|88|98|0.0665|60|80|92|76|0.0976|
> |Nano Banana 2|92|50|86|100|0.0215|66|86|96|94|0.1200|
>
> Compared with frontier enterprise-level large models that trained on complex pipelines, our method is simple, practical efficient, while remains competitive under interventions on age, gender, and beard. These results are consistent with our original findings: __recent foundation models still struggle with fine-grained causal interventions, especially for continuous factors__. They often modify visually salience instead of faithfully following the intended intervention, which makes them less suitable for counterfactual generation.
>
> __Q3__: "__Gain over SOTA backbones__"
>
> __R3__: Thank you for this important point. We compare SD3+FlowEdit (using prompt engineering alone) against our Causal-Adapter on CelebA-HQ ([Link](https://anonymous.4open.science/r/Rebuttal_materials-3763/pdfs/SD3_ablation.pdf)).
>
> Since our method performs control at the attribute level, we design the FlowEdit prompt to explicitly encode the same semantic relations for fair comparison, e.g., “A portrait photo of a female person not smiling, with mouth closed, and wearing eyeglasses …”, so that the relation between smile and mouth open is also reflected in the prompt. We also use the same hyparameters setting for the 2 methods (1.5 guidance scale, 50 denosing steps). The quantatitive results are shown as follows:
>
> __Intervention on smile (`do(s)`) and glasses (`do(g)`), measured by accuracy__
>
> | Method | do(s)-smile | do(s)-glass | do(g)-smile | do(g)-glass |
> |---|---:|---:|---:|---:|
> | FlowEdit | 46 | 98 | 94 | 16 |
> | Ours | 90 | 100 | 100 | 96 |
>
> Both the qualitative and quantitative results show that __Causal-Adapter taming the backbone to perform more causally faithful editing while also improving generation fidelity, compared with base model alone__.
>
> __Q4__: "__Code release__"
>
> __R4__: You can find our submitted code with clear documentation through the anonymous link: [code](https://anonymous.4open.science/r/Rebuttal_materials-3763/code/README.md). We will open-source the code upon manuscript acceptance.

---

### Decision · Program_Chairs · 2026-04-30

**Decision:**

Accept (regular)

**Comment:**

Causal-Adapter is a modular framework that equips frozen text-to-image diffusion models with structured causal interventions for counterfactual image generation, enabling faithful attribute editing while preserving core identity and achieving state-of-the-art performance.

The rebuttal addressed most of the reviewers’ concerns, and the paper was ultimately accepted unanimously. Congratulations.